# On the Benefits of Weight Normalization for Overparameterized Matrix Sensing

**Yudong Wei**  **Liang Zhang**  **Bingcong Li**[*]  **Niao He**[*]
ETH Zurich
yudwei@ethz.ch
{liang.zhang, bingcong.li, niao.he}@inf.ethz.ch

## Abstract

While normalization techniques are widely used in deep learning, their theoretical understanding remains relatively limited. In this work, we establish the benefits of (generalized) weight normalization (WN) applied to the overparameterized matrix sensing problem. We prove that WN with Riemannian optimization achieves linear convergence, yielding an *exponential* speedup over standard methods that do not use WN. Our analysis further demonstrates that both iteration and sample complexity improve polynomially as the level of overparameterization increases. To the best of our knowledge, this work provides the first characterization of how WN leverages overparameterization for faster convergence in matrix sensing.

## 1 Introduction

Normalization schemes, such as layer, batch, and weight normalization, are essential in modern deep networks and have proven highly effective for stabilizing training in both vision and language models (Ioffe & Szegedy, 2015; Ba et al., 2016; Salimans & Kingma, 2016). Despite their practical success, theoretical explanations of why they work remain elusive, even for relatively simple problems.

This work focuses on weight normalization (WN), which decouples parameters (i.e., variables) into directions and magnitudes, and then optimizes them separately. It has recently regained considerable attention because of the seamless integration with LoRA (Hu et al., 2022), leading to several powerful strategies for parameter-efficient fine-tuning of large language models; see e.g., (Liu et al., 2024; Lion et al., 2025). Yet, theoretical support for WN remains relatively limited. Prior results in (Wu et al., 2020) show that WN applied to overparameterized least squares induces implicit regularization towards the minimum $\ell_2$-norm solution. The implicit regularization of WN on diagonal linear neural networks is studied in (Chou et al., 2024). WN is also observed to reduce Hessian spectral norm and improve generalization in deep networks (Cisneros-Velarde et al., 2025).

Our work broadens the understanding of WN by establishing its merits for the overparameterized matrix sensing problem. The goal is to recover a low-rank positive semi-definite (PSD) matrix $\mathbf{A} \in \mathbb{S}_+^m$ from linear measurements. In the vanilla formulation without WN, one can exploit the low-rankness of ground-truth matrix, i.e., $r_A := \mathsf{rank}(\mathbf{A}) \ll m$ for efficient parameterization. Specifically, we can optimize for $\mathbf{Y} \in \mathbb{R}^{m \times r}$ such that $\mathbf{Y}\mathbf{Y}^\top \approx \mathbf{A}$ (Burer & Monteiro, 2005). The overparameterized regime $r > r_A$ is of interest due to the need of exact recovery without knowing $r_A$ a priori. This problem has wide applications in machine learning and signal processing (Candès et al., 2013), and serves as a popular testbed for theoretical deep learning given its non-convexity and rich loss landscape; see e.g., (Li et al., 2018; Jin et al., 2023; Arora et al., 2019).

Without WN, prior work (Xiong et al., 2024) establishes a sublinear lower bound on the convergence rate when the above sensing problem is optimized via gradient descent (GD), even with infinite data samples. We circumvent this lower bound by i) extending WN for coping with matrix variables; and, ii) proving that applying this generalized WN with Riemannian gradient descent (RGD) enables a *linear* convergence rate in the finite sample regime, leading to an *exponential* improvement. Remarkably, *WN leverages higher level of overparameterization to achieve both faster convergence*

---

[*]Equal supervision.

Table 1: Comparison with existing algorithms for overparameterized matrix sensing. WN gives exact convergence with linear rate. "E.C." is the abbreviation of "exact convergence", that is, whether the reconstruction error bound will go to zero when the iteration number $t \to \infty$. UB and LB are short for upper and lower bound, respectively. OP stands for overparameterization.

| Algorithm | WN | E.C. | Initialization | Convergence Rate | Faster with OP |
|-----------|----|------|----------------|------------------|----------------|
| GD (UB) (Stöger & Soltanolkotabi, 2021) | ✗ | ✗ | Small & random | N/A | - |
| GD (LB) (Xiong et al., 2024) | ✗ | ✓ | Small & random | $\Omega\left(\frac{\kappa^2}{\log(mr_A^2)t}\right)$ | ✗ |
| RGD (**Theorem** 3.2) | ✓ | ✓ | Random | $\exp\left(-\mathcal{O}\left(\frac{(r-r_A)^4}{\kappa^4 m^2 r^2 r_A}t\right)\right)$ | ✓ |

*and lower sample complexity*. To the best of our knowledge, this is the first theoretical result demonstrating that normalization benefits from overparameterization.

More concretely, our contributions are summarized as follows:

❖ **Exponentially faster rate.** For overparameterized matrix sensing problems, we prove that randomly initialized WN achieves a linear convergence rate of $\exp(-\mathcal{O}(\frac{(r-r_A)^4}{\kappa^4 m^2 r^2 r_A}t))$, where $\kappa$ is the condition number of the ground-truth matrix $\mathbf{A}$. This linear rate is exponentially faster than the sublinear lower bound $\Omega\left(\frac{\kappa^2}{\log(mr_A^2)t}\right)$ obtained without WN. Moreover, additional overparameterization in WN provides quantifiable benefits: the iteration complexity scales down polynomially as the overparameterization level $r$ increases; see Table 1 for a summary.

❖ **Two-phase convergence behavior.** We further investigate the optimization trajectory and reveal a two-phase behavior in WN. The iterates first move from a random initialization to a neighborhood around global optimum, potentially traversing and escaping from several saddle points in polynomial time. Our results demonstrate that this phase ends faster with additional overparameterization. Once iterates approach the global optimum, a local phase begins. With the benign loss landscape shaped by WN, we prove that a linear convergence rate can be obtained.

❖ **Empirical validation.** We conduct experiments on overparameterized matrix sensing using both synthetic and real-world datasets. The numerical results corroborate our theoretical findings.

## 1.1 RELATED WORK

**Overparameterized matrix sensing.** Overparameterized matrix sensing arises from many machine learning and signal processing applications such as collaborative filtering and phase retrieval (Schafer et al., 2007; Srebro & Salakhutdinov, 2010; Candès et al., 2013; Duchi et al., 2020). The problem is now a canonical benchmark in theoretical deep learning, mainly because the loss landscape is riddled with saddle points and lacks global smoothness or a global PL condition. Convergence analyses for various algorithms on its population loss, i.e., matrix factorization, can be found in (Ward & Kolda, 2023; Li et al., 2025; Kawakami & Sugiyama, 2021). Small random initialization in overparameterized matrix sensing has been studied in (Stöger & Soltanolkotabi, 2021; Jin et al., 2023; Xiong et al., 2024; Xu et al., 2023), while (Ma et al., 2023; Zhuo et al., 2024; Cheng & Zhao, 2024) are based on spectral initialization. Besides saddle escaping under small initialization, another intriguing phenomenon is that overparameterization can exponentially slow the convergence of GD compared to the exactly parameterized case (Zhuo et al., 2024; Xiong et al., 2024). Our work proves that WN avoids this slowdown and achieves an improved rate. Moreover, additional overparameterization leads to faster convergence and lower sample complexity.

**Riemannian optimization.** Riemannian optimization is naturally connected to WN for learning the direction variables, which are constrained on a smooth manifold, e.g., a sphere. Existing literature has extended gradient-based methods to problems with smooth manifold constraints; see e.g., (Absil et al., 2008; Smith, 2014; Mishra et al., 2012; Boumal, 2023). This work follows standard notions of Riemannian gradient descent (RGD). In its simplest form, RGD iteratively moves along the negative direction of the Riemannian gradient, obtained by projecting the Euclidean gradient onto the tangent space, and then maps the iterate back to the manifold via a retraction.

**Notational conventions.** Bold capital (lowercase) letters denote matrices (column vectors); $(\cdot)^\top$ and $\|\cdot\|_\mathsf{F}$ refer to transpose and Frobenius norm of a matrix; $\|\cdot\|$ denotes the spectral ($\ell_2$) norm for matrices (vectors); $\langle \mathbf{A}, \mathbf{B} \rangle = \mathsf{Tr}(\mathbf{A}^\top \mathbf{B})$ represents the standard matrix inner product; and, $\sigma_i(\mathbf{A})$ denotes the $i$-th largest singular value of matrix $\mathbf{A}$. Moreover, $\mathbb{S}^m$ and $\mathbb{S}^m_+$ denote symmetric and positive semi-definite (PSD) matrices of size $m \times m$, respectively.

## 2 PROBLEM FORMULATION

We focus on applying WN to the symmetric low-rank matrix sensing problem. The objective is to recover a low-rank and positive semi-definite (PSD) matrix $\mathbf{A} \in \mathbb{S}^m_+$ from a collection of $n$ data $\{(\mathbf{M}_i, y_i)\}_{i=1}^n$, where each feature matrix $\mathbf{M}_i \in \mathbb{S}^m$ is symmetric and the corresponding label is $y_i = \mathsf{Tr}(\mathbf{M}_i^\top \mathbf{A})$. For notational conciseness, we let $\mathbf{y} = [y_1, \ldots, y_n]^\top \in \mathbb{R}^n$ and define a linear mapping $\mathcal{M} : \mathbb{S}^m \mapsto \mathbb{R}^n$ with $[\mathcal{M}(\mathbf{A})]_i = \mathsf{Tr}(\mathbf{M}_i^\top \mathbf{A})$. Given that $r_A = \mathsf{rank}(\mathbf{A}) \ll m$, a parameter economical formulation is based on the Burer-Monteiro factorization (Burer & Monteiro, 2005) that introduces a matrix $\mathbf{Y} \in \mathbb{R}^{m \times r}$ such that $\mathbf{Y}\mathbf{Y}^\top$ approximates $\mathbf{A}$ accurately. This leads to

$$\min_{\mathbf{Y} \in \mathbb{R}^{m \times r}} \frac{1}{4}\|\mathcal{M}(\mathbf{Y}\mathbf{Y}^\top) - \mathbf{y}\|^2. \tag{1}$$

Despite its seemingly simple formulation, the loss landscape contains saddle points, hence achieving a *global* optimum from a random initialization is nontrivial. Moreover, overparameterization, i.e., $r > r_A$, is often considered in practice to ensure exact recovery of $\mathbf{A}$ without prior knowledge of its rank. It is established in (Xiong et al., 2024) that such overparameterization induces optimization challenges even in the population setting ($n \to \infty$). In particular, a lower bound of GD shows that $\|\mathbf{Y}_t \mathbf{Y}_t^\top - \mathbf{A}\|_\mathsf{F}$ converges no faster than $\Omega(1/t)$, where $t$ is the iteration number. This rate is exponentially slower than the linear one when $r_A$ is known to employ $r = r_A$ (Ye & Du, 2021).

**Applying WN to problem (1).** For a vector variable, WN decouples it into direction and magnitude, and optimizes them separately. Extending this idea to matrix variables in (1), we leverage polar decomposition to write $\mathbf{Y} = \mathbf{X}\tilde{\boldsymbol{\Theta}}$, where $\mathbf{X} \in \mathsf{St}(m, r)$ lies in a Stiefel manifold and $\tilde{\boldsymbol{\Theta}} \in \mathbb{S}^r_+$. Here, the Stiefel manifold $\mathsf{St}(m, r)$ is defined as $\{\mathbf{X} \in \mathbb{R}^{m \times r} | \mathbf{X}^\top \mathbf{X} = \mathbf{I}_r\}$. One can geometrically interpret $\mathbf{X}$ as orthonormal bases for an $r$-dimensional subspace, thus representing "directions", and $\tilde{\boldsymbol{\Theta}}$ captures the "magnitude" of a matrix. Substituting $\mathbf{Y}$ in (1), we arrive at

$$\min_{\mathbf{X}, \tilde{\boldsymbol{\Theta}}} \frac{1}{4}\|\mathcal{M}(\mathbf{X}\tilde{\boldsymbol{\Theta}}\tilde{\boldsymbol{\Theta}}^\top \mathbf{X}^\top) - \mathbf{y}\|^2 \quad \text{s.t.} \quad \mathbf{X} \in \mathsf{St}(m, r), \ \tilde{\boldsymbol{\Theta}} \in \mathbb{S}^r_+.$$

The above problem can be further simplified by i) merging $\tilde{\boldsymbol{\Theta}}\tilde{\boldsymbol{\Theta}}^\top$ into a single matrix $\boldsymbol{\Theta} \in \mathbb{S}^r_+$; and ii) relaxing the PSD constraint on $\boldsymbol{\Theta}$ to only symmetry, i.e., $\boldsymbol{\Theta} \in \mathbb{S}^r$. This relaxation achieves the same global objective in the overparameterized regime, yet significantly improves computational efficiency by avoiding SVDs or matrix exponentials needed for optimizing over PSD cones (Vandenberghe & Boyd, 1996; Todd, 2001). In sum, applying WN gives the objective

$$\min_{\mathbf{X}, \boldsymbol{\Theta}} f(\mathbf{X}, \boldsymbol{\Theta}) := \frac{1}{4}\|\mathcal{M}(\mathbf{X}\boldsymbol{\Theta}\mathbf{X}^\top) - \mathbf{y}\|^2 \quad \text{s.t.} \quad \mathbf{X} \in \mathsf{St}(m, r), \ \boldsymbol{\Theta} \in \mathbb{S}^r. \tag{2}$$

For convenience, we continue to refer to this generalized variant as WN, since it aligns with the direction-magnitude decomposition paradigm. Similar reformulations of (1) have appeared in (Mishra et al., 2014; Levin et al., 2025). The former empirically studies the faster convergence on matrix completion problems, while the latter tackles local geometry around stationary points. Our work, on the other hand, characterizes the behaviors of WN along the entire trajectory and clarifies its interaction with overparameterization.

### 2.1 SOLVING WN VIA RIEMANNIAN OPTIMIZATION

Generalizing the vector WN[1] on matrix problems, Riemannian optimization is adopted for coping with the manifold constraint $\mathbf{X} \in \mathsf{St}(m, r)$. We simply treat the manifold as an embedded one

---

[1]While the practical update rule of WN (Salimans & Kingma, 2016, eq. (4)) lies between Riemannian and Euclidean optimization, (Wu et al., 2020, Lemma 2.2) shows that the limiting flow is Riemannian flow.

---

**Algorithm 1** Riemannian gradient descent (RGD) for solving WN (2)

---

1: **Input:** Initial point $\mathbf{X}_0 \in \mathsf{St}(m, r), \boldsymbol{\Theta}_0 \in \mathbb{S}^r$, stepsizes $\eta, \mu$
2: **for** $t = 0, 1, \dots, T$ **do**
3:     Calculate $\mathbf{G}_t$, the Riemannian gradient of $\mathbf{X}_t$, via (3)
4:     Update $\mathbf{X}_{t+1}$ via (4)                       *// The direction variable*
5:     Calculate $\mathbf{K}_t := \nabla_{\boldsymbol{\Theta}} f(\mathbf{X}_{t+1}, \boldsymbol{\Theta}_t)$ via (9)
6:     Update $\boldsymbol{\Theta}_{t+1}$ via (5)                       *// The magnitude variable*
7: **end for**
8: **Output:** $\mathbf{X}_{T+1}, \boldsymbol{\Theta}_{T+1}$

---

in Euclidean space. Extensions to other geometry are straightforward. To optimize the direction variable $\mathbf{X}_t$, let $\tilde{\mathbf{G}}_t := \nabla_{\mathbf{X}} f(\mathbf{X}_t, \boldsymbol{\Theta}_t)$ denote the Euclidean gradient on $\mathbf{X}_t$ (a detailed expression is given in (8) of Appendix C). The Riemannian gradient for $\mathbf{X}_t$ can be written as

$$\mathbf{G}_t := (\mathbf{I}_m - \mathbf{X}_t \mathbf{X}_t^\top)\tilde{\mathbf{G}}_t + \frac{\mathbf{X}_t}{2}(\mathbf{X}_t^\top \tilde{\mathbf{G}}_t - \tilde{\mathbf{G}}_t^\top \mathbf{X}_t). \tag{3}$$

Further applying the polar retraction[2] to ensure feasibility, the update for $\mathbf{X}_t$ is given by

$$\mathbf{X}_{t+1} = (\mathbf{X}_t - \eta \mathbf{G}_t)(\mathbf{I}_r + \eta^2 \mathbf{G}_t^\top \mathbf{G}_t)^{-1/2} \tag{4}$$

where $\eta > 0$ is the stepsize. Detailed derivations of (3) and (4) are deferred to Appendix C. Note that polar retraction is used here for theoretical simplicity. Shown later sections, other popular retractions for Stiefel manifolds such as QR and Cayley[3] share almost identical performance numerically.

An alternative update method is adopted for the magnitude variable $\boldsymbol{\Theta}_t$. Denote its gradient as $\mathbf{K}_t := \nabla_{\boldsymbol{\Theta}} f(\mathbf{X}_{t+1}, \boldsymbol{\Theta}_t)$, whose expression can be found from (9) in Appendix C. We use GD with a stepsize $\mu > 0$ to optimize $\boldsymbol{\Theta}_t$, i.e.,

$$\boldsymbol{\Theta}_{t+1} = \boldsymbol{\Theta}_t - \mu \mathbf{K}_t. \tag{5}$$

This update ensures feasibility of the symmetric constraint on $\boldsymbol{\Theta}_t, \forall t \geq 0$, whenever initialized with $\boldsymbol{\Theta}_0 \in \mathbb{S}^r$; see a proof in Lemma F.9. The step-by-step procedure for solving (2) is summarized in Algorithm 1, and it is termed as RGD for future reference.

## 3   On the benefits of WN

This section demonstrates that WN delivers exact convergence at a linear rate for overparameterized matrix sensing (2) and leverages additional overparameterization to yield faster optimization and lower sample complexity. Recall that the rank of $\mathbf{A}$ is denoted by $r_A$. Let the compact SVD of $\mathbf{A}$ be $\mathbf{A} = \mathbf{U}\boldsymbol{\Sigma}\mathbf{U}^\top$, where $\mathbf{U} \in \mathbb{R}^{m \times r_A}$ and $\boldsymbol{\Sigma} \in \mathbb{S}_+^{r_A}$. Without loss of generality, we assume $\sigma_1(\boldsymbol{\Sigma}) = 1$ and $\sigma_{r_A}(\boldsymbol{\Sigma}) = 1/\kappa$ with $\kappa \geq 1$ denoting the condition number. We will use the restricted isometry property (RIP) (Recht et al., 2010), a standard assumption in matrix sensing, in our proofs; see more in, e.g., (Zhang et al., 2021; Stöger & Soltanolkotabi, 2021; Xu et al., 2023; Xiong et al., 2024).

**Definition 3.1 (Restricted Isometry Property (RIP))** *The linear mapping $\mathcal{M}(\cdot)$ is $(r, \delta)$-RIP, with $\delta \in [0, 1)$, if for all matrices $\mathbf{A} \in \mathbb{S}^m$ of rank at most $r$, it satisfies*

$$(1 - \delta)\|\mathbf{A}\|_{\mathsf{F}}^2 \leq \|\mathcal{M}(\mathbf{A})\|^2 \leq (1 + \delta)\|\mathbf{A}\|_{\mathsf{F}}^2.$$

RIP ensures that the linear measurement approximately preserves the Frobenius norm of low-rank matrices. This property has been shown to hold for a wide range of measurement operators. For example, when $\mathbf{M}_i$ is symmetric Gaussian, a sample size of $n = \mathcal{O}(mr/\delta^2)$ suffices to guarantee that $(r, \delta)$-RIP holds with high probability. A detailed discussion with illustrative examples is provided in Appendix A.3. With these preparations, we are ready to uncover the merits of WN.

---

[2]Let $\mathbf{X} \in \mathsf{St}(m, r)$ and a point in its tangent space $\mathbf{G} \in \mathcal{T}_{\mathbf{X}}\mathsf{St}(m, r)$. The polar retraction for $\mathbf{X} + \mathbf{G}$ is given by $\mathcal{R}_{\mathbf{X}}(\mathbf{G}) = (\mathbf{X} + \mathbf{G})(\mathbf{I}_r + \mathbf{G}^\top \mathbf{G})^{-1/2}$.

[3]See e.g., (Absil et al., 2008), for more detailed discussions on retractions.

### 3.1 MAIN RESULTS

We consider WN under random initialization, meaning that $\mathbf{X}_0$ is chosen uniformly at random from the manifold $\mathsf{St}(m, r)$. One possible approach is to set $\mathbf{X}_0 = \mathbf{Z}_0(\mathbf{Z}_0^\top \mathbf{Z}_0)^{-1/2}$, where the entries of $\mathbf{Z}_0 \in \mathbb{R}^{m \times r}$ are i.i.d. Gaussian random variables $\mathcal{N}(0, 1)$ (Chikuse, 2012).

**Theorem 3.2** *Consider solving the WN-aided sensing problem (2) initialized with random $\mathbf{X}_0 \in \mathsf{St}(m, r)$ and $\mathbf{\Theta}_0 \in \mathbb{S}^r$ satisfying $\|\mathbf{\Theta}_0\| \leq 2$. Assume that $r_A \leq \frac{m}{2}$ and $\mathcal{M}(\cdot)$ is $(r + r_A + 1, \delta)$-RIP with $\delta = \mathcal{O}\big(\frac{(r - r_A)^6}{\kappa^2 m^3 r^4 r_A}\big)$. Algorithm 1 using stepsizes $\eta = \mathcal{O}\big(\frac{(r - r_A)^4}{\kappa^2 m^2 r^2 r_A}\big)$ and $\mu = 2$ generates a sequence $\{\mathbf{X}_t, \mathbf{\Theta}_t\}_{t=0}^\infty$. With high probability over the initialization, this sequence proceeds in two distinct phases, separated by a burn-in time $t_0$ with an upper bound $\mathcal{O}\big(\frac{\kappa^4 m^4 r^4 r_A^2}{(r - r_A)^8}\big)$:*

*i) Saddle phase. For some universal constant $c_2 \in (0, 1)$, it follows that*

$$\|\mathbf{X}_t \mathbf{\Theta}_t \mathbf{X}_t^\top - \mathbf{A}\|_{\mathsf{F}} \leq 2\sqrt{r_A - \frac{c_2(r - r_A)^8 t}{\kappa^4 m^4 r^4 r_A} + 1}, \quad 1 \leq t \leq t_0.$$

*ii) Linearly convergent phase. For some universal constant $c_3 \in (0, 1)$, it is guaranteed that*

$$\|\mathbf{X}_t \mathbf{\Theta}_t \mathbf{X}_t^\top - \mathbf{A}\|_{\mathsf{F}} \leq 3\left(1 - \frac{c_3(r - r_A)^4}{\kappa^4 m^2 r^2 r_A}\right)^{t - t_0}, \quad \forall t \geq t_0 + 1.$$

This theorem showcases the two regimes of convergence behavior using randomly initialized RGD. In the first phase, the upper bound of reconstruction error $\|\mathbf{X}_t \mathbf{\Theta}_t \mathbf{X}_t^\top - \mathbf{A}\|_{\mathsf{F}}$ is proved to monotonically decrease over iterations. This seemingly slow convergence arises from the potential saddle escape that will be discussed in the next section. Eventually, RGD achieves a linear rate until exact convergence, i.e., $\lim_{t \to \infty} \|\mathbf{X}_t \mathbf{\Theta}_t \mathbf{X}_t^\top - \mathbf{A}\|_{\mathsf{F}} = 0$. Next, we break down Theorem 3.2 to demonstrate the benefits of WN for the overparameterized matrix sensing from two different perspectives.

**Optimization benefits of WN** include i) faster convergence rate, and ii) less stringent initialization requirements. Theorem 3.2 shows that WN achieves exact convergence to the ground-truth matrix $\mathbf{A}$ with a linear rate. In contrast, without WN, the convergence behavior of randomly initialized GD on (1) is weaker. Specifically, (Stöger & Soltanolkotabi, 2021) shows that GD can only attain a constant reconstruction error with early stopping, but not guarantee last-iteration convergence. On the other hand, (Xiong et al., 2024) establishes a lower bound for exact recovery of GD, giving a sublinear dependence on $t$; see a detailed comparison in Table 1. In addition, our guarantee of this linear rate is obtained without strict requirements on initialization, which stands in stark contrast to the non-WN setting, where the magnitude of random initialization must be carefully controlled, often inversely proportional to $\kappa$ (Stöger & Soltanolkotabi, 2021; Jin et al., 2023; Xu et al., 2023).

**WN makes overparameterization a friend.** Because the additional parameters induce computation and memory overheads, it is natural to expect more gains from overparameterization. It can be seen from Table 1 that GD does not benefit from overparameterization, while the benefits of overparameterization for WN are twofold. Setting $r = p r_A$ for some $p > 1$, one can rewrite the upper bound of the burn-in time $t_0$ as $\mathcal{O}\big(\frac{\kappa^4 m^4 p^4}{(p-1)^8 r_A^2}\big)$, which decreases polynomially with $p$. In the linearly convergent phase, WN achieves a convergence rate of $\exp\big(-\mathcal{O}\big(\frac{(p-1)^4 r_A}{\kappa^4 m^2 p^2} t\big)\big)$, which is also faster with a larger $p$. In terms of iteration complexity, this translates into a polynomial improvement with the level of overparameterization. To quantitatively understand the merits of overparameterization, we consider two cases. In the mildly overparameterized regime, where $r = r_A + c$ for some constant $c = \mathcal{O}(1)$, the convergence rate reads $\exp\big(-\mathcal{O}(\frac{t}{\kappa^4 m^2 r_A^3})\big)$. When the level of overparameterization increases to $r = c r_A$, the rate improves to $\exp\big(-\mathcal{O}(\frac{r_A t}{\kappa^4 m^2})\big)$. Through comparison, it is readily seen that additional overparameterization yields up to a factor of $\mathcal{O}(r_A^4)$ improvement in the exponent. On the statistical side, the sample complexity of WN is determined by the RIP assumption on $\mathcal{M}(\cdot)$. Under the Gaussian design, as detailed in Appendix A.3, the RIP holds w.h.p. when $n = \mathcal{O}\big(\frac{\kappa^4 m^7 r^9 r_A^2}{(r - r_A)^{12}}\big)$. Notably, the sample complexity $n$ reduces polynomially as $r$ increases. In particular, following the same analysis as for the convergence rate, this reduction can reach up to a factor of $\mathcal{O}(r_A^{12})$.

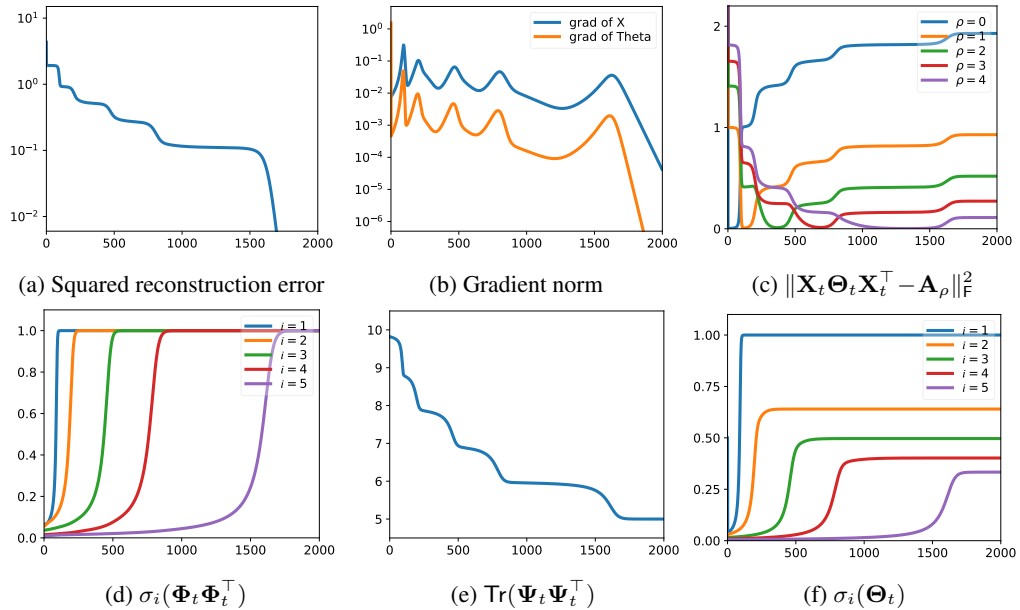

Figure 1: The saddle-to-saddle (i.e., sequential learning) behaviors in WN. The x-axis corresponds to the iteration number, and the y-axis follows the subfigure title. (a) Each plateau signifies a saddle; (b) gradient norm at saddles drops by orders; (c) saddles strongly relate to the best rank-$\rho$ approximation of $\mathbf{A}$; (d) sequential learning in the alignment between $\mathbf{X}_t$ and $\mathbf{U}$; (e) sequential learning in the alignment between $\mathbf{X}_t$ and $\mathbf{U}_\perp$; and, (f) sequential pattern in the magnitude variable $\mathbf{\Theta}_t$.

## 3.2 A GEOMETRY PROOF SKETCH

The proof of Theorem 3.2, while involved, admits a clear geometrical interpretation originated from the direction-magnitude decoupling of WN. Here, we only focus on the "direction" $\mathbf{X}_t$ to gain more intuition. Given that both $\mathbf{X}_t$ and $\mathbf{U}$ are bases of a linear subspace, it is desirable that $\mathsf{span}(\mathbf{U}) \subset \mathsf{span}(\mathbf{X}_t)$ at convergence. Equivalently, the principle angles between $\mathsf{span}(\mathbf{U})$ and $\mathsf{span}(\mathbf{X}_t)$ at optimal should all be 0. This can be depicted via the alignment matrix $\mathbf{\Phi}_t := \mathbf{U}^\top \mathbf{X}_t$, whose singular values coincide with the cosine of these principle angles (Björck & Golub, 1973). Our proof builds upon this and shows that $\mathsf{Tr}(\mathbf{\Phi}_t \mathbf{\Phi}_t^\top) \to r_A$, i.e., the two subspaces align.

The convergence unfolds in two phases. In the first phase, $\mathsf{Tr}(\mathbf{\Phi}_t \mathbf{\Phi}_t^\top)$ grows from near 0 (due to random initialization) to near optimal $r_A - 0.5$. Through consecutive lemmas, it can be shown that $\mathsf{Tr}(\mathbf{\Phi}_{t+1} \mathbf{\Phi}_{t+1}^\top) - \mathsf{Tr}(\mathbf{\Phi}_t \mathbf{\Phi}_t^\top) \geq \mathcal{O}(\frac{(r - r_A)^8}{\kappa^4 m^4 r^4 r_A})$. This monotonic increase in alignment ensures a polynomial time to escape (potential) saddles, and also translates to the decreasing bound of $\|\mathbf{X}_t \mathbf{\Theta}_t \mathbf{X}_t^\top - \mathbf{A}\|_\mathsf{F}$ in Theorem 3.2. The second phase starts after $\mathsf{Tr}(\mathbf{\Phi}_t \mathbf{\Phi}_t^\top) > r_A - 0.5$, where the alignment error $r_A - \mathsf{Tr}(\mathbf{\Phi}_t \mathbf{\Phi}_t^\top) = \mathsf{Tr}(\mathbf{I}_{r_A} - \mathbf{\Phi}_t \mathbf{\Phi}_t^\top)$ decreases linearly to 0.

## 4 DIVING DEEPER INTO THE SADDLE PHASE

Next, we take a closer look at the convergence of RGD on WN in the saddle phase, that is $t \leq t_0$, or equivalently $\mathsf{Tr}(\mathbf{\Phi}_t \mathbf{\Phi}_t^\top) \leq r_A - 0.5$. Our numerical experiments in Figure 1 indicate that RGD traverse a sequence of saddles. The saddle-to-saddle behavior is known for GD on (1) (Li et al., 2021; Jin et al., 2023). This section shows that this behavior persists for (2), yet can be faster with a higher level of overparameterization. To bypass the randomness associated with $\mathbf{M}_i$, we begin by pinpointing the saddles for the population loss, i.e., problem (2) in the infinite data limit $n \to \infty$. More precisely, the objective is given by $f_\infty(\mathbf{X}, \mathbf{\Theta}) = \frac{1}{4} \|\mathbf{X}\mathbf{\Theta}\mathbf{X}^\top - \mathbf{A}\|_\mathsf{F}^2$.

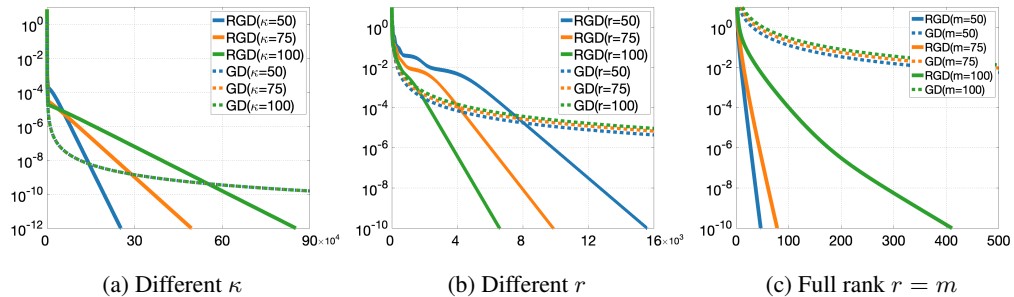

(a) Different $\kappa$          (b) Different $r$          (c) Full rank $r = m$

Figure 2: Convergence comparison of RGD on WN and GD on (1) under varying problem conditions (squared reconstruction error vs. iteration). (a): WN enables RGD to converge linearly regardless of $\kappa$; (b): with WN, larger $r$ leads to a shorter saddle phase and a faster convergence rate; (c): WN converges remarkably fast in the full rank case $r = m$.

**Lemma 4.1** *For a given $\rho \in \{0, 1, \ldots, r_A - 1\}$, let $\mathbf{A}_\rho$ be the best rank-$\rho$ approximation of $\mathbf{A}$, i.e., $\mathbf{A}_\rho = \arg\min_{\mathsf{rank}(\hat{\mathbf{A}}) \leq \rho} \|\hat{\mathbf{A}} - \mathbf{A}\|_{\mathsf{F}}^2$. In particular, we let $\mathbf{A}_0 = \mathbf{0}$. A point $(\mathbf{X}, \mathbf{\Theta})$ is a saddle of the population loss $f_\infty$ if $\mathbf{X}\mathbf{\Theta}\mathbf{X}^\top = \mathbf{A}_\rho$ and $\mathsf{Tr}(\mathbf{X}^\top \mathbf{U}\mathbf{U}^\top \mathbf{X}) = \rho$.*

Lemma 4.1 indicates that the saddles of $f_\infty$ are closely related to the best rank-$\rho$ approximation of $\mathbf{A}$. It further suggests that a saddle-to-saddle dynamic is aligned with incremental learning[4]: the algorithm successively learns $\mathbf{A}_\rho$ for increasing $\rho$ until the ground-truth matrix is recovered. Lemma 4.2 below shows that in the finite-sample regime, the saddles of $f_\infty$ also have small gradient norm on $f$, i.e., no larger than $\mathcal{O}\left(\frac{(r-r_A)^6}{\kappa^2 m^2 r^4 r_A}\right)$ under the parameter choices of Theorem 3.2.

**Lemma 4.2** *Assume that $\mathcal{M}(\cdot)$ is $(r + r_A + 1, \delta)$-RIP, and $\|\mathbf{\Theta}\| \leq 2$, the finite sample loss in (2) satisfies $\|\nabla_{\mathbf{X}}^R f_\infty(\mathbf{X}, \mathbf{\Theta}) - \nabla_{\mathbf{X}}^R f(\mathbf{X}, \mathbf{\Theta})\|_{\mathsf{F}} \leq 12m\delta$ and $\|\nabla_{\mathbf{\Theta}} f_\infty(\mathbf{X}, \mathbf{\Theta}) - \nabla_{\mathbf{\Theta}} f(\mathbf{X}, \mathbf{\Theta})\|_{\mathsf{F}} \leq \frac{3}{2}m\delta$. Here, $\nabla_{\mathbf{X}}^R$ denotes the Riemannian gradient with respect to $\mathbf{X}$.*

Having characterized the saddles, we now turn to the saddle-to-saddle trajectory in Figure 1. This figure traces the optimization trajectory of Algorithm 1 on WN with $m = 300$, $r_A = 5$, $r = 10$, and $\kappa = 3$, with more details shown in Appendix G.1. Figure 1a plots the squared reconstruction error across iterations. Each plateau marks escape from a saddle, as confirmed by the small gradient norm shown in Figure 1b. Figure 1c further shows that these saddles are exactly those characterized in Lemma 4.1, where $\|\mathbf{X}_t \mathbf{\Theta}_t \mathbf{X}_t^\top - \mathbf{A}_\rho\|_{\mathsf{F}}^2$ for $\rho \in \{0, 1, \ldots, r_A - 1\}$ stays close to $0$ sequentially. In other words, each saddle escape corresponds to leaving the neighborhood of $\mathbf{A}_\rho$.

In addition, the optimization variables, geometrically interpretable as direction and magnitude, also exhibit a sequential learning behavior. For the direction variable $\mathbf{X}_t$, the singular values of $\mathbf{\Phi}_t \mathbf{\Phi}_t^\top$ (which characterize the squared cosine of the principle angles between $\mathbf{X}_t$ and $\mathbf{U}$) are visualized in Figure 1d. Further, let $\mathbf{U}_\perp \in \mathbb{R}^{m \times (m - r_A)}$ be an orthonormal basis for the orthogonal complement of $\mathsf{span}(\mathbf{U})$. The alignment of $\mathbf{X}_t$ and $\mathbf{U}_\perp$ is plotted in Figure 1e, with the alignment matrix defined as $\mathbf{\Psi}_t := \mathbf{U}_\perp^\top \mathbf{X}_t$. The singular values of the magnitude variable $\mathbf{\Theta}_t$ are plotted in Figure 1f. A clear sequential learning pattern is observed among all these figures.

Lastly, we highlight that *polynomial* time is needed to escape all saddles: Theorem 3.2 bounds the duration of this phase to be at most $\mathcal{O}\left(\frac{\kappa^4 m^4 r^4 r_A^2}{(r - r_A)^8}\right)$ iterations. This bound decreases with larger $r$, indicating that overparameterization facilitates saddle escape under WN.

## 5   NUMERICAL EXPERIMENTS

Numerical experiments using both synthetic and real-world data are conducted in this section to validate our theoretical findings for WN on overparameterized matrix sensing problems. In the experiments with synthetic data, the target matrix is generated as $\mathbf{A} = \mathbf{U}\mathbf{\Sigma}\mathbf{U}^\top \in \mathbb{R}^{m \times m}$, where

---

[4]Also known as deflation; see e.g., (Ge et al., 2021; Anandkumar et al., 2014; Seddik et al., 2023)

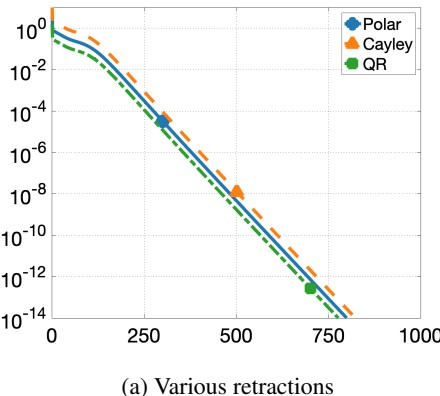
(a) Various retractions

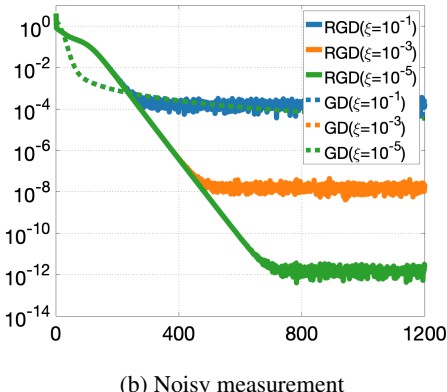
(b) Noisy measurement

Figure 3: Additional numerical results of WN (squared reconstruction error vs. iteration).

$\mathbf{U} \in \mathbb{R}^{m \times r_A}$ is a random matrix with orthonormal columns, and $\mathbf{\Sigma} \in \mathbb{S}_+^{r_A}$ is a diagonal matrix with condition number $\kappa$. In the image reconstruction experiments, the target matrix $\mathbf{A}$ is directly constructed from the underlying image. For the measurements, we use $n$ independent random Gaussian feature matrices $\{\mathbf{M}_i\}_{i=1}^n$ to ensure RIP. More details on setups are deferred to Appendix G.2.

## 5.1 FASTER CONVERGENCE OF WN

Under various choices of the condition number $\kappa$, we compare the convergence behavior of RGD on problem (2) with random initialization, against that of GD on (1) with small random initialization.

In this experiment, we consider target matrices with large condition numbers, i.e., $\kappa \in \{50, 75, 100\}$. We set $m = 10, r = 5, r_A = 3$, and $n = 60000$. The squared reconstruction error versus the number of iterations is plotted in Figure 2a. We observe that WN enables RGD to converge linearly to zero after a saddle phase, regardless of the condition number $\kappa$. This is consistent with our theoretical result in Theorem 3.2. In contrast, GD slows down to a sublinear rate after its initial phase, yielding substantially larger errors at the same iteration count.

## 5.2 ON THE BENEFIT OF OVERPARAMETERIZATION

Next, we demonstrate that WN leverages overparameterization for faster convergence. To this end, we consider randomly initialized problem instances of (1) and (2) under different $r$.

In this experiment, we focus on a setting with $m = 300$, $r_A = 5$, and $\kappa = 10$. The level of overparameterization is chosen from $r \in \{50, 75, 100\}$, and the number of measurements is set to $n = 50000$. RGD is run with random initialization and GD is run with small random initialization. The squared reconstruction error versus the number of iterations is plotted in Figure 2b.

The results show that under WN, RGD converges faster as $r$ increases. This behavior is consistent with our analysis in Theorem 3.2. In comparison, although the theoretical convergence rate given by (Xiong et al., 2024) is independent of $r$, our empirical results indicate that a larger $r$ leads to slightly slower convergence of GD. Moreover, Figure 2b clearly shows that saddle escape becomes faster with larger $r$, as reflected in shorter plateaus or earlier onset of linear convergence. Figure 2b also shows that a larger $r$ in WN leads to a steeper slope in the linearly convergent phase, demonstrating that additional overparameterization prompts a faster rate. This aligns well with our theoretical observations and discussions in Sections 3.1 and 4.

We also demonstrate that WN is remarkably effective in the full rank setting with $r = m$ in Figure 2c, where the convergence on three instances with $m = r \in \{50, 75, 100\}, r_A \in \{10, 15, 20\}, \kappa \in \{1, 15, 50\}$, and $n = 30000$ is plotted. The faster convergence arises from the fact that at initialization, $\mathbf{X}_0 \in \mathsf{St}(m, m)$ already aligns with the target subspace spanned by $\mathbf{U}$, i.e., $\mathsf{Tr}(\mathbf{I}_{r_A} - \mathbf{\Phi}_0 \mathbf{\Phi}_0^\top) = 0$. Equivalently, this is the case where only the magnitude $\mathbf{\Theta}$ is optimized. This faster convergence implies that learning the correct direction (i.e., $\mathbf{U}$) is more challenging than magnitude.

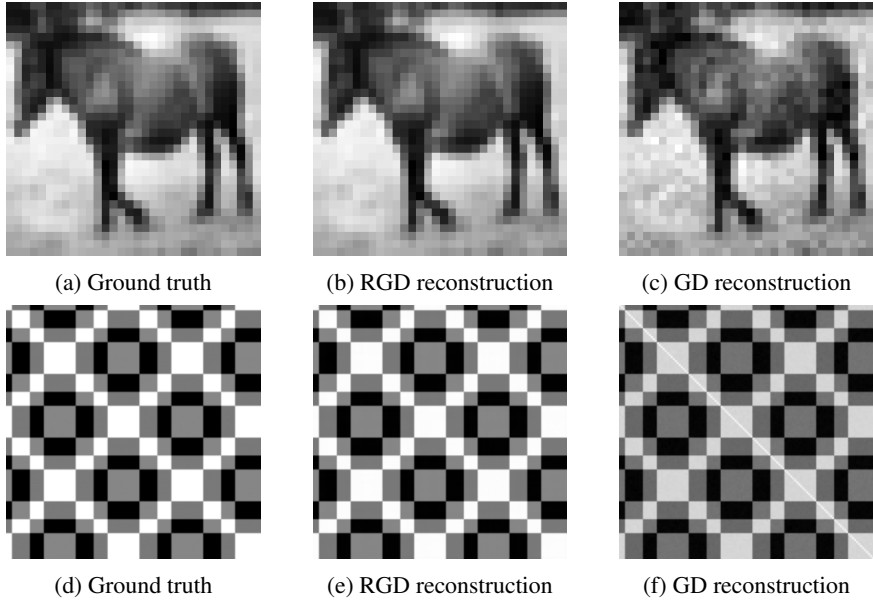

(a) Ground truth      (b) RGD reconstruction      (c) GD reconstruction

(d) Ground truth      (e) RGD reconstruction      (f) GD reconstruction

Figure 4: The advantages of WN on image reconstruction.

### 5.3 ADDITIONAL EXPERIMENTS

Moreover, additional experiments reveal other interesting behaviors of WN.

**Alternative manners of retraction.** Although our algorithm for WN tackles only the polar retraction, other popular retractions share similar performance. In Figure 3a, we plot the performance of Algorithm 1 with different manners for retraction, such as Cayley and QR, on an instance of (2) with $m = 10, r = 5, r_A = 3, \kappa = 2, n = 1000$. The three curves of squared reconstruction errors nearly coincide. For better visualization, we scale the errors of Cayley and QR by 3 and $1/3$, respectively.

**Noisy measurements.** To examine the robustness of WN, we consider a setting with corrupted labels, i.e., $y_i = \mathsf{Tr}(\mathbf{M}_i^\top \mathbf{A}) + b_i$ for i.i.d. Gaussian noise $b_i \sim \mathcal{N}(0, \xi^2)$. Figure 3b compares WN with the vanilla problem (1) under the choices of $\xi = 10^{-1}$, $\xi = 10^{-3}$, and $\xi = 10^{-5}$. It can be seen that RGD holds a linear rate under all choices of $\xi$, and the final squared reconstruction error stabilizes around $\mathcal{O}(\xi^2)$. On the other hand, the error of GD is mainly confined by its slow convergence rate. This demonstrates that the power of WN carries to noisy settings as well.

### 5.4 IMAGE RECONSTRUCTION EXPERIMENTS

Lastly, we evaluate the advantages of WN on two image reconstruction problems.

The first experiment follows (Duchi et al., 2020) to consider a generalized phase retrieval problem on a $32 \times 32$ horse image from the CIFAR-10 dataset (Krizhevsky & Hinton, 2009). The image is converted to grayscale and vectorized as $\boldsymbol{a} \in \mathbb{R}^{1024}$. Standard lifting reformulation converts this problem to a sensing problem on a rank-one ground-truth matrix $\mathbf{A} = \boldsymbol{a}\boldsymbol{a}^\top \in \mathbb{S}_+^{1024}$; see (Candès & Li, 2014). The second considers direct matrix sensing of a structured image given by $\mathbf{A} \in \mathbb{S}_+^{128}$ with $r_A = 2$. In both cases, we set the overparameterization level to $r = 100$ and use $n = 50000$ feature matrices. RGD and GD are randomly initialized and run for $t_{\mathrm{RGD}} = 100, t_{\mathrm{GD}} = 200$ iterations in both experiments to make the overall runtime comparable; see Appendix G.2.2 for details.

The reconstructions from the two experiments are presented in Figure 4. As shown, WN enables RGD to achieve more accurate recovery of the ground truth compared to GD. These results demonstrate that WN provides a significant improvement for image reconstruction problems.

## 6 CONCLUSION

This work provides new theoretical insights into the role of weight normalization (WN) in over-parameterized matrix sensing. We prove that randomly initialized WN with proper Riemannian optimization guarantees a linear rate, yielding an exponential improvement on overparameterized sensing problems without WN. Moreover, we show that overparameterization can be exploited under WN to achieve faster optimization and lower sample complexity. Our analysis also reveals a two-phase convergence behavior, with detailed characterizations of faster convergence in both phases. Numerical experiments on both synthetic and real-world data further validate our findings. Future work includes extending these results to broader non-convex learning settings, such as tensor problems (Tong et al., 2022), and developing new algorithms that build on WN.

## REPRODUCIBILITY STATEMENT

We have taken several steps to ensure the reproducibility of our work. For the theoretical results, we provide complete proofs of all theorems and lemmas in Appendices E and F. For the empirical results, Section 5 contains detailed descriptions of the numerical experiments, and the experimental setups are presented in Appendix G.

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

## USAGE OF LLMS

The authors conducted all aspects of the research, including conception, theoretical proofs, experimentation, analysis, and writing of the manuscripts. Large language models (LLMs) were employed exclusively for the purpose of language refinement.

## A    MORE ON BACKGROUNDS

### A.1    POLAR DECOMPOSITION

The definition of the polar decomposition is provided below; see (Golub & Van Loan, 2013, Section 9.4.3) for a detailed discussion and theoretical background.

**Definition A.1** *The polar decomposition of a matrix $\mathbf{X} \in \mathbb{R}^{m \times r}$ with $m \geq r$ is defined as*

$$\mathbf{X} = \mathbf{UP},$$

*where $\mathbf{U} \in \mathbb{R}^{m \times r}$ has orthonormal columns and $\mathbf{P} \in \mathbb{S}_+^r$ is a positive semi-definite matrix.*

This decomposition can be interpreted as expressing $\mathbf{X}$ as the product of directions ($\mathbf{U}$) and a magnitude part ($\mathbf{P}$). It is unique when $\mathbf{X}$ has full column rank.

### A.2    RIEMANNIAN OPTIMIZATION

Riemannian optimization provides a principled framework for optimization problems whose variables are constrained on a smooth manifold, such as spheres, Stiefel and Grassmann manifolds.

Let M be a smooth manifold and $f : \mathsf{M} \to \mathbb{R}$ be a differentiable objective function. At any point $\mathbf{X} \in \mathsf{M}$, the feasible directions form the tangent space $\mathcal{T}_{\mathbf{X}}\mathsf{M}$. The Riemannian gradient, denoted $\nabla^R f(\mathbf{X})$, is defined as the orthogonal projection of the Euclidean gradient $\nabla f(\mathbf{X})$ onto $\mathcal{T}_{\mathbf{X}}\mathsf{M}$. Intuitively, it is the direction of steepest descent that remains compatible with the manifold geometry.

A basic Riemannian gradient descent (RGD) iteration consists of two steps:

$$\mathbf{G}_t = \nabla^R f(\mathbf{X}_t) \in \mathcal{T}_{\mathbf{X}_t}\mathsf{M}, \qquad \mathbf{X}_{t+1} = \mathcal{R}_{\mathbf{X}_t}(\mathbf{G}_t),$$

where $\mathcal{R}_{\mathbf{X}_t} : \mathcal{T}_{\mathbf{X}_t}\mathsf{M} \to \mathsf{M}$ is retraction, namely a smooth mapping satisfying $\mathcal{R}_{\mathbf{X}_t}(\mathbf{0}) = \mathbf{X}_t$ and whose curve $c(s) = \mathcal{R}_{\mathbf{X}_t}(s\mathbf{G}_t)$ has initial velocity $c'(0) = \mathbf{G}_t$. Such a mapping brings a tangent step back to the manifold while approximating the true geodesic. Retractions admit simple closed forms on many manifolds, such as normalization on the sphere.

This framework generalizes standard gradient methods to curved spaces while preserving their intuitive interpretation. As a result, Riemannian optimization has become a popular tool for problems with geometric constraints, and is supported by a rich theoretical foundation and efficient algorithms; see, e.g., (Absil et al., 2008; Smith, 2014; Mishra et al., 2012; Boumal, 2023).

### A.3    RESTRICTED ISOMETRY PROPERTY (RIP)

The RIP condition (Recht et al., 2010) in Definition 3.1 is a standard assumption in matrix sensing, ensuring that the linear measurement operator approximately preserves the Frobenius norm of low-rank matrices. This property has been verified to hold with high probability for a wide variety of measurement operators. The following lemma establishes RIP for Gaussian design measurements.

**Lemma A.2** (Candès & Plan, 2011) *If $\mathcal{M}(\cdot)$ is a Gaussian random measurement ensemble, i.e., the entries of $\{\mathbf{M}_i\}_{i=1}^n \subset \mathbb{S}^m$ are independent up to symmetry with diagonal elements sampled from $\mathcal{N}(0, 1/n)$ and off-diagonal elements from $\mathcal{N}(0, 1/2n)$, then with high probability, $\mathcal{M}(\cdot)$ is $(r, \delta_r)$-RIP, as long as $n \geq Cmr/\delta_r^2$ for some sufficiently large universal constant $C > 0$.*

### A.4    OVERPARAMETERIZATION IN OTHER NONCONVEX ESTIMATION PROBLEMS

Beyond matrix sensing, the role of overparameterization has also been examined in a range of nonconvex estimation problems. For matrix completion, (Ma & Fattahi, 2024) proves that the vanilla

gradient descent with small initialization converges to the ground truth matrix without requiring any explicit regularization, even in the overparameterized scenario. In Gaussian mixture learning, (Zhou et al., 2025) establishes that Gradient EM achieves global convergence at a polynomial rate with polynomial samples, when the model is mildly overparameterized. For neural network training, (Xu & Du, 2023) shows that in the problem of learning a single neuron with ReLU activation, randomly initialized gradient descent can suffer from an exponential slowdown when the model is overparameterized. These studies illustrate that overparameterization appears across diverse problem settings, while its precise influence on the convergence behavior is problem-dependent.

## A.5 PRECONDITIONED ALGORITHMS

Preconditioning is a popular tool for BM-based matrix sensing to improve the convergence rate. For example, considering the problem

$$\min_{\mathbf{Y} \in \mathbb{R}^{m \times r}} g(\mathbf{Y}) := \frac{1}{4} \|\mathcal{M}(\mathbf{Y}\mathbf{Y}^\top) - \mathbf{y}\|^2,$$

preconditioned gradient descent (PrecGD) (Zhang et al., 2021) and scaled gradient descent (ScaledGD) (Tong et al., 2021) adopt the following updates:

$$\text{PrecGD:} \quad \mathbf{Y}_{t+1} = \mathbf{Y}_t - \eta \nabla g(\mathbf{Y}_t)(\mathbf{Y}_t^\top \mathbf{Y}_t + \lambda \mathbf{I})^{-1},$$

$$\text{ScaledGD:} \quad \mathbf{Y}_{t+1} = \mathbf{Y}_t - \eta \nabla g(\mathbf{Y}_t)(\mathbf{Y}_t^\top \mathbf{Y}_t)^{-1}.$$

Since $(\mathbf{Y}_t^\top \mathbf{Y}_t)$ may be singular in the overparameterized regime, ScaledGD cannot be directly applied. The variant ScaledGD($\lambda$) proposed in (Xu et al., 2023) addresses this by using a similar update to PrecGD with a particular choice of $\lambda$. Next, we provide a detailed comparison of proposed approach with these preconditioned methods.

**Comparison with PrecGD.** PrecGD establishes only a local convergence guarantee, requiring initialization sufficiently close to the ground truth. Although (Zhang et al., 2021) also discusses globally convergent variants of PrecGD, they rely on gradient perturbations of the form $\mathbf{Y}_{t+1} = \mathbf{Y}_t - \eta[\nabla g(\mathbf{Y}_t)(\mathbf{Y}_t^\top \mathbf{Y}_t + \lambda \mathbf{I})^{-1} + \zeta_t]$ with some random noise $\zeta_t$ to escape potential saddles. In addition, they require a multi-stage switching mechanism that monitors several quantities, including $\|\nabla f(\mathbf{Y}_t)\|_\mathsf{F}$, $\lambda_{\min}(\nabla^2 f(\mathbf{Y}_t))$ and $\lambda_{\min}(\mathbf{Y}_t^\top \mathbf{Y}_t)$. Notably, $\nabla^2 f(\mathbf{Y}_t) \in \mathbb{R}^{m \times r \times m \times r}$ is a fourth-order tensor, which is memory-intensive. In fact, one key motivation for adopting the Burer–Monteiro factorization is to reduce the parameter dimension to $mr$ by exploiting the low-rank structure, whereas forming such a tensor negates this advantage. Moreover, computing $\lambda_{\min}(\nabla^2 f(\mathbf{Y}_t))$ is especially expensive in large-scale matrix sensing problems. In contrast, our algorithm has a global convergence guarantee from random initialization without requiring perturbations or multi-stage switching rules.

**Comparison with ScaledGD($\lambda$).** ScaledGD($\lambda$) requires a carefully controlled small initialization with magnitude $\alpha$. To reach accuracy $\varepsilon$, the method must satisfy $\alpha \leq \mathcal{O}(\varepsilon^3)$, implying that exact convergence ($\varepsilon = 0$) can not be guaranteed. Moreover, ScaledGD($\lambda$) requires an $(r_A + 1, \delta)$-RIP condition with $\delta \leq \mathcal{O}(\kappa^{-C_\delta})$ for a sufficiently large constant $C_\delta$. In contrast, we just need $\delta \leq \mathcal{O}(\kappa^{-2})$. As a result, our sample complexity is significantly smaller, especially when the condition number $\kappa$ is large, i.e., in ill-conditioned settings.

**Comparison of the benefits of overparameterization.** A major advantage of our approach is that a higher level of overparameterization can not only improve the convergence rate, but also reduce the required sample complexity. In contrast, ScaledGD($\lambda$) does not show explicit benefits from increasing $r$. PrecGD's local convergence improves only with a square-root dependence on $r$, which is much weaker than the polynomial improvement achieved by our algorithm. In addition, PrecGD does not gain reduction in sample complexity from additional overparameterization.

**Comparison of potential extensions.** A further benefit is the generality of our weight normalization formulation. This way of factorization can be directly applied to arbitrary low-rank PSD optimization problems. In contrast, PrecGD and ScaledGD($\lambda$) rely on second-order information of the loss function $g$, restricting their applicability beyond matrix sensing.

**Comparison of iteration complexity.** For the iteration complexity, PrecGD and ScaledGD($\lambda$) achieve better $\kappa$-dependence than our algorithm. However, the faster rates partially arise from the

quasi-newton nature of their update, where $(\mathbf{Y}_t^\top \mathbf{Y}_t + \lambda \mathbf{I})$ is an estimation to Hessian. On the other hand, our algorithm is a purely first-order method, and we believe that incorporating second-order information can improve the convergence of our algorithm as well. To further validate this point, we initialize a preconditioned version of proposed approach as follows.

**WN with preconditioner.** Motivated by the designs of PrecGD and ScaledGD($\lambda$), we also explore incorporating second-order information to improve convergence empirically. To this end, we first derive a preconditioner for RGD.

For any direction $\mathbf{H} \in \mathbb{R}^{m \times r}$, the Hessian with respect to $\mathbf{X}$ takes the form

$$\nabla_{\mathbf{X}}^2 f(\mathbf{H}, \mathbf{\Theta}) = [\mathcal{M}^*\mathcal{M}(\mathbf{H}\mathbf{\Theta}\mathbf{X}^\top + \mathbf{X}\mathbf{\Theta}\mathbf{H}^\top)]\mathbf{X}\mathbf{\Theta} + [\mathcal{M}^*\mathcal{M}(\mathbf{X}\mathbf{\Theta}\mathbf{X}^\top - \mathbf{A})]\mathbf{H}\mathbf{\Theta}$$
$$= \mathcal{M}^*\mathcal{M}(\mathbf{H}\mathbf{\Theta}\mathbf{X}^\top)\mathbf{X}\mathbf{\Theta} + \mathcal{M}^*\mathcal{M}(\mathbf{X}\mathbf{\Theta}\mathbf{H}^\top)\mathbf{X}\mathbf{\Theta} + \mathcal{M}^*\mathcal{M}(\mathbf{X}\mathbf{\Theta}\mathbf{X}^\top - \mathbf{A})\mathbf{H}\mathbf{\Theta}.$$

When the RIP constant $\delta \ll 1$, we can approximate $\mathcal{M}^*\mathcal{M} \approx \mathcal{I}$, which yields

$$\nabla_{\mathbf{X}}^2 f(\mathbf{H}, \mathbf{\Theta}) \approx \mathbf{H}\mathbf{\Theta}^2 + \mathbf{X}\mathbf{\Theta}\mathbf{H}^\top\mathbf{X}\mathbf{\Theta} + (\mathbf{X}\mathbf{\Theta}\mathbf{X}^\top - \mathbf{A})\mathbf{H}\mathbf{\Theta}.$$

Near the optimum, the residual term satisfies $(\mathbf{X}\mathbf{\Theta}\mathbf{X}^\top - \mathbf{A})\mathbf{H}\mathbf{\Theta} \approx \mathbf{0}$. If we further ignore the term $\mathbf{X}\mathbf{\Theta}\mathbf{H}^\top\mathbf{X}\mathbf{\Theta}$, the Hessian is well approximated by $\mathbf{H}\mathbf{\Theta}^2$. Vectorizing both sides gives $\text{vec}(\nabla_{\mathbf{X}}^2 f(\mathbf{H}, \mathbf{\Theta})) \approx (\mathbf{I} \otimes \mathbf{\Theta}^2) \cdot \text{vec}(\mathbf{H})$, which implies the approximated Hessian structure $\nabla_{\mathbf{X}}^2 f \approx \mathbf{I} \otimes \mathbf{\Theta}^2$. Motivated by this approximation, we design a preconditioner $(\mathbf{\Theta}^2 + \lambda\mathbf{I})^{-1}$ for RGD, i.e., replacing the Euclidean gradient of $\mathbf{X}_t$ by $\nabla_{\mathbf{X}} f(\mathbf{X}_t, \mathbf{\Theta}_t)(\mathbf{\Theta}_t^2 + \lambda\mathbf{I})^{-1}$, where $\lambda$ is a regularization parameter that may be changed from iteration to iteration. We call this variant preconditioned Riemannian gradient descent (PrecRGD).

We conduct experiments on an instance of $m = 50, r = 40, r_A = 3, \kappa = 10$, with $n = 8000$ sensing matrices as well as a second instance with a larger overparameterization level $r = 45$ while keeping all other parameters fixed; see Appendix G.3 for more details on setups. As shown in Figure 5, PrecRGD exhibits a higher convergence rate than RGD, demonstrating that our preconditioner design is highly effective for faster convergence under the WN formulation. Moreover, PrecRGD outperforms PrecGD and achieves a convergence behavior that is comparable to ScaledGD($\lambda$). This suggests that WN is fully compatible with preconditioning tech-

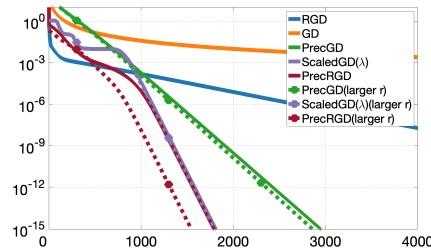

Figure 5: Comparison with preconditioned algorithms (squared reconstruction error vs. iteration).

niques, and we believe that this is a promising direction for further improving the convergence rate. Furthermore, when the level of overparameterization $r$ increases, PrecRGD converges even faster, while PrecGD and ScaledGD($\lambda$) do not show explicit improvements when $r$ increases. This again highlights the benefits of overparameterization for WN.

# B OTHER EXTENSIONS

## B.1 EXTENSION TO ASYMMETRIC PROBLEMS

We have shown the high effectiveness of WN in overparameterized matrix sensing problems, and its underlying parameterization reveals that it is broadly applicable to a wide range of low-rank optimization tasks, even when the target matrix is asymmetric. Consider a general matrix $\mathbf{A} \in \mathbb{R}^{m \times n}$ with Burer-Monteiro factorization $\mathbf{Y}_1 \mathbf{Y}_2^\top$, where $\mathbf{Y}_1 \in \mathbb{R}^{m \times r}$ and $\mathbf{Y}_2 \in \mathbb{R}^{n \times r}$. We can apply the Polar decomposition to each factor, i.e., $\mathbf{Y}_1 = \mathbf{X}_1 \mathbf{\Theta}_1, \mathbf{Y}_2 = \mathbf{X}_2 \mathbf{\Theta}_2$, with $\mathbf{X}_1 \in \text{St}(m, r), \mathbf{X}_2 \in \text{St}(n, r)$ and $\mathbf{\Theta}_1, \mathbf{\Theta}_2 \in \mathbb{S}_+^r$. By combining the two magnitude matrices into $\mathbf{\Theta} = \mathbf{\Theta}_1 \mathbf{\Theta}_2^\top \in \mathbb{R}^{r \times r}$, we obtain the representation $\mathbf{A} = \mathbf{X}_1 \mathbf{\Theta} \mathbf{X}_2^\top$.

This generality suggests that WN has substantial potential in a variety of applications, including collaborative filtering (Schafer et al., 2007), compressed sensing (Candès et al., 2013), matrix completion (Recht, 2011), and other related problems.

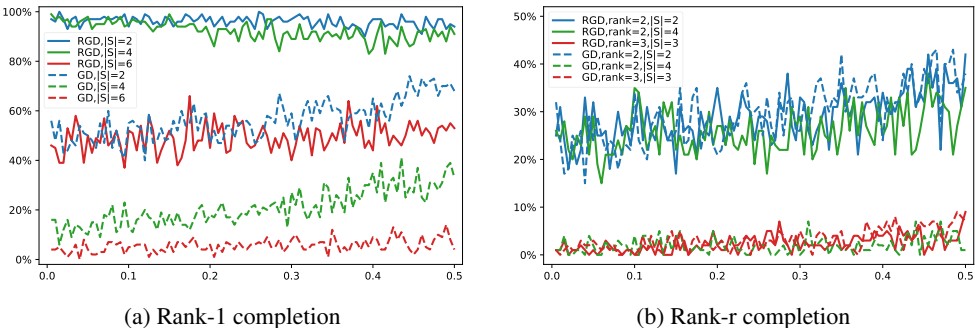

(a) Rank-1 completion          (b) Rank-r completion

Figure 6: Comparison of RGD with WN and GD on the challenging matrix completion problems (% successful convergence vs. perturbation size).

## B.2 EXTENSION TO NON-BENIGN LOSS LANDSCAPE

We further evaluate WN on the challenging matrix completion tasks proposed in (Yalçın et al., 2022), where the loss is constructed to have exponentially many spurious local minima, leading to the failure of most gradient-based methods. Using Burer-Monteiro factorization, the matrix completion objective can be written as

$$\min_{\mathbf{Y}\in\mathbb{R}^{m\times r}} \frac{1}{4}\|(\mathbf{Y}\mathbf{Y}^\top - \mathbf{M}^*_\varepsilon)_\Omega\|^2_{\mathsf{F}}, \tag{6}$$

where $\mathbf{M}^*_\varepsilon$ is a low-rank ground truth matrix and the measurement operator $(\cdot)_\Omega$ is constructed from specially designed combinatorial structures.

Applying WN, the problem becomes

$$\min_{\mathbf{X},\boldsymbol{\Theta}} \frac{1}{4}\|(\mathbf{X}\boldsymbol{\Theta}\mathbf{X}^\top - \mathbf{M}^*_\varepsilon)_\Omega\|^2_F, \quad \text{s.t.} \quad \mathbf{X}\in\mathsf{St}(m,r), \boldsymbol{\Theta}\in\mathbb{S}^r. \tag{7}$$

We use an update rule similar to Algorithm (1), with the operator $\mathcal{M}(\cdot)$ replaced by $(\cdot)_\Omega$. Following the experimental setup in (Yalçın et al., 2022) (see details in Appendix G.4), we evaluate the success rate across a range of ranks $r = 1, 2, 3$, maximum independent set sizes $|S| = 2, 4, 6$ and perturbation levels $\varepsilon \in [0.05, 0.5]$. Our findings are summarized as follows:

- Rank $r = 1$: Our algorithm performs particularly well. For $|S| = 2$ and $|S| = 4$, the success rates are over $90\%$ under almost all the perturbation levels, substantially higher than that of GD. Even for the more difficult case $|S| = 6$, our method still achieves a successful rate around $40\%$, again significantly outperforming GD.
- Rank $r = 2, 3$: In these regimes, both our algorithm and GD exhibit similarly low success rates, consistent with the intrinsic difficulty of the problem reported in (Yalçın et al., 2022).

These experiments on this challenging setting show that our approach has clear advantages over GD. The results indicate that WN remains effective even when the objective involves specially designed combinatorial structures or exhibits highly nontrivial optimization landscapes. This further highlights the potential of WN as a broadly applicable framework for low-rank optimization problems.

## C ALGORITHM 1 DERIVATION

We consider the overparameterized setting $r > r_A$ and apply a joint update on both $\mathbf{X}_t$ and $\boldsymbol{\Theta}_t$ in an alternating manner. Let $\mathcal{M}^* : \mathbb{R}^n \mapsto \mathbb{S}^m$ denote the adjoint of $\mathcal{M}$ with explicit form $\mathcal{M}^*(\mathbf{y}) = \sum_{i=1}^n y_i \mathbf{M}_i$. The Stiefel manifold $\mathsf{St}(m, r)$ is embedded in the Euclidean space, then we first compute the Euclidean gradient of $\mathbf{X}_t$ as

$$\tilde{\mathbf{G}}_t = \left[\mathcal{M}^*\mathcal{M}(\mathbf{X}_t\boldsymbol{\Theta}_t\mathbf{X}_t^\top - \mathbf{A})\right]\mathbf{X}_t\boldsymbol{\Theta}_t \tag{8}$$
$$= (\mathbf{X}_t\boldsymbol{\Theta}_t\mathbf{X}_t^\top - \mathbf{A})\mathbf{X}_t\boldsymbol{\Theta}_t + \left[(\mathcal{M}^*\mathcal{M} - \mathcal{I})(\mathbf{X}_t\boldsymbol{\Theta}_t\mathbf{X}_t^\top - \mathbf{A})\right]\mathbf{X}_t\boldsymbol{\Theta}_t.$$

Projecting it onto the tangent space of $\mathsf{St}(m, r)$ yields the Riemannian gradient

$$\mathbf{G}_t := (\mathbf{I}_m - \mathbf{X}_t\mathbf{X}_t^\top)\tilde{\mathbf{G}}_t + \frac{\mathbf{X}_t}{2}(\mathbf{X}_t^\top\tilde{\mathbf{G}}_t - \tilde{\mathbf{G}}_t^\top\mathbf{X}_t).$$

Using polar retraction, the update of $\mathbf{X}_t$ along the direction $\mathbf{G}_t$ with stepsize $\eta$ is given by

$$\mathbf{X}_{t+1} = (\mathbf{X}_t - \eta\mathbf{G}_t)(\mathbf{I}_r + \eta^2\mathbf{G}_t^\top\mathbf{G}_t)^{-1/2}.$$

For the magnitude variable $\boldsymbol{\Theta}_t$, the Euclidean gradient is

$$\mathbf{K}_t = \frac{1}{2}\mathbf{X}_{t+1}^\top\big[\mathcal{M}^*\mathcal{M}(\mathbf{X}_{t+1}\boldsymbol{\Theta}_t\mathbf{X}_{t+1}^\top - \mathbf{A})\big]\mathbf{X}_{t+1}. \tag{9}$$

Denoting the identity mapping by $\mathcal{I}$, the update of $\boldsymbol{\Theta}_t$ with stepsize $\mu$ becomes

$$\boldsymbol{\Theta}_{t+1} = \boldsymbol{\Theta}_t - \frac{\mu}{2}\mathbf{X}_{t+1}^\top\big[\mathcal{M}^*\mathcal{M}(\mathbf{X}_{t+1}\boldsymbol{\Theta}_t\mathbf{X}_{t+1}^\top - \mathbf{A})\big]\mathbf{X}_{t+1} \tag{10}$$
$$= \mathbf{X}_{t+1}^\top\mathbf{A}\mathbf{X}_{t+1} - \mathbf{X}_{t+1}^\top\big[(\mathcal{M}^*\mathcal{M} - \frac{\mu}{2}\mathcal{I})(\mathbf{X}_{t+1}\boldsymbol{\Theta}_t\mathbf{X}_{t+1}^\top - \mathbf{A})\big]\mathbf{X}_{t+1}.$$

Note that we do not impose diagonal or nonnegative constraints on $\boldsymbol{\Theta}$ during the updates. In fact, forcing $\boldsymbol{\Theta}$ to be diagonal and nonnegative often worsen the loss landscape and may lead to non-convergence Levin et al. (2025). We illustrate this phenomenon with a simple experiment in Figure 7, where we constrain $\boldsymbol{\Theta}$ to be diagonal and nonnegative via SVD followed by hard-thresholding; see Appendix G.5 for details of the experimental setup. As shown by the curve labeled "Diagonalized", such constraints indeed hinder the algorithm from converging to the ground truth.

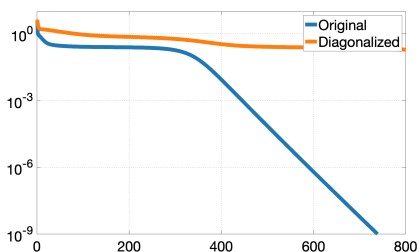

Figure 7: Lack of convergence with diagonal $\boldsymbol{\Theta}$ (squared reconstruction error vs. iteration).

# D  PROOF STRATEGIES AND SUPPORTING LEMMAS

## D.1  PROOF STRATEGIES

To establish convergence of Theorem 3.2, we analyze the evolution of the principle angles between $\mathsf{span}(\mathbf{U})$ and $\mathsf{span}(\mathbf{X}_t)$. Specifically, we track the quantity $\mathsf{Tr}(\mathbf{I}_{r_A} - \boldsymbol{\Phi}_t\boldsymbol{\Phi}_t^\top)$. This term reflects the subspace alignment error between $\mathsf{span}(\mathbf{U})$ and $\mathsf{span}(\mathbf{X}_t)$. For notational convenience, we set $\mu = 2$, which is consistent with our choice in Theorem 3.2.

Our proof is structured into two phases:

- Phase I (Saddle phase): When the alignment error is large, i.e., $\mathsf{Tr}(\mathbf{I}_{r_A} - \boldsymbol{\Phi}_t\boldsymbol{\Phi}_t^\top) \geq 0.5$[5], we rely on the fact that $\sigma_{r_A}^2(\boldsymbol{\Phi}_t)$ remains bounded away from zero. This property guarantees that the alignment error decreases by at least a constant amount at each iteration.

- Phase II (Linearly convergent phase): Once $\mathsf{Tr}(\mathbf{I}_{r_A} - \boldsymbol{\Phi}_t\boldsymbol{\Phi}_t^\top) < 0.5$, we enter a contraction regime. In this regime, we establish that the reconstruction error and the alignment error decrease jointly, governed by a coupled inequality system.

Throughout both phases, two error terms caused by the limited number of measurements must be carefully controlled. Formally, we introduce the following definitions:

$$\boldsymbol{\Delta}_t := (\mathcal{M}^*\mathcal{M} - \mathcal{I})(\mathbf{X}_{t+1}\boldsymbol{\Theta}_t\mathbf{X}_{t+1}^\top - \mathbf{A}),$$
$$\boldsymbol{\Xi}_t := (\mathcal{M}^*\mathcal{M} - \mathcal{I})(\mathbf{X}_t\boldsymbol{\Theta}_t\mathbf{X}_t^\top - \mathbf{A}).$$

---

[5]0.5 is chosen for simplicity, any constant $c \in (0, 1)$ is valid; see proof E.8 for a detailed analysis.

Incorporating these two error terms, we can rewrite $\tilde{\mathbf{G}}_t$ and $\boldsymbol{\Theta}_{t+1}$ as follows:

$$\tilde{\mathbf{G}}_t = (\mathbf{X}_t\boldsymbol{\Theta}_t\mathbf{X}_t^\top - \mathbf{A})\mathbf{X}_t\boldsymbol{\Theta}_t + \boldsymbol{\Xi}_t\mathbf{X}_t\boldsymbol{\Theta}_t,$$

$$\boldsymbol{\Theta}_{t+1} = \mathbf{X}_{t+1}^\top\mathbf{A}\mathbf{X}_{t+1} - \mathbf{X}_{t+1}^\top\boldsymbol{\Delta}_t\mathbf{X}_{t+1}.$$

These two terms will be used repeatedly throughout the proofs in the following sections.

## D.2 Supporting Lemmas

Since Theorem 3.2 considers random initialization, it is conditioned on the following high-probability event $F$, which gives a lower bound on the smallest singular value of $\boldsymbol{\Phi}_0 = \mathbf{U}^\top\mathbf{X}_0$:

$$F = \{\sigma_{r_A}^2(\mathbf{U}^\top\mathbf{X}_0) \geq \frac{(r - r_A)^2}{c_1 mr}\},$$

where $c_1 > \max\{1, 36C_1^2\}$ is a universal constant, with $C_1$ given in Lemma F.7.

**Lemma D.1** *With respect to the randomness in $\mathbf{X}_0$, event $F$ occurs with probability at least*

$$1 - \exp(-m/2) - C_3^{r-r_A+1} - \exp(-C_2 r),$$

*where $C_2 > 0$ and $C_3 = \frac{6C_1}{\sqrt{c_1}} \in (0,1)$ are universal constants.*

This lemma ensures that the smallest singular value of the initial alignment between $\mathbf{U}$ and $\mathbf{X}_0$ is bounded away from zero with high probability, which is critical to initialize Phase I. g

**Lemma D.2** *Suppose that at iteration $t$, the alignment error satisfies that*

$$\mathsf{Tr}(\mathbf{I}_{r_A} - \boldsymbol{\Phi}_t\boldsymbol{\Phi}_t^\top) \leq \rho,$$

*then the reconstruction error at iteration $t$ satisfies that*

$$\|\mathbf{X}_t\boldsymbol{\Theta}_t\mathbf{X}_t^\top - \mathbf{A}\|_\mathsf{F} \leq 2\sqrt{\rho} + \|\boldsymbol{\Delta}_{t-1}\|_\mathsf{F}.$$

The lemma above connects the reconstruction error $\|\mathbf{X}_t\boldsymbol{\Theta}_t\mathbf{X}_t^\top - \mathbf{A}\|_\mathsf{F}$ with the alignment error $\mathsf{Tr}(\mathbf{I}_{r_A} - \boldsymbol{\Phi}_t\boldsymbol{\Phi}_t^\top)$ and the measurement error $\|\boldsymbol{\Delta}_{t-1}\|_\mathsf{F}$. It means that the reconstruction error is small once $\mathbf{X}_t$ and $\mathbf{U}$ are sufficiently aligned and the measurement error is small.

**Lemma D.3** *Assuming $\eta < \frac{1}{300\kappa^2 r_A}$, $\mathcal{M}(\cdot)$ is $(r + r_A + 1, \delta)$-RIP with $\delta = \frac{\xi}{\sqrt{mr}}$, $\xi \in [0,1)$, and $\|\boldsymbol{\Theta}_t\| \leq 2$. Then, the measurement errors satisfy that*

$$\|\boldsymbol{\Delta}_t\|_\mathsf{F} \leq \xi\|\mathbf{X}_t\boldsymbol{\Theta}_t\mathbf{X}_t^\top - \mathbf{A}\|_\mathsf{F},$$

$$\|\boldsymbol{\Xi}_t\|_\mathsf{F} \leq \xi\|\mathbf{X}_t\boldsymbol{\Theta}_t\mathbf{X}_t^\top - \mathbf{A}\|_\mathsf{F}.$$

This provides upper bounds on the norm of the measurement error terms $\boldsymbol{\Delta}_t, \boldsymbol{\Xi}_t$ by the reconstruction error $\|\mathbf{X}_t\boldsymbol{\Theta}_t\mathbf{X}_t^\top - \mathbf{A}\|_\mathsf{F}$, which is guaranteed by the RIP property of $\mathcal{M}(\cdot)$.

**Lemma D.4** *Let $\chi_t := (\|\boldsymbol{\Delta}_{t-1}\| + \|\boldsymbol{\Xi}_t\|)^2 + \sqrt{\mathsf{Tr}(\mathbf{I}_{r_A} - \boldsymbol{\Phi}_t\boldsymbol{\Phi}_t^\top)}(\|\boldsymbol{\Delta}_{t-1}\| + \|\boldsymbol{\Xi}_t\|)$,*

$$\beta_t := \sigma_1(\mathbf{I}_{r_A} - \boldsymbol{\Phi}_t\boldsymbol{\Phi}_t^\top),$$

$$\mathbf{H}_t := (\mathbf{I}_m - \mathbf{X}_t\mathbf{X}_t^\top)(\mathbf{A}\mathbf{X}_t\mathbf{X}_t^\top\boldsymbol{\Delta}_{t-1}\mathbf{X}_t + \boldsymbol{\Xi}_t\mathbf{X}_t\boldsymbol{\Theta}_t)$$

$$+ \frac{1}{2}(\mathbf{X}_t\mathbf{X}_t^\top\boldsymbol{\Xi}_t\mathbf{X}_t\boldsymbol{\Theta}_t - \mathbf{X}_t\boldsymbol{\Theta}_t\mathbf{X}_t^\top\boldsymbol{\Xi}_t\mathbf{X}_t)$$

$$+ \frac{1}{2}(\mathbf{X}_t\mathbf{X}_t^\top\mathbf{A}\mathbf{X}_t\mathbf{X}_t^\top\boldsymbol{\Delta}_{t-1}\mathbf{X}_t - \mathbf{X}_t\mathbf{X}_t^\top\boldsymbol{\Delta}_{t-1}\mathbf{X}_t\mathbf{X}_t^\top\mathbf{A}\mathbf{X}_t).$$

*Assuming $\|\boldsymbol{\Delta}_{t-1}\|_\mathsf{F}, \|\boldsymbol{\Xi}_t\|_\mathsf{F} \leq 1$, $\eta \leq \frac{1}{10r_A}$, and $\|\boldsymbol{\Theta}_t\| \leq 2$, then the following inequality holds:*

$$\mathsf{Tr}(\mathbf{I}_{r_A} - \boldsymbol{\Phi}_{t+1}\boldsymbol{\Phi}_{t+1}^\top) - \mathsf{Tr}(\mathbf{I}_{r_A} - \boldsymbol{\Phi}_t\boldsymbol{\Phi}_t^\top) \tag{11}$$

$$\leq \eta^2(\beta_t + 16\chi_t)\mathsf{Tr}(\boldsymbol{\Phi}_t\boldsymbol{\Phi}_t^\top) - \frac{2\eta(1 - \eta^2\beta_t - 16\eta^2\chi_t)\sigma_{r_A}^2(\boldsymbol{\Phi}_t)}{\kappa^2}\mathsf{Tr}\big((\mathbf{I}_{r_A} - \boldsymbol{\Phi}_t\boldsymbol{\Phi}_t^\top)\boldsymbol{\Phi}_t\boldsymbol{\Phi}_t^\top\big)$$

$$+ 2\eta\sqrt{\mathsf{Tr}(\mathbf{I}_{r_A} - \boldsymbol{\Phi}_t\boldsymbol{\Phi}_t^\top)}(\|\boldsymbol{\Delta}_{t-1}\|_\mathsf{F} + 2\|\boldsymbol{\Xi}_t\|_\mathsf{F})$$

$$+ 2\eta^2\sqrt{\mathsf{Tr}(\mathbf{I}_{r_A} - \boldsymbol{\Phi}_t\boldsymbol{\Phi}_t^\top)}\|\mathbf{H}_t\|_\mathsf{F}.$$

This lemma quantifies how the alignment error $\mathrm{Tr}(\mathbf{I}_{r_A} - \mathbf{\Phi}_t\mathbf{\Phi}_t^\top)$ evolves between iterations. This is the key lemma that drives the reduction of the alignment error.

**Lemma D.5** *Assuming $\mathcal{M}(\cdot)$ is $(r + r_A + 1, \delta)$-RIP with $\delta \leq \frac{1}{3\sqrt{m}}$. If $\|\mathbf{\Theta}_t\| \leq 2$, then it is guaranteed that*

$$\|\mathbf{\Theta}_{t+1}\| \leq 2.$$

As shown in Lemmas D.3 and Lemma D.4, the analyses require that $\|\mathbf{\Theta}_t\|$ is upper bounded by 2. This condition has already been guaranteed at initialization. Moreover, based on the update rule of $\mathbf{\Theta}_t$ given in (10), we observe that $\|\mathbf{\Theta}_t\|$ remains close to $\|\mathbf{X}_t^\top \mathbf{A}\mathbf{X}_t\|$ in each iteration.

**Lemma D.6** *Assuming $\eta \leq 1, \mathcal{M}(\cdot)$ is $(r + r_A + 1, \delta)$-RIP, and $\|\mathbf{\Theta}_{t-1}\|, \|\mathbf{\Theta}_t\| \leq 2$, we have that*

$$\mathbf{\Psi}_{t+1}\mathbf{\Psi}_{t+1}^\top \preceq \left(1 + 6\eta(\sqrt{r_A} + 2\sqrt{r})\delta\right)^2 \mathbf{\Psi}_t\mathbf{\Psi}_t^\top + \left(4\eta(\sqrt{r_A} + 2\sqrt{r})\delta + 28\eta^2(\sqrt{r_A} + 2\sqrt{r})^2\delta^2\right)\mathbf{I}_{m-r_A}.$$

*Moreover, it is also guaranteed that*

$$\mathbf{\Psi}_1\mathbf{\Psi}_1^\top \preceq \left(1 + 2\eta + 2\eta(\sqrt{r_A} + 2\sqrt{r})\delta\right)^2 \mathbf{\Psi}_0\mathbf{\Psi}_0^\top + \left(12\eta(\sqrt{r_A} + 2\sqrt{r})\delta + 8\eta^2(\sqrt{r_A} + 2\sqrt{r})^2\delta^2\right)\mathbf{I}_{m-r_A}.$$

This lemma establishes an upper bound on the growth of $\mathbf{\Psi}_t\mathbf{\Psi}_t^\top$. Together with Lemma F.2, we can ensure that $\sigma_{r_A}^2(\mathbf{\Phi}_t)$ remains adequately large throughout Phase I.

**Lemma D.7** *Assuming $\eta \leq \frac{1}{500 r_A}, \mathcal{M}(\cdot)$ is $(r + r_A + 1, \delta)$-RIP with $\delta \leq \frac{1}{\sqrt{m}}$, and $\|\mathbf{\Theta}_t\| \leq 2$. Then for any $t \geq 0$, the alignment error satisfies that*

$$\mathrm{Tr}(\mathbf{I}_{r_A} - \mathbf{\Phi}_{t+1}\mathbf{\Phi}_{t+1}^\top) \leq \mathrm{Tr}(\mathbf{I}_{r_A} - \mathbf{\Phi}_t\mathbf{\Phi}_t^\top) + 0.1.$$

This guarantees that the alignment error $\mathrm{Tr}(\mathbf{I}_{r_A} - \mathbf{\Phi}_t\mathbf{\Phi}_t^\top)$ does not increase too much in one step when we choose suitable stepsize $\eta$, which is crucial for bridging Phase I and Phase II.

## E  PROOFS

### E.1  PROOF OF LEMMA D.1

*Proof.* Since the initialization $\mathbf{X}_0$ satisfies the conditions stated in Lemma F.8, we can apply the lemma directly. In particular, substituting $\tau = \frac{6}{\sqrt{c_1}}$ yields the desired result. $\qquad\square$

### E.2  PROOF OF LEMMA D.2

*Proof.* Directly substituting the expression of $\mathbf{\Theta}_t$ into the Frobenius norm term, we have that

$$
\begin{aligned}
\|\mathbf{X}_t\mathbf{\Theta}_t\mathbf{X}_t^\top - \mathbf{A}\|_\mathsf{F} &= \|\mathbf{X}_t\mathbf{X}_t^\top \mathbf{A}\mathbf{X}_t\mathbf{X}_t^\top - \mathbf{A} - \mathbf{X}_t\mathbf{X}_t^\top \mathbf{\Delta}_{t-1}\mathbf{X}_t\mathbf{X}_t^\top\|_\mathsf{F} \\
&\leq \|\mathbf{X}_t\mathbf{X}_t^\top \mathbf{A}\mathbf{X}_t\mathbf{X}_t^\top - \mathbf{A}\|_\mathsf{F} + \|\mathbf{X}_t\mathbf{X}_t^\top \mathbf{\Delta}_{t-1}\mathbf{X}_t\mathbf{X}_t^\top\|_\mathsf{F} \\
&\leq \|\mathbf{X}_t\mathbf{X}_t^\top \mathbf{A}\mathbf{X}_t\mathbf{X}_t^\top - \mathbf{A}\mathbf{X}_t\mathbf{X}_t^\top\|_\mathsf{F} + \|\mathbf{A}\mathbf{X}_t\mathbf{X}_t^\top - \mathbf{A}\|_\mathsf{F} + \|\mathbf{X}_t\mathbf{X}_t^\top \mathbf{\Delta}_{t-1}\mathbf{X}_t\mathbf{X}_t^\top\|_\mathsf{F} \\
&\overset{(a)}{\leq} 2\|\mathbf{\Sigma}\|\|(\mathbf{I}_m - \mathbf{X}_t\mathbf{X}_t^\top)\mathbf{U}\|_\mathsf{F} + \|\mathbf{\Delta}_{t-1}\|_\mathsf{F} \\
&= 2\|\mathbf{\Sigma}\|\sqrt{\mathrm{Tr}(\mathbf{I}_{r_A} - \mathbf{\Phi}_t\mathbf{\Phi}_t^\top)} + \|\mathbf{\Delta}_{t-1}\|_\mathsf{F} \\
&\leq 2\|\mathbf{\Sigma}\|\sqrt{\rho} + \|\mathbf{\Delta}_{t-1}\|_\mathsf{F} \\
&= 2\sqrt{\rho} + \|\mathbf{\Delta}_{t-1}\|_\mathsf{F},
\end{aligned}
$$

where $(a)$ is by the inequality $\|\mathbf{A}\mathbf{B}\|_\mathsf{F} \leq \|\mathbf{A}\|\|\mathbf{B}\|_\mathsf{F}$ that is valid for any conformable matrices. $\quad\square$

### E.3 PROOF OF LEMMA D.3

*Proof.* We first prove that $\|\mathbf{G}_t\|_\mathsf{F} \le 2\|\tilde{\mathbf{G}}_t\|_\mathsf{F}$. Indeed,

$$
\begin{aligned}
\|\mathbf{G}_t\|_\mathsf{F} &= \|(\mathbf{I}_m - \mathbf{X}_t\mathbf{X}_t^\top)\tilde{\mathbf{G}}_t + \frac{\mathbf{X}_t}{2}(\mathbf{X}_t^\top\tilde{\mathbf{G}}_t - \tilde{\mathbf{G}}_t^\top\mathbf{X}_t)\|_\mathsf{F} \\
&\le \|\mathbf{I}_m - \mathbf{X}_t\mathbf{X}_t^\top\|\|\tilde{\mathbf{G}}_t\|_\mathsf{F} + \|\mathbf{X}_t\mathbf{X}_t^\top\|\|\tilde{\mathbf{G}}_t\|_\mathsf{F} \\
&\le 2\|\tilde{\mathbf{G}}_t\|_\mathsf{F}.
\end{aligned}
$$

We now proceed to estimate the update distance $\|\mathbf{X}_{t+1} - \mathbf{X}_t\|_\mathsf{F}$.

$$
\begin{aligned}
\|\mathbf{X}_{t+1} - \mathbf{X}_t\|_\mathsf{F} &= \|(\mathbf{X}_t - \eta\mathbf{G}_t)(\mathbf{I}_r + \eta^2\mathbf{G}_t^\top\mathbf{G}_t)^{-1/2} - \mathbf{X}_t\|_\mathsf{F} \\
&\le \|\mathbf{X}_t((\mathbf{I}_r + \eta^2\mathbf{G}_t^\top\mathbf{G}_t)^{-1/2} - \mathbf{I}_r)\|_\mathsf{F} + \|\eta\mathbf{G}_t(\mathbf{I}_r + \eta^2\mathbf{G}_t^\top\mathbf{G}_t)^{-1/2}\|_\mathsf{F} \\
&\le \|\mathbf{X}_t\|\|(\mathbf{I}_r + \eta^2\mathbf{G}_t^\top\mathbf{G}_t)^{-1/2} - \mathbf{I}_r\|_\mathsf{F} + \eta\|(\mathbf{I}_r + \eta^2\mathbf{G}_t^\top\mathbf{G}_t)^{-1/2}\|\|\mathbf{G}_t\|_\mathsf{F} \\
&\le \sqrt{r}\|(\mathbf{I}_r + \eta^2\mathbf{G}_t^\top\mathbf{G}_t)^{-1/2} - \mathbf{I}_r\| + \eta\|\mathbf{G}_t\|_\mathsf{F} \\
&\le \sqrt{r}\|(\mathbf{I}_r + \eta^2\mathbf{G}_t^\top\mathbf{G}_t)^{-1/2} - \mathbf{I}_r\| + 2\eta\|\tilde{\mathbf{G}}_t\|_\mathsf{F} \\
&\le \sqrt{r}(1 - (1 + \eta^2\sigma_1(\mathbf{G}_t^\top\mathbf{G}_t))^{-1/2}) + 2\eta\|\tilde{\mathbf{G}}_t\|_\mathsf{F} \\
&\overset{(a)}{\le} \sqrt{r}(1 - \frac{1}{1 + (\eta^2\|\mathbf{G}_t\|_\mathsf{F}^2)^{1/2}}) + 2\eta\|\tilde{\mathbf{G}}_t\|_\mathsf{F} \\
&\le \sqrt{r}\eta\|\mathbf{G}_t\|_\mathsf{F} + 2\eta\|\tilde{\mathbf{G}}_t\|_\mathsf{F},
\end{aligned}
$$

where $(a)$ is by $\sqrt{1+x} \le 1 + \sqrt{x}$ for any $x \ge 0$. Since $\|\mathbf{G}_t\|_\mathsf{F} \le 2\|\tilde{\mathbf{G}}_t\|_\mathsf{F}$, we arrive at

$$
\begin{aligned}
\|\mathbf{X}_{t+1} - \mathbf{X}_t\|_\mathsf{F} &\le 2\eta(\sqrt{r} + 1)\|\tilde{\mathbf{G}}_t\|_\mathsf{F} \\
&= 2\eta(\sqrt{r} + 1)\|(\mathbf{X}_t\mathbf{\Theta}_t\mathbf{X}_t^\top - \mathbf{A})\mathbf{X}_t\mathbf{\Theta}_t + \mathbf{\Xi}_t\mathbf{X}_t\mathbf{\Theta}_t\|_\mathsf{F} \\
&\overset{(b)}{\le} 2\eta(\sqrt{r} + 1)\|\mathbf{\Theta}_t\|\|\mathbf{X}_t\|\|(\mathbf{X}_t\mathbf{\Theta}_t\mathbf{X}_t^\top - \mathbf{A}) + \mathbf{\Xi}_t\|_\mathsf{F} \\
&\overset{(c)}{\le} 4\eta(\sqrt{r} + 1)\big(\|\mathbf{\Xi}_t\|_\mathsf{F} + \|\mathbf{X}_t\mathbf{\Theta}_t\mathbf{X}_t^\top - \mathbf{A}\|_\mathsf{F}\big) \\
&\le 4\eta(\sqrt{r} + 1)\big(\sqrt{m}\|(\mathcal{M}^*\mathcal{M} - \mathcal{I})(\mathbf{X}_t\mathbf{\Theta}_t\mathbf{X}_t^\top - \mathbf{A})\| + \|\mathbf{X}_t\mathbf{\Theta}_t\mathbf{X}_t^\top - \mathbf{A}\|_\mathsf{F}\big) \\
&\overset{(d)}{\le} 4\eta(\sqrt{r} + 1)(\sqrt{m}\delta + 1)\|\mathbf{X}_t\mathbf{\Theta}_t\mathbf{X}_t^\top - \mathbf{A}\|_\mathsf{F},
\end{aligned}
$$

where $(b)$ is from $\|\mathbf{AB}\|_\mathsf{F} \le \|\mathbf{A}\|\|\mathbf{B}\|_\mathsf{F}$; $(c)$ is due to $\|\mathbf{\Theta}_t\| \le 2$, $\|\mathbf{X}_t\| \le 1$; and $(d)$ follows from Lemma F.11 and $\text{rank}(\mathbf{X}_t\mathbf{\Theta}_t\mathbf{X}_t^\top - \mathbf{A}) \le \text{rank}(\mathbf{X}_t\mathbf{\Theta}_t\mathbf{X}_t^\top) + \text{rank}(\mathbf{A}) \le r + r_A$.

Finally, we turn to estimating $\|\mathbf{\Delta}_t\|_{\mathsf{F}}$ and $\|\mathbf{\Xi}_t\|_{\mathsf{F}}$.

$$
\begin{aligned}
\|\mathbf{\Delta}_t\|_{\mathsf{F}} &= \|(\mathcal{M}^*\mathcal{M} - \mathcal{I})(\mathbf{X}_{t+1}\mathbf{\Theta}_t\mathbf{X}_{t+1}^\top - \mathbf{A})\|_{\mathsf{F}} \\
&\leq \sqrt{m}\|(\mathcal{M}^*\mathcal{M} - \mathcal{I})(\mathbf{X}_{t+1}\mathbf{\Theta}_t\mathbf{X}_{t+1}^\top - \mathbf{A})\| \\
&\overset{(e)}{\leq} \sqrt{m}\delta\|\mathbf{X}_{t+1}\mathbf{\Theta}_t\mathbf{X}_{t+1}^\top - \mathbf{A}\|_{\mathsf{F}} \\
&\leq \sqrt{m}\delta(\|\mathbf{X}_t\mathbf{\Theta}_t\mathbf{X}_t^\top - \mathbf{A}\|_{\mathsf{F}} + \|\mathbf{X}_{t+1}\mathbf{\Theta}_t(\mathbf{X}_{t+1}^\top - \mathbf{X}_t^\top)\|_{\mathsf{F}} + \|(\mathbf{X}_{t+1} - \mathbf{X}_t)\mathbf{\Theta}_t\mathbf{X}_t^\top\|_{\mathsf{F}}) \\
&\overset{(f)}{\leq} \sqrt{m}\delta(\|\mathbf{X}_t\mathbf{\Theta}_t\mathbf{X}_t^\top - \mathbf{A}\|_{\mathsf{F}} + 4\|\mathbf{X}_{t+1} - \mathbf{X}_t\|_{\mathsf{F}}) \\
&\leq \sqrt{m}\delta\big(1 + 16\eta(\sqrt{r}+1)(\sqrt{m}\delta+1)\big)\|\mathbf{X}_t\mathbf{\Theta}_t\mathbf{X}_t^\top - \mathbf{A}\|_{\mathsf{F}} \\
&\overset{(g)}{\leq} \frac{\xi\sqrt{m}}{\sqrt{mr}}(1 + \frac{64}{300}\sqrt{r})\|\mathbf{X}_t\mathbf{\Theta}_t\mathbf{X}_t^\top - \mathbf{A}\|_{\mathsf{F}} \\
&\leq \xi\|\mathbf{X}_t\mathbf{\Theta}_t\mathbf{X}_t^\top - \mathbf{A}\|_{\mathsf{F}}, \\
\|\mathbf{\Xi}_t\|_{\mathsf{F}} &= \|(\mathcal{M}^*\mathcal{M} - \mathcal{I})(\mathbf{X}_t\mathbf{\Theta}_t\mathbf{X}_t^\top - \mathbf{A})\|_{\mathsf{F}} \\
&\leq \sqrt{m}\|(\mathcal{M}^*\mathcal{M} - \mathcal{I})(\mathbf{X}_t\mathbf{\Theta}_t\mathbf{X}_t^\top - \mathbf{A})\| \\
&\overset{(h)}{\leq} \sqrt{m}\delta\|\mathbf{X}_t\mathbf{\Theta}_t\mathbf{X}_t^\top - \mathbf{A}\|_{\mathsf{F}} \\
&\overset{(g)}{\leq} \xi\|\mathbf{X}_t\mathbf{\Theta}_t\mathbf{X}_t^\top - \mathbf{A}\|_{\mathsf{F}},
\end{aligned}
$$

where $(e)$ is by Lemma F.11 and $\mathsf{rank}(\mathbf{X}_{t+1}\mathbf{\Theta}_t\mathbf{X}_{t+1}^\top - \mathbf{A}) \leq \mathsf{rank}(\mathbf{X}_{t+1}\mathbf{\Theta}_t\mathbf{X}_{t+1}^\top) + \mathsf{rank}(\mathbf{A}) \leq r + r_A$; $(f)$ is from $\|\mathbf{X}_{t+1}\| \leq 1$ and $\|\mathbf{\Theta}_t\| \leq 2$; $(g)$ is due to $\eta \leq \frac{1}{300\kappa^2 r_A}$ and $\delta \leq \frac{\xi}{\sqrt{mr}}$; and $(h)$ follows from Lemma F.11 and $\mathsf{rank}(\mathbf{X}_t\mathbf{\Theta}_t\mathbf{X}_t^\top - \mathbf{A}) \leq \mathsf{rank}(\mathbf{X}_t\mathbf{\Theta}_t\mathbf{X}_t^\top) + \mathsf{rank}(\mathbf{A}) \leq r + r_A$. □

### E.4 PROOF OF LEMMA D.4

*Proof.* Noting that $\|\mathbf{X}_t\| \leq 1, \|\mathbf{A}\| \leq 1, \|\mathbf{\Theta}_t\| \leq 2, \|\mathbf{I}_m - \mathbf{X}_t\mathbf{X}_t^\top\| \leq 1$, we obtain

$$
\begin{aligned}
\|\mathbf{H}_t\|_{\mathsf{F}} &\leq \|(\mathbf{I}_m - \mathbf{X}_t\mathbf{X}_t^\top)\mathbf{A}\mathbf{X}_t\mathbf{X}_t^\top\mathbf{\Delta}_{t-1}\mathbf{X}_t\|_{\mathsf{F}} + \|(\mathbf{I}_m - \mathbf{X}_t\mathbf{X}_t^\top)\mathbf{\Xi}_t\mathbf{X}_t\mathbf{\Theta}_t\|_{\mathsf{F}} \\
&\quad + \frac{1}{2}(\|\mathbf{X}_t\mathbf{X}_t^\top\mathbf{\Xi}_t\mathbf{X}_t\mathbf{\Theta}_t\|_{\mathsf{F}} + \|\mathbf{X}_t\mathbf{\Theta}_t\mathbf{X}_t^\top\mathbf{\Xi}_t\mathbf{X}_t\|_{\mathsf{F}}) \\
&\quad + \frac{1}{2}(\|\mathbf{X}_t\mathbf{X}_t^\top\mathbf{A}\mathbf{X}_t\mathbf{X}_t^\top\mathbf{\Delta}_{t-1}\mathbf{X}_t\|_{\mathsf{F}} + \|\mathbf{X}_t\mathbf{X}_t^\top\mathbf{\Delta}_{t-1}\mathbf{X}_t\mathbf{X}_t^\top\mathbf{A}\mathbf{X}_t\|_{\mathsf{F}}) \\
&\leq 2\|\mathbf{\Delta}_{t-1}\|_{\mathsf{F}} + 4\|\mathbf{\Xi}_t\|_{\mathsf{F}}. \quad (12)
\end{aligned}
$$

In the same way, it follows that $\|\mathbf{H}_t\| \leq 2\|\mathbf{\Delta}_{t-1}\| + 4\|\mathbf{\Xi}_t\|$.

From the update of $\mathbf{X}_t$, we have $\mathbf{X}_{t+1}\mathbf{X}_{t+1}^\top = (\mathbf{X}_t - \eta\mathbf{G}_t)(\mathbf{I}_r + \eta^2\mathbf{G}_t^\top\mathbf{G}_t)^{-1}(\mathbf{X}_t - \eta\mathbf{G}_t)^\top$. Premultiplying by $\mathbf{U}^\top$ and postmultiplying by $\mathbf{U}$, it follows that

$$
\begin{aligned}
&\mathbf{\Phi}_{t+1}\mathbf{\Phi}_{t+1}^\top \\
&= (\mathbf{\Phi}_t - \eta\mathbf{U}^\top\mathbf{G}_t)(\mathbf{I}_r + \eta^2\mathbf{G}_t^\top\mathbf{G}_t)^{-1}(\mathbf{\Phi}_t^\top - \eta\mathbf{G}_t^\top\mathbf{U}) \\
&\overset{(a)}{=} \Big([\mathbf{I}_{r_A} + \eta(\mathbf{I}_{r_A} - \mathbf{\Phi}_t\mathbf{\Phi}_t^\top)\mathbf{\Sigma}\mathbf{\Phi}_t\mathbf{\Phi}_t^\top\mathbf{\Sigma}]\mathbf{\Phi}_t - \eta\mathbf{U}^\top\mathbf{H}_t\Big)(\mathbf{I}_r + \eta^2\mathbf{G}_t^\top\mathbf{G}_t)^{-1} \\
&\qquad\qquad\qquad\qquad \Big([\mathbf{I}_{r_A} + \eta(\mathbf{I}_{r_A} - \mathbf{\Phi}_t\mathbf{\Phi}_t^\top)\mathbf{\Sigma}\mathbf{\Phi}_t\mathbf{\Phi}_t^\top\mathbf{\Sigma}]\mathbf{\Phi}_t - \eta\mathbf{U}^\top\mathbf{H}_t\Big)^\top \\
&\overset{(b)}{\succeq} \Big([\mathbf{I}_{r_A} + \eta(\mathbf{I}_{r_A} - \mathbf{\Phi}_t\mathbf{\Phi}_t^\top)\mathbf{\Sigma}\mathbf{\Phi}_t\mathbf{\Phi}_t^\top\mathbf{\Sigma}]\mathbf{\Phi}_t - \eta\mathbf{U}^\top\mathbf{H}_t\Big)(\mathbf{I}_r - \eta^2\mathbf{G}_t^\top\mathbf{G}_t) \\
&\qquad\qquad\qquad\qquad \Big([\mathbf{I}_{r_A} + \eta(\mathbf{I}_{r_A} - \mathbf{\Phi}_t\mathbf{\Phi}_t^\top)\mathbf{\Sigma}\mathbf{\Phi}_t\mathbf{\Phi}_t^\top\mathbf{\Sigma}]\mathbf{\Phi}_t - \eta\mathbf{U}^\top\mathbf{H}_t\Big)^\top,
\end{aligned}
$$

where $(a)$ is from directly expanding $\mathbf{G}_t$; and $(b)$ is by Lemma F.1.

We next derive an upper bound for $\mathbf{G}_t^\top \mathbf{G}_t$. Substituting the expression of $\mathbf{G}_t$, we obtain

$$
\begin{aligned}
\mathbf{G}_t^\top \mathbf{G}_t &= \mathbf{X}_t^\top \mathbf{A} \mathbf{X}_t \mathbf{X}_t^\top \mathbf{A} (\mathbf{I}_m - \mathbf{X}_t \mathbf{X}_t^\top) \mathbf{A} \mathbf{X}_t \mathbf{X}_t^\top \mathbf{A} \mathbf{X}_t + \mathbf{H}_t^\top \mathbf{H}_t \\
&\quad - \mathbf{X}_t^\top \mathbf{A} \mathbf{X}_t \mathbf{X}_t^\top \mathbf{A} (\mathbf{I}_m - \mathbf{X}_t \mathbf{X}_t^\top) \mathbf{H}_t \\
&\quad - \mathbf{H}_t^\top (\mathbf{I}_m - \mathbf{X}_t \mathbf{X}_t^\top) \mathbf{A} \mathbf{X}_t \mathbf{X}_t^\top \mathbf{A} \mathbf{X}_t \\
&\overset{(c)}{\preceq} \sigma_1 (\mathbf{I}_{r_A} - \mathbf{\Phi}_t \mathbf{\Phi}_t^\top) \mathbf{I}_r + (\|\mathbf{H}_t\|^2 + 2\|(\mathbf{I}_m - \mathbf{X}_t \mathbf{X}_t^\top)\mathbf{U}\|\|\mathbf{H}_t\|) \mathbf{I}_r \\
&\overset{(d)}{\preceq} \sigma_1 (\mathbf{I}_{r_A} - \mathbf{\Phi}_t \mathbf{\Phi}_t^\top) \mathbf{I}_r + 16 \chi_t \mathbf{I}_r,
\end{aligned}
$$

where $(c)$ follows from Lemma F.12, $\|\mathbf{X}_t\| \leq 1$ and $\|\mathbf{A}\| \leq 1$; and $(d)$ is due to $\|(\mathbf{I}_m - \mathbf{X}_t \mathbf{X}_t^\top)\mathbf{U}\| \leq \|(\mathbf{I}_m - \mathbf{X}_t \mathbf{X}_t^\top)\mathbf{U}\|_{\mathsf{F}} = \sqrt{\mathsf{Tr}(\mathbf{I}_{r_A} - \mathbf{\Phi}_t \mathbf{\Phi}_t^\top)}$, and $\|\mathbf{H}_t\| \leq 2\|\mathbf{\Delta}_{t-1}\| + 4\|\mathbf{\Xi}_t\| \leq 6$.

Combining the lower bound on $\mathbf{\Phi}_{t+1} \mathbf{\Phi}_{t+1}^\top$, the upper bound on $\mathbf{G}_t^\top \mathbf{G}_t$ derived above, and the inequality $1 - \eta^2 \beta_t - 16\eta^2 \chi_t \geq 1 - \frac{1}{100}(1 + 96) > 0$, we derive

$$
\begin{aligned}
&\frac{1}{1 - \eta^2 \beta_t - 16\eta^2 \chi_t} \mathbf{\Phi}_{t+1} \mathbf{\Phi}_{t+1}^\top \\
&\succeq \left( \left[ \mathbf{I}_{r_A} + \eta (\mathbf{I}_{r_A} - \mathbf{\Phi}_t \mathbf{\Phi}_t^\top) \mathbf{\Sigma} \mathbf{\Phi}_t \mathbf{\Phi}_t^\top \mathbf{\Sigma} \right] \mathbf{\Phi}_t - \eta \mathbf{U}^\top \mathbf{H}_t \right) \qquad (13) \\
&\qquad\qquad \left( \left[ \mathbf{I}_{r_A} + \eta (\mathbf{I}_{r_A} - \mathbf{\Phi}_t \mathbf{\Phi}_t^\top) \mathbf{\Sigma} \mathbf{\Phi}_t \mathbf{\Phi}_t^\top \mathbf{\Sigma} \right] \mathbf{\Phi}_t - \eta \mathbf{U}^\top \mathbf{H}_t \right)^\top.
\end{aligned}
$$

Let the compact SVD of $\mathbf{\Phi}_t$ be $\mathbf{Q}_t \mathbf{\Lambda}_t \mathbf{P}_t^\top$, where $\mathbf{Q}_t \in \mathbb{R}^{r_A \times r_A}$, $\mathbf{\Lambda}_t \in \mathbb{R}^{r_A \times r_A}$, and $\mathbf{P}_t \in \mathbb{R}^{r \times r_A}$. Denote $\mathbf{S}_t := \mathbf{Q}_t^\top \mathbf{\Sigma} \mathbf{Q}_t$. It is a positive definite matrix. This gives that

$$
\begin{aligned}
&\mathsf{Tr}\left( \left[ \mathbf{I}_{r_A} + \eta (\mathbf{I}_{r_A} - \mathbf{\Phi}_t \mathbf{\Phi}_t^\top) \mathbf{\Sigma} \mathbf{\Phi}_t \mathbf{\Phi}_t^\top \mathbf{\Sigma} \right] \mathbf{\Phi}_t \mathbf{\Phi}_t^\top \left[ \mathbf{I}_{r_A} + \eta (\mathbf{I}_{r_A} - \mathbf{\Phi}_t \mathbf{\Phi}_t^\top) \mathbf{\Sigma} \mathbf{\Phi}_t \mathbf{\Phi}_t^\top \mathbf{\Sigma} \right]^\top \right) \\
&= \mathsf{Tr}\left( \mathbf{Q}_t \left[ \mathbf{I}_{r_A} + \eta (\mathbf{I}_{r_A} - \mathbf{\Lambda}_t^2) \mathbf{S}_t \mathbf{\Lambda}_t^2 \mathbf{S}_t \right] \mathbf{\Lambda}_t^2 \left[ \mathbf{I}_{r_A} + \eta \mathbf{S}_t \mathbf{\Lambda}_t^2 \mathbf{S}_t (\mathbf{I}_{r_A} - \mathbf{\Lambda}_t^2) \right] \mathbf{Q}_t^\top \right) \\
&\overset{(e)}{\geq} \mathsf{Tr}\left( \mathbf{Q}_t \left[ \mathbf{\Lambda}_t^2 + \eta (\mathbf{I}_{r_A} - \mathbf{\Lambda}_t^2) \mathbf{S}_t \mathbf{\Lambda}_t^2 \mathbf{S}_t \mathbf{\Lambda}_t^2 + \eta \mathbf{\Lambda}_t^2 \mathbf{S}_t \mathbf{\Lambda}_t^2 \mathbf{S}_t (\mathbf{I}_{r_A} - \mathbf{\Lambda}_t^2) \right] \mathbf{Q}_t^\top \right) \\
&= \mathsf{Tr}(\mathbf{Q}_t \mathbf{\Lambda}_t^2 \mathbf{Q}_t^\top) + \eta \mathsf{Tr}\left( (\mathbf{I}_{r_A} - \mathbf{\Lambda}_t^2) \mathbf{S}_t \mathbf{\Lambda}_t^2 \mathbf{S}_t \mathbf{\Lambda}_t^2 + \mathbf{\Lambda}_t^2 \mathbf{S}_t \mathbf{\Lambda}_t^2 \mathbf{S}_t (\mathbf{I}_{r_A} - \mathbf{\Lambda}_t^2) \right) \\
&\overset{(f)}{\geq} \mathsf{Tr}(\mathbf{Q}_t \mathbf{\Lambda}_t^2 \mathbf{Q}_t^\top) + \frac{2\eta \sigma_{r_A}(\mathbf{\Lambda}_t^2)}{\kappa^2} \mathsf{Tr}\left( (\mathbf{I}_{r_A} - \mathbf{\Lambda}_t^2) \mathbf{\Lambda}_t^2 \right) \\
&= \mathsf{Tr}(\mathbf{\Phi}_t \mathbf{\Phi}_t^\top) + \frac{2\eta \sigma_{r_A}(\mathbf{\Lambda}_t^2)}{\kappa^2} \mathsf{Tr}\left( (\mathbf{I}_{r_A} - \mathbf{\Phi}_t \mathbf{\Phi}_t^\top) \mathbf{\Phi}_t \mathbf{\Phi}_t^\top \right),
\end{aligned}
$$

where $(e)$ follows from the fact that $\eta^2 \mathbf{Q}_t (\mathbf{I}_{r_A} - \mathbf{\Lambda}_t^2) \mathbf{S}_t \mathbf{\Lambda}_t^2 \mathbf{S}_t \mathbf{\Lambda}_t^2 \mathbf{S}_t \mathbf{\Lambda}_t^2 \mathbf{S}_t (\mathbf{I}_{r_A} - \mathbf{\Lambda}_t^2) \mathbf{Q}_t^\top$ is PSD; and $(f)$ is by Lemma F.3 and Lemma F.4. More precisely, we use $\sigma_{r_A}(\mathbf{S}_t \mathbf{\Lambda}_t^2 \mathbf{S}_t) \geq \sigma_{r_A}^2(\mathbf{S}_t) \sigma_{r_A}(\mathbf{\Lambda}_t^2) = \sigma_{r_A}(\mathbf{\Lambda}_t^2)/\kappa^2$.

Taking trace on both sides of (13), we arrive at

$$\frac{1}{1 - \eta^2 \beta_t - 16\eta^2 \chi_t} \mathsf{Tr}(\boldsymbol{\Phi}_{t+1} \boldsymbol{\Phi}_{t+1}^\top) \tag{14}$$

$$\geq \mathsf{Tr}(\boldsymbol{\Phi}_t \boldsymbol{\Phi}_t^\top) + \frac{2\eta \sigma_{r_A}(\boldsymbol{\Lambda}_t^2)}{\kappa^2} \mathsf{Tr}\big((\mathbf{I}_{r_A} - \boldsymbol{\Phi}_t \boldsymbol{\Phi}_t^\top) \boldsymbol{\Phi}_t \boldsymbol{\Phi}_t^\top\big)$$

$$\quad - 2\eta \mathsf{Tr}\Big(\big[\mathbf{I}_{r_A} + \eta(\mathbf{I}_{r_A} - \boldsymbol{\Phi}_t \boldsymbol{\Phi}_t^\top)\boldsymbol{\Sigma}\boldsymbol{\Phi}_t \boldsymbol{\Phi}_t^\top \boldsymbol{\Sigma}\big]\boldsymbol{\Phi}_t \mathbf{H}_t^\top \mathbf{U}\Big)$$

$$\quad + \eta^2 \mathsf{Tr}(\mathbf{U}^\top \mathbf{H}_t \mathbf{H}_t^\top \mathbf{U})$$

$$\geq \mathsf{Tr}(\boldsymbol{\Phi}_t \boldsymbol{\Phi}_t^\top) + \frac{2\eta \sigma_{r_A}(\boldsymbol{\Lambda}_t^2)}{\kappa^2} \mathsf{Tr}\big((\mathbf{I}_{r_A} - \boldsymbol{\Phi}_t \boldsymbol{\Phi}_t^\top) \boldsymbol{\Phi}_t \boldsymbol{\Phi}_t^\top\big)$$

$$\quad - 2\eta \mathsf{Tr}\Big(\boldsymbol{\Phi}_t \mathbf{H}_t^\top \mathbf{U} + \eta(\mathbf{I}_{r_A} - \boldsymbol{\Phi}_t \boldsymbol{\Phi}_t^\top)\boldsymbol{\Sigma}\boldsymbol{\Phi}_t \boldsymbol{\Phi}_t^\top \boldsymbol{\Sigma}\boldsymbol{\Phi}_t \mathbf{H}_t^\top \mathbf{U}\Big)$$

$$\overset{(g)}{=} \mathsf{Tr}(\boldsymbol{\Phi}_t \boldsymbol{\Phi}_t^\top) + \frac{2\eta \sigma_{r_A}(\boldsymbol{\Lambda}_t^2)}{\kappa^2} \mathsf{Tr}\big((\mathbf{I}_{r_A} - \boldsymbol{\Phi}_t \boldsymbol{\Phi}_t^\top) \boldsymbol{\Phi}_t \boldsymbol{\Phi}_t^\top\big)$$

$$\quad - 2\eta \mathsf{Tr}\big(\mathbf{U}^\top (\mathbf{I}_m - \mathbf{X}_t \mathbf{X}_t^\top)\mathbf{A}\mathbf{X}_t \mathbf{X}_t^\top \boldsymbol{\Delta}_{t-1} \mathbf{X}_t \boldsymbol{\Phi}_t^\top\big)$$

$$\quad - 2\eta \mathsf{Tr}\big(\mathbf{U}^\top (\mathbf{I}_m - \mathbf{X}_t \mathbf{X}_t^\top)\boldsymbol{\Xi}_t \mathbf{X}_t \boldsymbol{\Theta}_t \boldsymbol{\Phi}_t^\top\big)$$

$$\quad - \eta \mathsf{Tr}\big(\boldsymbol{\Phi}_t \mathbf{X}_t^\top \boldsymbol{\Xi}_t \mathbf{X}_t \boldsymbol{\Theta}_t \boldsymbol{\Phi}_t^\top - \boldsymbol{\Phi}_t \boldsymbol{\Theta}_t \mathbf{X}_t^\top \boldsymbol{\Xi}_t \mathbf{X}_t \boldsymbol{\Phi}_t^\top\big)$$

$$\quad - \eta \mathsf{Tr}\big(\boldsymbol{\Phi}_t \mathbf{X}_t^\top \mathbf{A}\mathbf{X}_t \mathbf{X}_t^\top \boldsymbol{\Delta}_{t-1} \mathbf{X}_t \boldsymbol{\Phi}_t^\top - \boldsymbol{\Phi}_t \mathbf{X}_t^\top \boldsymbol{\Delta}_{t-1} \mathbf{X}_t \mathbf{X}_t^\top \mathbf{A}\mathbf{X}_t \boldsymbol{\Phi}_t^\top\big)$$

$$\quad - 2\eta^2 \mathsf{Tr}\big((\mathbf{I}_{r_A} - \boldsymbol{\Phi}_t \boldsymbol{\Phi}_t^\top)\boldsymbol{\Sigma}\boldsymbol{\Phi}_t \boldsymbol{\Phi}_t^\top \boldsymbol{\Sigma}\boldsymbol{\Phi}_t \mathbf{H}_t^\top \mathbf{U}\big)$$

$$\overset{(h)}{=} \mathsf{Tr}(\boldsymbol{\Phi}_t \boldsymbol{\Phi}_t^\top) + \frac{2\eta \sigma_{r_A}(\boldsymbol{\Lambda}_t^2)}{\kappa^2} \mathsf{Tr}\big((\mathbf{I}_{r_A} - \boldsymbol{\Phi}_t \boldsymbol{\Phi}_t^\top) \boldsymbol{\Phi}_t \boldsymbol{\Phi}_t^\top\big)$$

$$\quad - 2\eta \mathsf{Tr}\big(\mathbf{U}^\top (\mathbf{I}_m - \mathbf{X}_t \mathbf{X}_t^\top)\mathbf{U}\boldsymbol{\Sigma}\mathbf{U}^\top \mathbf{X}_t \mathbf{X}_t^\top \boldsymbol{\Delta}_{t-1} \mathbf{X}_t \boldsymbol{\Phi}_t^\top\big)$$

$$\quad - 2\eta \mathsf{Tr}\big(\mathbf{U}^\top (\mathbf{I}_m - \mathbf{X}_t \mathbf{X}_t^\top)\boldsymbol{\Xi}_t \mathbf{X}_t \boldsymbol{\Theta}_t \boldsymbol{\Phi}_t^\top\big)$$

$$\quad - 2\eta^2 \mathsf{Tr}\big((\mathbf{I}_{r_A} - \boldsymbol{\Phi}_t \boldsymbol{\Phi}_t^\top)\boldsymbol{\Sigma}\boldsymbol{\Phi}_t \boldsymbol{\Phi}_t^\top \boldsymbol{\Sigma}\boldsymbol{\Phi}_t \mathbf{H}_t^\top \mathbf{U}\big)$$

where $(g)$ is by substituting $\mathbf{H}_t$ in; and $(h)$ arises from $\mathsf{Tr}(\mathbf{M}) = \mathsf{Tr}(\mathbf{M}^\top)$ for any $\mathbf{M} \in \mathbb{R}^{r_A \times r_A}$. By the Cauchy–Schwarz inequality, we can upper bound the three trace terms as follows.

For the first term, we have that

$$\mathsf{Tr}\big(\mathbf{U}^\top (\mathbf{I}_m - \mathbf{X}_t \mathbf{X}_t^\top)\mathbf{U}\boldsymbol{\Sigma}\mathbf{U}^\top \mathbf{X}_t \mathbf{X}_t^\top \boldsymbol{\Delta}_{t-1} \mathbf{X}_t \boldsymbol{\Phi}_t^\top\big) \tag{15}$$

$$\leq \|\mathbf{U}^\top (\mathbf{I}_m - \mathbf{X}_t \mathbf{X}_t^\top)\mathbf{U}\boldsymbol{\Sigma}\mathbf{U}^\top\|_{\mathsf{F}} \|\mathbf{X}_t \mathbf{X}_t^\top \boldsymbol{\Delta}_{t-1} \mathbf{X}_t \boldsymbol{\Phi}_t^\top\|_{\mathsf{F}}$$

$$\overset{(i)}{\leq} \|(\mathbf{I}_m - \mathbf{X}_t \mathbf{X}_t^\top)\mathbf{U}\|_{\mathsf{F}} \|\boldsymbol{\Delta}_{t-1}\|_{\mathsf{F}}$$

$$= \sqrt{\mathsf{Tr}(\mathbf{I}_{r_A} - \boldsymbol{\Phi}_t \boldsymbol{\Phi}_t^\top)} \|\boldsymbol{\Delta}_{t-1}\|_{\mathsf{F}}.$$

For the second term, we can obtain that

$$\mathsf{Tr}(\mathbf{U}^\top (\mathbf{I}_m - \mathbf{X}_t \mathbf{X}_t^\top)\boldsymbol{\Xi}_t \mathbf{X}_t \boldsymbol{\Theta}_t \boldsymbol{\Phi}_t^\top) \tag{16}$$

$$\leq \|\mathbf{U}^\top (\mathbf{I}_m - \mathbf{X}_t \mathbf{X}_t^\top)\|_{\mathsf{F}} \|\boldsymbol{\Xi}_t \mathbf{X}_t \boldsymbol{\Theta}_t \boldsymbol{\Phi}_t^\top\|_{\mathsf{F}}$$

$$\overset{(i)}{\leq} 2\|\mathbf{U}^\top (\mathbf{I}_m - \mathbf{X}_t \mathbf{X}_t^\top)\|_{\mathsf{F}} \|\boldsymbol{\Xi}_t\|_{\mathsf{F}}$$

$$= 2\sqrt{\mathsf{Tr}(\mathbf{I}_{r_A} - \boldsymbol{\Phi}_t \boldsymbol{\Phi}_t^\top)} \|\boldsymbol{\Xi}_t\|_{\mathsf{F}}.$$

For the third term, it holds that

$$\mathsf{Tr}\big((\mathbf{I}_{r_A} - \boldsymbol{\Phi}_t \boldsymbol{\Phi}_t^\top)\boldsymbol{\Sigma}\boldsymbol{\Phi}_t \boldsymbol{\Phi}_t^\top \boldsymbol{\Sigma}\boldsymbol{\Phi}_t \mathbf{H}_t^\top \mathbf{U}\big) \tag{17}$$

$$\leq \|\mathbf{I}_{r_A} - \boldsymbol{\Phi}_t \boldsymbol{\Phi}_t^\top\|_{\mathsf{F}} \|\boldsymbol{\Sigma}\boldsymbol{\Phi}_t \boldsymbol{\Phi}_t^\top \boldsymbol{\Sigma}\boldsymbol{\Phi}_t \mathbf{H}_t^\top \mathbf{U}\|_{\mathsf{F}}$$

$$= \|\mathbf{U}^\top (\mathbf{I}_m - \mathbf{X}_t \mathbf{X}_t^\top)\mathbf{U}\|_{\mathsf{F}} \|\boldsymbol{\Sigma}\boldsymbol{\Phi}_t \boldsymbol{\Phi}_t^\top \boldsymbol{\Sigma}\boldsymbol{\Phi}_t \mathbf{H}_t^\top \mathbf{U}\|_{\mathsf{F}}$$

$$\overset{(i)}{\leq} \|(\mathbf{I}_m - \mathbf{X}_t \mathbf{X}_t^\top)\mathbf{U}\|_{\mathsf{F}} \|\mathbf{H}_t\|_{\mathsf{F}}$$

$$= \sqrt{\mathsf{Tr}(\mathbf{I}_{r_A} - \boldsymbol{\Phi}_t \boldsymbol{\Phi}_t^\top)} \|\mathbf{H}_t\|_{\mathsf{F}}.$$

Here $(i)$ is from $\|\mathbf{U}\| \leq 1$, $\|\mathbf{\Sigma}\| \leq 1$, $\|\mathbf{X}_t\| \leq 1$, $\|\mathbf{\Phi}_t\| \leq 1$, and $\|\mathbf{\Theta}_t\| \leq 2$. Combining inequalities (14), (15), (16), and (17), it follows that

$$\frac{1}{1 - \eta^2 \beta_t - 16\eta^2 \chi_t} \mathsf{Tr}(\mathbf{\Phi}_{t+1}\mathbf{\Phi}_{t+1}^\top) \geq \mathsf{Tr}(\mathbf{\Phi}_t \mathbf{\Phi}_t^\top) + \frac{2\eta\sigma_{r_A}(\mathbf{\Lambda}_t^2)}{\kappa^2} \mathsf{Tr}\big((\mathbf{I}_{r_A} - \mathbf{\Phi}_t\mathbf{\Phi}_t^\top)\mathbf{\Phi}_t\mathbf{\Phi}_t^\top\big)$$
$$- 2\eta\sqrt{\mathsf{Tr}(\mathbf{I}_{r_A} - \mathbf{\Phi}_t\mathbf{\Phi}_t^\top)}(\|\mathbf{\Delta}_{t-1}\|_\mathsf{F} + 2\|\mathbf{\Xi}_t\|_\mathsf{F})$$
$$- 2\eta^2\sqrt{\mathsf{Tr}(\mathbf{I}_{r_A} - \mathbf{\Phi}_t\mathbf{\Phi}_t^\top)}\|\mathbf{H}_t\|_\mathsf{F}.$$

Reorganizing the terms, we arrive at

$$\mathsf{Tr}(\mathbf{I}_{r_A} - \mathbf{\Phi}_{t+1}\mathbf{\Phi}_{t+1}^\top) - \mathsf{Tr}(\mathbf{I}_{r_A} - \mathbf{\Phi}_t\mathbf{\Phi}_t^\top)$$
$$\leq \eta^2(\beta_t + 16\chi_t)\mathsf{Tr}(\mathbf{\Phi}_t\mathbf{\Phi}_t^\top) - \frac{2\eta(1 - \eta^2\beta_t - 16\eta^2\chi_t)\sigma_{r_A}(\mathbf{\Lambda}_t^2)}{\kappa^2}\mathsf{Tr}\big((\mathbf{I}_{r_A} - \mathbf{\Phi}_t\mathbf{\Phi}_t^\top)\mathbf{\Phi}_t\mathbf{\Phi}_t^\top\big)$$
$$+ (2\eta - 2\eta^3\beta_t - 32\eta^3\chi_t)\sqrt{\mathsf{Tr}(\mathbf{I}_{r_A} - \mathbf{\Phi}_t\mathbf{\Phi}_t^\top)}(\|\mathbf{\Delta}_{t-1}\|_\mathsf{F} + 2\|\mathbf{\Xi}_t\|_\mathsf{F})$$
$$+ (2\eta^2 - 2\eta^4\beta_t - 32\eta^4\chi_t)\sqrt{\mathsf{Tr}(\mathbf{I}_{r_A} - \mathbf{\Phi}_t\mathbf{\Phi}_t^\top)}\|\mathbf{H}_t\|_\mathsf{F}$$
$$\leq \eta^2(\beta_t + 16\chi_t)\mathsf{Tr}(\mathbf{\Phi}_t\mathbf{\Phi}_t^\top) - \frac{2\eta(1 - \eta^2\beta_t - 16\eta^2\chi_t)\sigma_{r_A}(\mathbf{\Lambda}_t^2)}{\kappa^2}\mathsf{Tr}\big((\mathbf{I}_{r_A} - \mathbf{\Phi}_t\mathbf{\Phi}_t^\top)\mathbf{\Phi}_t\mathbf{\Phi}_t^\top\big)$$
$$+ 2\eta\sqrt{\mathsf{Tr}(\mathbf{I}_{r_A} - \mathbf{\Phi}_t\mathbf{\Phi}_t^\top)}(\|\mathbf{\Delta}_{t-1}\|_\mathsf{F} + 2\|\mathbf{\Xi}_t\|_\mathsf{F})$$
$$+ 2\eta^2\sqrt{\mathsf{Tr}(\mathbf{I}_{r_A} - \mathbf{\Phi}_t\mathbf{\Phi}_t^\top)}\|\mathbf{H}_t\|_\mathsf{F}.$$

Together with $\sigma_{r_A}(\mathbf{\Lambda}_t^2) = \sigma_{r_A}(\mathbf{Q}_t^\top \mathbf{\Phi}_t \mathbf{\Phi}_t^\top \mathbf{Q}_t) = \sigma_{r_A}^2(\mathbf{\Phi}_t)$, we conclude the proof. $\qquad\square$

### E.5 PROOF OF LEMMA D.5

*Proof.* From the update formula of $\mathbf{\Theta}_t$, we obtain

$$\begin{aligned}
\|\mathbf{\Theta}_{t+1}\| &= \|\mathbf{X}_{t+1}^\top \mathbf{A}\mathbf{X}_{t+1} - \mathbf{X}_{t+1}^\top \mathbf{\Delta}_t\mathbf{X}_{t+1}\| \\
&\leq \|\mathbf{X}_{t+1}^\top \mathbf{A}\mathbf{X}_{t+1}\| + \|\mathbf{X}_{t+1}^\top \mathbf{\Delta}_t\mathbf{X}_{t+1}\| \\
&\overset{(a)}{\leq} 1 + \|\mathbf{\Delta}_t\| \\
&= 1 + \|(\mathcal{M}^*\mathcal{M} - \mathcal{I})(\mathbf{X}_{t+1}\mathbf{\Theta}_t\mathbf{X}_{t+1}^\top - \mathbf{A})\| \\
&\overset{(b)}{\leq} 1 + \frac{1}{3\sqrt{m}}\|\mathbf{X}_{t+1}\mathbf{\Theta}_t\mathbf{X}_{t+1}^\top - \mathbf{A}\|_\mathsf{F} \\
&\leq 1 + \frac{\sqrt{m}}{3\sqrt{m}}\|\mathbf{X}_{t+1}\mathbf{\Theta}_t\mathbf{X}_{t+1}^\top - \mathbf{A}\| \\
&\leq 1 + \frac{1}{3}(\|\mathbf{\Theta}_t\| + \|\mathbf{A}\|) \\
&\leq 2,
\end{aligned}$$

where $(a)$ is by $\|\mathbf{X}_t\|, \|\mathbf{A}\| \leq 1$; and $(b)$ follows from Lemma F.11 and $\mathrm{rank}(\mathbf{X}_{t+1}\mathbf{\Theta}_t\mathbf{X}_{t+1}^\top - \mathbf{A}) \leq \mathrm{rank}(\mathbf{X}_{t+1}\mathbf{\Theta}_t\mathbf{X}_{t+1}^\top) + \mathrm{rank}(\mathbf{A}) \leq r + r_A$. $\qquad\square$

### E.6 PROOF OF LEMMA D.6

*Proof.* Let $\mathbf{L}_t := \mathbf{X}_t^\top \mathbf{A}\mathbf{X}_t\mathbf{X}_t^\top \mathbf{\Delta}_{t-1}\mathbf{X}_t + \mathbf{X}^\top \mathbf{\Xi}_t\mathbf{X}_t\mathbf{\Theta}_t + \frac{1}{2}(\mathbf{\Theta}_t\mathbf{X}_t^\top \mathbf{\Xi}_t\mathbf{X}_t - \mathbf{X}_t^\top \mathbf{\Xi}_t\mathbf{X}_t\mathbf{\Theta}_t)$
$\qquad\qquad + \frac{1}{2}(\mathbf{X}_t^\top \mathbf{\Delta}_{t-1}\mathbf{X}_t\mathbf{X}_t^\top \mathbf{A}\mathbf{X}_t - \mathbf{X}_t^\top \mathbf{A}\mathbf{X}_t\mathbf{X}_t^\top \mathbf{\Delta}_{t-1}\mathbf{X}_t).$

Applying the triangular inequality, we obtain

$$
\begin{aligned}
\|\mathbf{L}_t\| \leq{} & \|\mathbf{X}_t^\top \mathbf{A}\mathbf{X}_t\mathbf{X}_t^\top \boldsymbol{\Delta}_{t-1}\mathbf{X}_t\| + \|\mathbf{X}^\top \boldsymbol{\Xi}_t\mathbf{X}_t\boldsymbol{\Theta}_t\| + \frac{1}{2}(\|\boldsymbol{\Theta}_t\mathbf{X}_t^\top \boldsymbol{\Xi}_t\mathbf{X}_t\| + \|\mathbf{X}_t^\top \boldsymbol{\Xi}_t\mathbf{X}_t\boldsymbol{\Theta}_t\|) \\
& + \frac{1}{2}(\|\mathbf{X}_t^\top \boldsymbol{\Delta}_{t-1}\mathbf{X}_t\mathbf{X}_t^\top \mathbf{A}\mathbf{X}_t\| + \|\mathbf{X}_t^\top \mathbf{A}\mathbf{X}_t\mathbf{X}_t^\top \boldsymbol{\Delta}_{t-1}\mathbf{X}_t\|) \\
& \overset{(a)}{\leq} 2\|\boldsymbol{\Delta}_{t-1}\| + 4\|\boldsymbol{\Xi}_t\|,
\end{aligned}
\tag{18}
$$

where $(a)$ is from $\|\mathbf{X}_t\| \leq 1, \|\mathbf{A}\| \leq 1$ and $\|\boldsymbol{\Theta}_t\| \leq 2$. Multiplying the update formula (4) on the left by $\mathbf{U}_\perp^\top$, we have that

$$
\begin{aligned}
\boldsymbol{\Psi}_{t+1} ={} & \mathbf{U}_\perp^\top (\mathbf{X}_t - \eta\mathbf{G}_t)(\mathbf{I}_r + \eta^2\mathbf{G}_t^\top \mathbf{G}_t)^{-1/2} \\
& \overset{(b)}{=} \left(\boldsymbol{\Psi}_t - \eta\boldsymbol{\Psi}_t\mathbf{X}_t^\top \mathbf{A}\mathbf{X}_t\mathbf{X}_t^\top \mathbf{A}\mathbf{X}_t + \eta\boldsymbol{\Psi}_t\mathbf{L}_t - \eta\mathbf{U}_\perp^\top \boldsymbol{\Xi}_t\mathbf{X}_t\boldsymbol{\Theta}_t\right)(\mathbf{I}_r + \eta^2\mathbf{G}_t^\top \mathbf{G}_t)^{-1/2} \\
& = \left(\boldsymbol{\Psi}_t\left(\mathbf{I}_r - \eta\mathbf{X}_t^\top \mathbf{A}\mathbf{X}_t\mathbf{X}_t^\top \mathbf{A}\mathbf{X}_t + \eta\mathbf{L}_t\right) - \eta\mathbf{U}_\perp^\top \boldsymbol{\Xi}_t\mathbf{X}_t\boldsymbol{\Theta}_t\right)(\mathbf{I}_r + \eta^2\mathbf{G}_t^\top \mathbf{G}_t)^{-1/2},
\end{aligned}
$$

where $(b)$ is by expanding $\mathbf{G}_t$ directly. Consequently, we have the following upper bound for $\boldsymbol{\Psi}_{t+1}\boldsymbol{\Psi}_{t+1}^\top$:

$$
\begin{aligned}
\boldsymbol{\Psi}_{t+1}\boldsymbol{\Psi}_{t+1}^\top ={} & \left(\boldsymbol{\Psi}_t\left(\mathbf{I}_r - \eta\mathbf{X}_t^\top \mathbf{A}\mathbf{X}_t\mathbf{X}_t^\top \mathbf{A}\mathbf{X}_t + \eta\mathbf{L}_t\right) - \eta\mathbf{U}_\perp^\top \boldsymbol{\Xi}_t\mathbf{X}_t\boldsymbol{\Theta}_t\right)(\mathbf{I}_r + \eta^2\mathbf{G}_t^\top \mathbf{G}_t)^{-1} \\
& \qquad \left(\boldsymbol{\Psi}_t\left(\mathbf{I}_r - \eta\mathbf{X}_t^\top \mathbf{A}\mathbf{X}_t\mathbf{X}_t^\top \mathbf{A}\mathbf{X}_t + \eta\mathbf{L}_t\right) - \eta\mathbf{U}_\perp^\top \boldsymbol{\Xi}_t\mathbf{X}_t\boldsymbol{\Theta}_t\right)^\top \\
& \overset{(c)}{\preceq} \left(\boldsymbol{\Psi}_t\left(\mathbf{I}_r - \eta\mathbf{X}_t^\top \mathbf{A}\mathbf{X}_t\mathbf{X}_t^\top \mathbf{A}\mathbf{X}_t + \eta\mathbf{L}_t\right) - \eta\mathbf{U}_\perp^\top \boldsymbol{\Xi}_t\mathbf{X}_t\boldsymbol{\Theta}_t\right) \\
& \qquad \left(\boldsymbol{\Psi}_t\left(\mathbf{I}_r - \eta\mathbf{X}_t^\top \mathbf{A}\mathbf{X}_t\mathbf{X}_t^\top \mathbf{A}\mathbf{X}_t + \eta\mathbf{L}_t\right) - \eta\mathbf{U}_\perp^\top \boldsymbol{\Xi}_t\mathbf{X}_t\boldsymbol{\Theta}_t\right)^\top \\
& = \boldsymbol{\Psi}_t\left(\mathbf{I}_r - \eta\mathbf{X}_t^\top \mathbf{A}\mathbf{X}_t\mathbf{X}_t^\top \mathbf{A}\mathbf{X}_t + \eta\mathbf{L}_t\right) \\
& \qquad \left(\mathbf{I}_r - \eta\mathbf{X}_t^\top \mathbf{A}\mathbf{X}_t\mathbf{X}_t^\top \mathbf{A}\mathbf{X}_t + \eta\mathbf{L}_t\right)^\top \boldsymbol{\Psi}_t^\top \\
& - \eta\mathbf{U}_\perp^\top \boldsymbol{\Xi}_t\mathbf{X}_t\boldsymbol{\Theta}_t\left(\mathbf{I}_r - \eta\mathbf{X}_t^\top \mathbf{A}\mathbf{X}_t\mathbf{X}_t^\top \mathbf{A}\mathbf{X}_t + \eta\mathbf{L}_t\right)^\top \boldsymbol{\Psi}_t^\top \\
& - \eta\boldsymbol{\Psi}_t\left(\mathbf{I}_r - \eta\mathbf{X}_t^\top \mathbf{A}\mathbf{X}_t\mathbf{X}_t^\top \mathbf{A}\mathbf{X}_t + \eta\mathbf{L}_t\right)\boldsymbol{\Theta}_t\mathbf{X}_t^\top \boldsymbol{\Xi}_t\mathbf{U}_\perp \\
& + \eta^2\mathbf{U}_\perp^\top \boldsymbol{\Xi}_t\mathbf{X}_t\boldsymbol{\Theta}_t^2\mathbf{X}_t^\top \boldsymbol{\Xi}_t\mathbf{U}_\perp,
\end{aligned}
\tag{19}
$$

where $(c)$ is from that $(\mathbf{I}_r + \eta^2\mathbf{G}_t^\top \mathbf{G}_t)^{-1}$ is PSD and all of its eigenvalues are smaller than 1. Since $\mathbf{Y}\mathbf{Y}^\top \preceq \|\mathbf{Y}\|^2\mathbf{I}_r$ holds for any symmetric matrix $\mathbf{Y} \in \mathbb{R}^{r\times r}$ and by Lemma F.12, we can upper bound the three terms as follows.

For the first term, we can obtain that

$$
\begin{aligned}
\boldsymbol{\Psi}_t\left(\mathbf{I}_r - \eta\mathbf{X}_t^\top \mathbf{A}\mathbf{X}_t\mathbf{X}_t^\top \mathbf{A}\mathbf{X}_t + \eta\mathbf{L}_t\right) & \\
\left(\mathbf{I}_r - \eta\mathbf{X}_t^\top \mathbf{A}\mathbf{X}_t\mathbf{X}_t^\top \mathbf{A}\mathbf{X}_t + \eta\mathbf{L}_t\right)^\top \boldsymbol{\Psi}_t^\top & \\
\preceq \|\mathbf{I}_r - \eta\mathbf{X}_t^\top \mathbf{A}\mathbf{X}_t\mathbf{X}_t^\top \mathbf{A}\mathbf{X}_t + \eta\mathbf{L}_t\|^2\boldsymbol{\Psi}_t\boldsymbol{\Psi}_t^\top. &
\end{aligned}
\tag{20}
$$

For the second term, it holds that

$$
\begin{aligned}
\mathbf{U}_\perp^\top \boldsymbol{\Xi}_t\mathbf{X}_t\boldsymbol{\Theta}_t\left(\mathbf{I}_r - \eta\mathbf{X}_t^\top \mathbf{A}\mathbf{X}_t\mathbf{X}_t^\top \mathbf{A}\mathbf{X}_t + \eta\mathbf{L}_t\right)^\top \boldsymbol{\Psi}_t^\top & \\
+ \boldsymbol{\Psi}_t\left(\mathbf{I}_r - \eta\mathbf{X}_t^\top \mathbf{A}\mathbf{X}_t\mathbf{X}_t^\top \mathbf{A}\mathbf{X}_t + \eta\mathbf{L}_t\right)\boldsymbol{\Theta}_t\mathbf{X}_t^\top \boldsymbol{\Xi}_t\mathbf{U}_\perp & \\
\preceq 2\|\boldsymbol{\Psi}_t\left(\mathbf{I}_r - \eta\mathbf{X}_t^\top \mathbf{A}\mathbf{X}_t\mathbf{X}_t^\top \mathbf{A}\mathbf{X}_t + \eta\mathbf{L}_t\right)\|\|\boldsymbol{\Theta}_t\mathbf{X}_t^\top \boldsymbol{\Xi}_t\mathbf{U}_\perp\|\mathbf{I}_{m-r_A}. &
\end{aligned}
\tag{21}
$$

For the third term, we have that

$$
\mathbf{U}_\perp^\top \boldsymbol{\Xi}_t\mathbf{X}_t\boldsymbol{\Theta}_t^2\mathbf{X}_t^\top \boldsymbol{\Xi}_t\mathbf{U}_\perp \preceq \|\mathbf{U}_\perp^\top \boldsymbol{\Xi}_t\mathbf{X}_t\boldsymbol{\Theta}_t^2\mathbf{X}_t^\top \boldsymbol{\Xi}_t\mathbf{U}_\perp\|\mathbf{I}_{m-r_A}.
\tag{22}
$$

Combining inequalities (19), (20), (21) and (22), it follows that

$$
\begin{aligned}
\boldsymbol{\Psi}_{t+1}\boldsymbol{\Psi}_{t+1}^\top \preceq\ & \|\mathbf{I}_r - \eta\mathbf{X}_t^\top\mathbf{A}\mathbf{X}_t\mathbf{X}_t^\top\mathbf{A}\mathbf{X}_t + \eta\mathbf{L}_t\|^2\boldsymbol{\Psi}_t\boldsymbol{\Psi}_t^\top \\
& + 2\eta\|\boldsymbol{\Psi}_t\left(\mathbf{I}_r - \eta\mathbf{X}_t^\top\mathbf{A}\mathbf{X}_t\mathbf{X}_t^\top\mathbf{A}\mathbf{X}_t + \eta\mathbf{L}_t\right)\|\|\boldsymbol{\Theta}_t\mathbf{X}_t^\top\boldsymbol{\Xi}_t\mathbf{U}_\perp\|\mathbf{I}_{m-r_A} \\
& + \eta^2\|\mathbf{U}_\perp^\top\boldsymbol{\Xi}_t\mathbf{X}_t\boldsymbol{\Theta}_t^2\mathbf{X}_t^\top\boldsymbol{\Xi}_t\mathbf{U}_\perp\|\mathbf{I}_{m-r_A} \\
\overset{(d)}{\preceq}\ & (\|\mathbf{I}_r - \eta\mathbf{X}_t^\top\mathbf{A}\mathbf{X}_t\mathbf{X}_t^\top\mathbf{A}\mathbf{X}_t\| + \eta\|\mathbf{L}_t\|)^2\boldsymbol{\Psi}_t\boldsymbol{\Psi}_t^\top \\
& + 2\eta\|\boldsymbol{\Psi}_t\|((\|\mathbf{I}_r - \eta\mathbf{X}_t^\top\mathbf{A}\mathbf{X}_t\mathbf{X}_t^\top\mathbf{A}\mathbf{X}_t\| + \eta\|\mathbf{L}_t\|)\|\boldsymbol{\Theta}_t\mathbf{X}_t^\top\boldsymbol{\Xi}_t\mathbf{U}_\perp\|\mathbf{I}_{m-r_A} \\
& + \eta^2\|\mathbf{U}_\perp^\top\boldsymbol{\Xi}_t\mathbf{X}_t\boldsymbol{\Theta}_t^2\mathbf{X}_t^\top\boldsymbol{\Xi}_t\mathbf{U}_\perp\|\mathbf{I}_{m-r_A} \\
\overset{(e)}{\preceq}\ & (1 + \eta\|\mathbf{L}_t\|)^2\boldsymbol{\Psi}_t\boldsymbol{\Psi}_t^\top + 4\eta(1 + \eta\|\mathbf{L}_t\|)\|\boldsymbol{\Xi}_t\|\mathbf{I}_{m-r_A} + 4\eta^2\|\boldsymbol{\Xi}_t\|^2\mathbf{I}_{m-r_A},
\end{aligned}
$$

where $(d)$ is by triangular inequality; and $(e)$ is from that all the eigenvalues of the PSD matrix $\mathbf{X}_t^\top\mathbf{A}\mathbf{X}_t\mathbf{X}_t^\top\mathbf{A}\mathbf{X}_t$ are smaller than 1, along with $\|\boldsymbol{\Psi}_t\| \le 1, \|\mathbf{X}_t\| \le 1, \|\mathbf{U}_\perp\| \le 1$ and $\|\boldsymbol{\Theta}_t\| \le 2$.

From (18), we obtain $\|\mathbf{L}_t\| \le 2\|\boldsymbol{\Delta}_{t-1}\| + 4\|\boldsymbol{\Xi}_t\|$. Then, we can further simplify the inequality as

$$
\begin{aligned}
\boldsymbol{\Psi}_{t+1}\boldsymbol{\Psi}_{t+1}^\top \preceq\ & \left(1 + 2\eta(\|\boldsymbol{\Delta}_{t-1}\| + 2\|\boldsymbol{\Xi}_t\|)\right)^2\boldsymbol{\Psi}_t\boldsymbol{\Psi}_t^\top \\
& + \left(4\eta\|\boldsymbol{\Xi}_t\| + 4\eta^2(5\|\boldsymbol{\Xi}_t\|^2 + 2\|\boldsymbol{\Delta}_{t-1}\|\|\boldsymbol{\Xi}_t\|)\right)\mathbf{I}_{m-r_A}. \tag{23}
\end{aligned}
$$

From Lemma F.11 and our assumption of the RIP property of $\mathcal{M}(\cdot)$, we obtain upper bounds for the two error terms.

$$
\begin{aligned}
\|\boldsymbol{\Delta}_{t-1}\| &= \|(\mathcal{M}^*\mathcal{M} - \mathcal{I})(\mathbf{X}_t\boldsymbol{\Theta}_{t-1}\mathbf{X}_t^\top - \mathbf{A})\| \\
&\le \delta\|\mathbf{X}_t\boldsymbol{\Theta}_{t-1}\mathbf{X}_t^\top - \mathbf{A}\|_\mathsf{F} \\
&\le \delta(\|\mathbf{X}_t\boldsymbol{\Theta}_{t-1}\mathbf{X}_t^\top\|_\mathsf{F} + \|\mathbf{A}\|_\mathsf{F}) \\
&\overset{(f)}{\le} (2\sqrt{r} + \sqrt{r_A})\delta,
\end{aligned}
$$

$$
\begin{aligned}
\|\boldsymbol{\Xi}_t\| &= \|(\mathcal{M}^*\mathcal{M} - \mathcal{I})(\mathbf{X}_t\boldsymbol{\Theta}_t\mathbf{X}_t^\top - \mathbf{A})\| \\
&\le \delta\|\mathbf{X}_t\boldsymbol{\Theta}_t\mathbf{X}_t^\top - \mathbf{A}\|_\mathsf{F} \\
&\le \delta(\|\mathbf{X}_t\boldsymbol{\Theta}_t\mathbf{X}_t\|_\mathsf{F} + \|\mathbf{A}\|_\mathsf{F}) \\
&\overset{(f)}{\le} (2\sqrt{r} + \sqrt{r_A})\delta,
\end{aligned}
$$

where $(f)$ is from $\|\mathbf{X}_t\| \le 1, \|\boldsymbol{\Theta}_{t-1}\|_\mathsf{F} \le \sqrt{r}\|\boldsymbol{\Theta}_{t-1}\| \le 2\sqrt{r}, \|\boldsymbol{\Theta}_t\|_\mathsf{F} \le \sqrt{r}\|\boldsymbol{\Theta}_t\| \le 2\sqrt{r}$, and $\|\mathbf{A}\|_\mathsf{F} \le \sqrt{r_A}\|\mathbf{A}\| \le \sqrt{r_A}$. Plugging these two upper bounds into (23), we arrive at

$$
\boldsymbol{\Psi}_{t+1}\boldsymbol{\Psi}_{t+1}^\top \preceq \left(1 + 6\eta(\sqrt{r_A} + 2\sqrt{r})\delta\right)^2\boldsymbol{\Psi}_t\boldsymbol{\Psi}_t^\top + \left(4\eta(\sqrt{r_A} + 2\sqrt{r})\delta + 28\eta^2(\sqrt{r_A} + 2\sqrt{r})^2\delta^2\right)\mathbf{I}_{m-r_A}.
$$

We now consider the relationship between $\boldsymbol{\Psi}_1\boldsymbol{\Psi}_1^\top$ and $\boldsymbol{\Psi}_0\boldsymbol{\Psi}_0^\top$.

Let $\tilde{\mathbf{L}}_0 := \frac{1}{2}(\mathbf{X}_0^\top\mathbf{A}\mathbf{X}_0\boldsymbol{\Theta}_0 + \boldsymbol{\Theta}_0\mathbf{X}_0^\top\mathbf{A}\mathbf{X}_0) - \frac{1}{2}(\mathbf{X}_0^\top\boldsymbol{\Xi}_0\mathbf{X}_0\boldsymbol{\Theta}_0 + \boldsymbol{\Theta}_0\mathbf{X}_0\boldsymbol{\Xi}_0\mathbf{X}_0)$.

Multiplying the update formula (4) at $t = 0$ on the left by $\mathbf{U}_\perp^\top$, we have that

$$
\boldsymbol{\Psi}_1 = \mathbf{U}_\perp^\top(\mathbf{X}_0 - \eta\mathbf{G}_0)(\mathbf{I}_r + \eta^2\mathbf{G}_0^\top\mathbf{G}_0)^{-1/2}.
$$

Consequently, we derive the following upper bound on $\boldsymbol{\Psi}_1\boldsymbol{\Psi}_1^\top$:

$$
\begin{aligned}
\boldsymbol{\Psi}_1\boldsymbol{\Psi}_1^\top &= \mathbf{U}_\perp^\top(\mathbf{X}_0 - \eta\mathbf{G}_0)(\mathbf{I}_r + \eta^2\mathbf{G}_0^\top\mathbf{G}_0)^{-1}(\mathbf{X}_0 - \eta\mathbf{G}_0)^\top\mathbf{U}_\perp \\
&\overset{(g)}{\preceq} \mathbf{U}_\perp^\top(\mathbf{X}_0 - \eta\mathbf{G}_0)(\mathbf{X}_0 - \eta\mathbf{G}_0)^\top\mathbf{U}_\perp \\
&\overset{(h)}{=} \left(\boldsymbol{\Psi}_0(\mathbf{I}_r - \eta\tilde{\mathbf{L}}_0) - \eta\mathbf{U}_\perp^\top\boldsymbol{\Xi}_0\mathbf{X}_0\boldsymbol{\Theta}_0\right)\left(\boldsymbol{\Psi}_0(\mathbf{I}_r - \eta\tilde{\mathbf{L}}_0) - \eta\mathbf{U}_\perp^\top\boldsymbol{\Xi}_0\mathbf{X}_0\boldsymbol{\Theta}_0\right)^\top \\
&= \boldsymbol{\Psi}_0(\mathbf{I}_r - \eta\tilde{\mathbf{L}}_0)(\mathbf{I}_r - \eta\tilde{\mathbf{L}}_0)^\top\boldsymbol{\Psi}_0^\top - \eta\boldsymbol{\Psi}_0(\mathbf{I}_r - \eta\tilde{\mathbf{L}}_0)\boldsymbol{\Theta}_0\mathbf{X}_0^\top\boldsymbol{\Xi}_0\mathbf{U}_\perp \\
&\quad - \eta\mathbf{U}_\perp^\top\boldsymbol{\Xi}_0\mathbf{X}_0\boldsymbol{\Theta}_0(\mathbf{I}_r - \eta\tilde{\mathbf{L}}_0)^\top\boldsymbol{\Psi}_0^\top + \eta^2\mathbf{U}_\perp^\top\boldsymbol{\Xi}_0\mathbf{X}_0\boldsymbol{\Theta}_0^2\mathbf{X}_0^\top\boldsymbol{\Xi}_0\mathbf{U}_\perp, \tag{24}
\end{aligned}
$$

where $(g)$ is from that $(\mathbf{I}_r + \eta^2 \mathbf{G}_0^\top \mathbf{G}_0)^{-1}$ is PSD and all of its eigenvalues are smaller than 1; and $(h)$ is by expanding the expression of $\mathbf{G}_0$ directly. Since $\mathbf{Y}\mathbf{Y}^\top \preceq \|\mathbf{Y}\|^2 \mathbf{I}_r$ holds for any symmetric matrix $\mathbf{Y} \in \mathbb{R}^{r \times r}$ and by Lemma F.12, we can upper bound the three terms as follows.

For the first term, it holds that

$$\boldsymbol{\Psi}_0(\mathbf{I}_r - \eta \tilde{\mathbf{L}}_0)(\mathbf{I}_r - \eta \tilde{\mathbf{L}}_0)^\top \boldsymbol{\Psi}_0^\top \preceq \|\mathbf{I}_r - \eta \tilde{\mathbf{L}}_0\|^2 \boldsymbol{\Psi}_0 \boldsymbol{\Psi}_0^\top. \tag{25}$$

For the second term, we have that

$$\boldsymbol{\Psi}_0(\mathbf{I}_r - \eta \tilde{\mathbf{L}}_0)\boldsymbol{\Theta}_0 \mathbf{X}_0^\top \boldsymbol{\Xi}_0 \mathbf{U}_\perp + \mathbf{U}_\perp^\top \boldsymbol{\Xi}_0 \mathbf{X}_0 \boldsymbol{\Theta}_0 (\mathbf{I}_r - \eta \tilde{\mathbf{L}}_0)^\top \boldsymbol{\Psi}_0^\top$$
$$\preceq 2\|\boldsymbol{\Psi}_0(\mathbf{I}_r - \eta \tilde{\mathbf{L}}_0)\|\|\boldsymbol{\Theta}_0 \mathbf{X}_0^\top \boldsymbol{\Xi}_0 \mathbf{U}_\perp\| \mathbf{I}_{m-r_A}. \tag{26}$$

For the third term, we can obtain that

$$\mathbf{U}_\perp^\top \boldsymbol{\Xi}_0 \mathbf{X}_0 \boldsymbol{\Theta}_0^2 \mathbf{X}_0^\top \boldsymbol{\Xi}_0 \mathbf{U}_\perp \preceq \|\mathbf{U}_\perp^\top \boldsymbol{\Xi}_0 \mathbf{X}_0 \boldsymbol{\Theta}_0^2 \mathbf{X}_0^\top \boldsymbol{\Xi}_0 \mathbf{U}_\perp\| \mathbf{I}_{m-r_A}. \tag{27}$$

Combining inequalities (24), (25), (26) and (27), it follows that

$$\boldsymbol{\Psi}_1 \boldsymbol{\Psi}_1^\top \preceq \|\mathbf{I}_r - \eta \tilde{\mathbf{L}}_0\|^2 \boldsymbol{\Psi}_0 \boldsymbol{\Psi}_0^\top + 2\eta \|\boldsymbol{\Psi}_0(\mathbf{I}_r - \eta \tilde{\mathbf{L}}_0)\|\|\boldsymbol{\Theta}_0 \mathbf{X}_0^\top \boldsymbol{\Xi}_0 \mathbf{U}_\perp\| \mathbf{I}_{m-r_A}$$
$$+ \eta^2 \|\mathbf{U}_\perp^\top \boldsymbol{\Xi}_0 \mathbf{X}_0 \boldsymbol{\Theta}_0^2 \mathbf{X}_0^\top \boldsymbol{\Xi}_0 \mathbf{U}_\perp\| \mathbf{I}_{m-r_A}$$
$$\stackrel{(i)}{\preceq} (1 + \eta\|\tilde{\mathbf{L}}_0\|)^2 \boldsymbol{\Psi}_0 \boldsymbol{\Psi}_0^\top + 4\eta(1 + \eta\|\tilde{\mathbf{L}}_0\|)\|\boldsymbol{\Xi}_0\| \mathbf{I}_{m-r_A} + 4\eta^2 \|\boldsymbol{\Xi}_0\|^2 \mathbf{I}_{m-r_A}, \tag{28}$$

where $(i)$ is by $\|\mathbf{X}_0\| \le 1$, $\|\mathbf{U}_\perp\| \le 1$, and $\|\boldsymbol{\Theta}_0\| \le 2$.

From Lemma F.11 and our assumption of the RIP property of $\mathcal{M}(\cdot)$, we have that

$$\|\boldsymbol{\Xi}_0\| = \|(\mathcal{M}^* \mathcal{M} - \mathcal{I})(\mathbf{X}_0 \boldsymbol{\Theta}_0 \mathbf{X}_0^\top - \mathbf{A})\|$$
$$\le \delta\|\mathbf{X}_0 \boldsymbol{\Theta}_0 \mathbf{X}_0^\top - \mathbf{A}\|_\mathsf{F}$$
$$\le \delta(\|\mathbf{X}_0 \boldsymbol{\Theta}_0 \mathbf{X}_0^\top\|_\mathsf{F} + \|\mathbf{A}\|_\mathsf{F})$$
$$\le (2\sqrt{r} + \sqrt{r_A})\delta.$$

Then, we can bound $\|\tilde{\mathbf{L}}_0\|$ as follows:

$$\|\tilde{\mathbf{L}}_0\| = \frac{1}{2}\|\mathbf{X}_0^\top \mathbf{A} \mathbf{X}_0 \boldsymbol{\Theta}_0 + \boldsymbol{\Theta}_0 \mathbf{X}_0^\top \mathbf{A} \mathbf{X}_0 - (\mathbf{X}_0^\top \boldsymbol{\Xi}_0 \mathbf{X}_0 \boldsymbol{\Theta}_0 + \boldsymbol{\Theta}_0 \mathbf{X}_0 \boldsymbol{\Xi}_0 \mathbf{X}_0)\|$$
$$\le 2 + 2\|\boldsymbol{\Xi}_0\|$$
$$\le 2 + 2(2\sqrt{r} + \sqrt{r_A})\delta.$$

Plugging theses two upper bounds into inequality (28), we finally arrive at

$$\boldsymbol{\Psi}_1 \boldsymbol{\Psi}_1^\top \preceq (1 + 2\eta + 2\eta(\sqrt{r_A} + 2\sqrt{r})\delta)^2 \boldsymbol{\Psi}_0 \boldsymbol{\Psi}_0^\top + \left(12\eta(\sqrt{r_A} + 2\sqrt{r})\delta + 8\eta^2(\sqrt{r_A} + 2\sqrt{r})^2 \delta^2\right) \mathbf{I}_{m-r_A}.$$

$\square$

### E.7 PROOF OF LEMMA D.7

*Proof.* We first estimate $\|\tilde{\mathbf{G}}_t\|$ and $\|\mathbf{G}_t\|$. From the expression of $\tilde{\mathbf{G}}_t$, we have that

$$\|\tilde{\mathbf{G}}_t\| = \|[\mathcal{M}^* \mathcal{M}(\mathbf{X}_t \boldsymbol{\Theta}_t \mathbf{X}_t^\top - \mathbf{A})]\mathbf{X}_t \boldsymbol{\Theta}_t\|$$
$$\stackrel{(a)}{\le} 2\|\mathcal{M}^* \mathcal{M}(\mathbf{X}_t \boldsymbol{\Theta}_t \mathbf{X}_t^\top - \mathbf{A})\|$$
$$\le 2(\|(\mathcal{M}^* \mathcal{M} - \mathcal{I})(\mathbf{X}_t \boldsymbol{\Theta}_t \mathbf{X}_t^\top - \mathbf{A})\| + \|\mathbf{X}_t \boldsymbol{\Theta}_t \mathbf{X}_t^\top - \mathbf{A}\|)$$
$$\stackrel{(b)}{\le} \frac{2}{\sqrt{m}}\|\mathbf{X}_t \boldsymbol{\Theta}_t \mathbf{X}_t^\top - \mathbf{A}\|_\mathsf{F} + 2\|\mathbf{X}_t \boldsymbol{\Theta}_t \mathbf{X}_t^\top - \mathbf{A}\|$$
$$\le 4\|\mathbf{X}_t \boldsymbol{\Theta}_t \mathbf{X}_t^\top - \mathbf{A}\|$$
$$\le 4(\|\mathbf{X}_t \boldsymbol{\Theta}_t \mathbf{X}_t^\top\| + \|\mathbf{A}\|)$$
$$\stackrel{(a)}{\le} 12,$$

where $(a)$ is due to $\|\mathbf{X}_t\| \leq 1$, $\|\mathbf{\Theta}_t\| \leq 2$, and $\|\mathbf{A}\| \leq 1$; and $(b)$ is from Lemma F.11. Analogously, we can upper bound $\|\mathbf{G}_t\|$ as follows:

$$
\begin{aligned}
\|\mathbf{G}_t\| &= \|(\mathbf{I}_m - \mathbf{X}_t\mathbf{X}_t^\top)\tilde{\mathbf{G}}_t + \frac{\mathbf{X}_t}{2}(\mathbf{X}_t^\top\tilde{\mathbf{G}}_t - \tilde{\mathbf{G}}_t^\top\mathbf{X}_t)\| \\
&\leq \|\mathbf{I}_m - \mathbf{X}_t\mathbf{X}_t^\top\|\|\tilde{\mathbf{G}}_t\| + \|\mathbf{X}_t\|\|\tilde{\mathbf{G}}_t^\top\mathbf{X}_t\| \\
&\overset{(c)}{\leq} 2\|\tilde{\mathbf{G}}_t\| \\
&\leq 24,
\end{aligned}
$$

where $(c)$ follows from $\|\mathbf{X}_t\| \leq 1$ and the fact that all the eigenvalues of the PSD matrix $\mathbf{I}_m - \mathbf{X}_t\mathbf{X}_t^\top$ are less than 1. Multiplying the update formula (4) on the left by $\mathbf{U}^\top$, we obtain

$$
\begin{aligned}
\mathbf{\Phi}_{t+1}\mathbf{\Phi}_{t+1}^\top &= \mathbf{U}^\top\mathbf{X}_{t+1}\mathbf{X}_{t+1}^\top\mathbf{U} \\
&= (\mathbf{\Phi}_t - \eta\mathbf{U}^\top\mathbf{G}_t)(\mathbf{I}_r + \eta^2\mathbf{G}_t^\top\mathbf{G}_t)^{-1}(\mathbf{\Phi}_t^\top - \eta\mathbf{G}_t^\top\mathbf{U}) \\
&\overset{(d)}{\succeq} (\mathbf{\Phi}_t - \eta\mathbf{U}^\top\mathbf{G}_t)(\mathbf{I}_r - \eta^2\mathbf{G}_t^\top\mathbf{G}_t)(\mathbf{\Phi}_t^\top - \eta\mathbf{G}_t^\top\mathbf{U}) \\
&= \mathbf{\Phi}_t\mathbf{\Phi}_t^\top - \eta^2\mathbf{\Phi}_t\mathbf{G}_t^\top\mathbf{G}_t\mathbf{\Phi}_t^\top - \eta(\mathbf{\Phi}_t\mathbf{G}_t^\top\mathbf{U} + \mathbf{U}^\top\mathbf{G}_t\mathbf{\Phi}_t^\top) \\
&\quad + \eta^3(\mathbf{\Phi}_t\mathbf{G}_t^\top\mathbf{G}_t\mathbf{G}_t^\top\mathbf{U} + \mathbf{U}^\top\mathbf{G}_t\mathbf{G}_t^\top\mathbf{G}_t\mathbf{\Phi}_t^\top) \\
&\quad - \eta^4\mathbf{U}^\top\mathbf{G}_t\mathbf{G}_t^\top\mathbf{G}_t\mathbf{G}_t^\top\mathbf{U} + \eta^2\mathbf{U}^\top\mathbf{G}_t\mathbf{G}_t^\top\mathbf{U} \\
&\overset{(e)}{\succeq} \mathbf{\Phi}_t\mathbf{\Phi}_t^\top - \big(\eta^2\|\mathbf{\Phi}_t\mathbf{G}_t^\top\mathbf{G}_t\mathbf{\Phi}_t^\top\| + 2\eta\|\mathbf{\Phi}_t\mathbf{G}_t^\top\mathbf{U}\| \\
&\quad + 2\eta^3\|\mathbf{\Phi}_t\mathbf{G}_t^\top\mathbf{G}_t\mathbf{G}_t^\top\mathbf{U}\| + \eta^4\|\mathbf{U}^\top\mathbf{G}_t\mathbf{G}_t^\top\mathbf{G}_t\mathbf{G}_t^\top\mathbf{U}\|\big)\mathbf{I}_{r_A} \\
&\overset{(f)}{\succeq} \mathbf{\Phi}_t\mathbf{\Phi}_t^\top - \frac{1}{10r_A}\mathbf{I}_{r_A},
\end{aligned}
$$

where $(d)$ is from Lemma F.1; $(e)$ is by Lemma F.12; and $(f)$ is due to $\|\mathbf{\Phi}_t\| \leq 1, \|\mathbf{U}\| \leq 1, \|\mathbf{G}_t\| \leq 24$ and $\eta \leq \frac{1}{500r_A}$. By subtracting the inequality from $\mathbf{I}_{r_A}$, it follows that

$$
\mathbf{I}_{r_A} - \mathbf{\Phi}_{t+1}\mathbf{\Phi}_{t+1}^\top \preceq \mathbf{I}_{r_A} - \mathbf{\Phi}_t\mathbf{\Phi}_t^\top + \frac{1}{10r_A}\mathbf{I}_{r_A}.
$$

Taking trace on both sides yields

$$
\mathsf{Tr}(\mathbf{I}_{r_A} - \mathbf{\Phi}_{t+1}\mathbf{\Phi}_{t+1}^\top) \leq \mathsf{Tr}(\mathbf{I}_{r_A} - \mathbf{\Phi}_t\mathbf{\Phi}_t^\top) + 0.1.
$$

$\square$

## E.8 Proof of Theorem 3.2

*Proof.* For the proof, we take $\eta = \frac{(r-r_A)^4}{975c_1^2\kappa^2m^2r^2r_A}$ and $\delta = \frac{c_4(r-r_A)^6}{\kappa^2m^3r^4r_A}$, where $c_4 = \mathcal{O}(\frac{1}{c_1^3})$. From Lemma D.5, we have that $\|\mathbf{\Theta}_t\| \leq 2$ holds for all $t \geq 0$ by mathematical induction. For later use, we define the following three terms in the same way as in Lemma D.4:

$$
\begin{aligned}
\beta_t &:= \sigma_1(\mathbf{I}_{r_A} - \mathbf{\Phi}_t\mathbf{\Phi}_t^\top) \leq 1, \\
\chi_t &:= (\|\mathbf{\Delta}_{t-1}\| + \|\mathbf{\Xi}_t\|)^2 + \sqrt{\mathsf{Tr}(\mathbf{I}_{r_A} - \mathbf{\Phi}_t\mathbf{\Phi}_t^\top)}(\|\mathbf{\Delta}_{t-1}\| + \|\mathbf{\Xi}_t\|), \\
\mathbf{H}_t &:= (\mathbf{I}_m - \mathbf{X}_t\mathbf{X}_t^\top)(\mathbf{A}\mathbf{X}_t\mathbf{X}_t^\top\mathbf{\Delta}_{t-1}\mathbf{X}_t + \mathbf{\Xi}_t\mathbf{X}_t\mathbf{\Theta}_t) \\
&\quad + \frac{1}{2}(\mathbf{X}_t\mathbf{X}_t^\top\mathbf{\Xi}_t\mathbf{X}_t\mathbf{\Theta}_t - \mathbf{X}_t\mathbf{\Theta}_t\mathbf{X}_t^\top\mathbf{\Xi}_t\mathbf{X}_t) \\
&\quad + \frac{1}{2}(\mathbf{X}_t\mathbf{X}_t^\top\mathbf{A}\mathbf{X}_t\mathbf{X}_t^\top\mathbf{\Delta}_{t-1}\mathbf{X}_t - \mathbf{X}_t\mathbf{X}_t^\top\mathbf{\Delta}_{t-1}\mathbf{X}_t\mathbf{X}_t^\top\mathbf{A}\mathbf{X}_t).
\end{aligned}
$$

Lemma D.3 with the RIP property of $\mathcal{M}(\cdot)$ implies that $\|\mathbf{\Delta}_{t-1}\|_\mathsf{F}, \|\mathbf{\Xi}_t\|_\mathsf{F} \leq 1$ for all $t \geq 1$. Thus, the assumptions of Lemma D.4 are met, guaranteeing that inequality (11) holds for all iterations. Building on inequality (11), we divide the convergence analysis into two phases.

**Phase I (Saddle phase).** $\mathsf{Tr}(\mathbf{I}_{r_A} - \mathbf{\Phi}_t\mathbf{\Phi}_t^\top) \geq 0.5$.

We assume for now that $\sigma_{r_A}^2(\mathbf{\Phi}_t) \geq (r-r_A)^2/(2c_1 mr)$ holds in Phase I, which will be proved later. Let the compact SVD of $\mathbf{\Phi}_t$ be $\mathbf{Q}_t\mathbf{\Lambda}_t\mathbf{P}_t^\top$, where $\mathbf{Q}_t \in \mathbb{R}^{r_A \times r_A}$, $\mathbf{\Lambda}_t \in \mathbb{R}^{r_A \times r_A}$, and $\mathbf{P}_t \in \mathbb{R}^{r \times r_A}$. We can simplify (11) as follows:

$$
\begin{aligned}
&\mathsf{Tr}(\mathbf{I}_{r_A} - \mathbf{\Phi}_{t+1}\mathbf{\Phi}_{t+1}^\top) - \mathsf{Tr}(\mathbf{I}_{r_A} - \mathbf{\Phi}_t\mathbf{\Phi}_t^\top) \\
&\leq \eta^2(1 + 16\chi_t)\mathsf{Tr}(\mathbf{\Phi}_t\mathbf{\Phi}_t^\top) - \frac{2\eta(1 - \eta^2 - 16\eta^2\chi_t)\sigma_{r_A}^2(\mathbf{\Phi}_t)}{\kappa^2}\mathsf{Tr}\big((\mathbf{I}_{r_A} - \mathbf{\Phi}_t\mathbf{\Phi}_t^\top)\mathbf{\Phi}_t\mathbf{\Phi}_t^\top\big) \\
&\quad + 2\eta\sqrt{r_A}(\|\mathbf{\Delta}_{t-1}\|_\mathsf{F} + 2\|\mathbf{\Xi}_t\|_\mathsf{F}) + 2\eta^2\sqrt{r_A}\|\mathbf{H}_t\|_\mathsf{F} \\
&\stackrel{(a)}{\leq} \eta^2(1 + 16\chi_t)r_A - \frac{2\eta(1 - \eta^2 - 16\eta^2\chi_t)\sigma_{r_A}^4(\mathbf{\Phi}_t)}{\kappa^2}\mathsf{Tr}(\mathbf{I}_{r_A} - \mathbf{\Phi}_t\mathbf{\Phi}_t^\top) \\
&\quad + 2\eta\sqrt{r_A}(\|\mathbf{\Delta}_{t-1}\|_\mathsf{F} + 2\|\mathbf{\Xi}_t\|_\mathsf{F}) + 2\eta^2\sqrt{r_A}\|\mathbf{H}_t\|_\mathsf{F} \\
&\stackrel{(b)}{\leq} \eta^2(1 + 16\chi_t)r_A - \frac{\eta(1 - \eta^2 - 16\eta^2\chi_t)(r - r_A)^4}{2c_1^2\kappa^2 m^2 r^2}\mathsf{Tr}(\mathbf{I}_{r_A} - \mathbf{\Phi}_t\mathbf{\Phi}_t^\top) \\
&\quad + 2\eta\sqrt{r_A}(\|\mathbf{\Delta}_{t-1}\|_\mathsf{F} + 2\|\mathbf{\Xi}_t\|_\mathsf{F}) + 2\eta^2\sqrt{r_A}\|\mathbf{H}_t\|_\mathsf{F},
\end{aligned}
\tag{29}
$$

where $(a)$ is by Lemma F.3 and $\mathsf{Tr}\big((\mathbf{I}_{r_A} - \mathbf{\Phi}_t\mathbf{\Phi}_t^\top)\mathbf{\Phi}_t\mathbf{\Phi}_t^\top\big) = \mathsf{Tr}\big((\mathbf{I}_{r_A} - \mathbf{\Lambda}_t^2)\mathbf{\Lambda}_t^2\big)$; and $(b)$ is from our assumption that $\sigma_{r_A}^2(\mathbf{\Phi}_t) \geq (r - r_A)^2/(2c_1 mr)$.

Using Lemma F.11 and the RIP property of $\mathcal{M}(\cdot)$, we can control the quantities of the two error terms. In particular, following inequalities imply that both $\|\mathbf{\Delta}_{t-1}\|_\mathsf{F}$ and $\|\mathbf{\Xi}_t\|_\mathsf{F}$ are uniformly bounded by a constant that depends only on $m, r$ and $r_A$ but is independent of $t$.

Expanding the expression of $\mathbf{\Delta}_{t-1}$ and applying Lemma F.11, we have that

$$
\begin{aligned}
\|\mathbf{\Delta}_{t-1}\|_\mathsf{F} &\leq \sqrt{m}\|(\mathcal{M}^*\mathcal{M} - \mathcal{I})(\mathbf{X}_t\mathbf{\Theta}_{t-1}\mathbf{X}_t^\top - \mathbf{A})\| \\
&\leq \frac{c_4(r - r_A)^6}{\kappa^2 m^{5/2} r^4 r_A}\|\mathbf{X}_t\mathbf{\Theta}_{t-1}\mathbf{X}_t^\top - \mathbf{A}\|_\mathsf{F} \\
&\leq \frac{c_4(r - r_A)^6}{\kappa^2 m^{5/2} r^4 r_A}(2\sqrt{r} + \sqrt{r_A}) \\
&\leq \frac{3c_4(r - r_A)^6}{\kappa^2 m^{5/2} r^{7/2} r_A} \\
&\stackrel{(c)}{\leq} \min\Big\{\frac{(r - r_A)^4}{48c_1^2\kappa^2 m^2 r^2 r_A}, \frac{1}{48\sqrt{r_A}}\Big\}.
\end{aligned}
\tag{30}
$$

Applying the same reasoning to $\mathbf{\Xi}_t$, it follows that

$$
\begin{aligned}
\|\mathbf{\Xi}_t\|_\mathsf{F} &\leq \sqrt{m}\|(\mathcal{M}^*\mathcal{M} - \mathcal{I})(\mathbf{X}_t\mathbf{\Theta}_t\mathbf{X}_t^\top - \mathbf{A})\| \\
&\leq \frac{c_4(r - r_A)^6}{\kappa^2 m^{5/2} r^4 r_A}\|\mathbf{X}_t\mathbf{\Theta}_t\mathbf{X}_t^\top - \mathbf{A}\|_\mathsf{F} \\
&\leq \frac{c_4(r - r_A)^6}{\kappa^2 m^{5/2} r^4 r_A}(2\sqrt{r} + \sqrt{r_A}) \\
&\leq \frac{3c_4(r - r_A)^6}{\kappa^2 m^{5/2} r^{7/2} r_A} \\
&\stackrel{(c)}{\leq} \min\Big\{\frac{(r - r_A)^4}{48c_1^2\kappa^2 m^2 r^2 r_A}, \frac{1}{48\sqrt{r_A}}\Big\}.
\end{aligned}
\tag{31}
$$

Here, $(c)$ is from $c_4 = \mathcal{O}(\frac{1}{c_1^3})$, $c_1 > 1$ and $r - r_A \le r \le m$. Since $\mathsf{Tr}(\mathbf{I}_{r_A} - \boldsymbol{\Phi}_t \boldsymbol{\Phi}_t^\top) \le r_A$, together with (30) and (31), we can upper bound $\chi_t$ as follows:

$$
\begin{aligned}
\chi_t &= (\|\boldsymbol{\Delta}_{t-1}\| + \|\boldsymbol{\Xi}_t\|)^2 + \sqrt{\mathsf{Tr}(\mathbf{I}_{r_A} - \boldsymbol{\Phi}_t \boldsymbol{\Phi}_t^\top)}(\|\boldsymbol{\Delta}_{t-1}\| + \|\boldsymbol{\Xi}_t\|) \\
&\le (\|\boldsymbol{\Delta}_{t-1}\|_{\mathsf{F}} + \|\boldsymbol{\Xi}_t\|_{\mathsf{F}})^2 + \sqrt{r_A}(\|\boldsymbol{\Delta}_{t-1}\|_{\mathsf{F}} + \|\boldsymbol{\Xi}_t\|_{\mathsf{F}}) \\
&\le (\frac{1}{48} + \frac{1}{48})^2 + \sqrt{r_A}(\frac{1}{48\sqrt{r_A}} + \frac{1}{48\sqrt{r_A}}) \\
&\le \frac{1}{16}.
\end{aligned}
$$

From inequalities (12), (30), and (31), we obtain the following upper bound on $\|\mathbf{H}_t\|_{\mathsf{F}}$.

$$
\begin{aligned}
\|\mathbf{H}_t\|_{\mathsf{F}} &\le 2\|\boldsymbol{\Delta}_{t-1}\|_{\mathsf{F}} + 4\|\boldsymbol{\Xi}_t\|_{\mathsf{F}} \\
&\le 2 \times \frac{1}{48\sqrt{r_A}} + 4 \times \frac{1}{48\sqrt{r_A}} \\
&\le \frac{1}{2\sqrt{r_A}}.
\end{aligned}
$$

With these upper bounds, inequality (29) can be simplified as follows:

$$
\begin{aligned}
&\mathsf{Tr}(\mathbf{I}_{r_A} - \boldsymbol{\Phi}_{t+1} \boldsymbol{\Phi}_{t+1}^\top) - \mathsf{Tr}(\mathbf{I}_{r_A} - \boldsymbol{\Phi}_t \boldsymbol{\Phi}_t^\top) \\
&\le 2\eta^2 r_A - \frac{\eta(1 - 2\eta^2)(r - r_A)^4}{2c_1^2 \kappa^2 m^2 r^2} \mathsf{Tr}(\mathbf{I}_{r_A} - \boldsymbol{\Phi}_t \boldsymbol{\Phi}_t^\top) + \frac{\eta(r - r_A)^4}{8c_1^2 \kappa^2 m^2 r^2} + \eta^2 \\
&\overset{(d)}{\le} \Big( -\frac{\eta(r - r_A)^4}{2c_1^2 \kappa^2 m^2 r^2} + \frac{\eta(r - r_A)^4}{4c_1^2 \kappa^2 m^2 r^2} + 6\eta^2 r_A + \frac{\eta^3(r - r_A)^4}{c_1^2 \kappa^2 m^2 r^2} \Big) \mathsf{Tr}(\mathbf{I}_{r_A} - \boldsymbol{\Phi}_t \boldsymbol{\Phi}_t^\top) \\
&= \Big( -\frac{\eta(r - r_A)^4}{4c_1^2 \kappa^2 m^2 r^2} + 6\eta^2 r_A + \frac{\eta^3(r - r_A)^4}{c_1^2 \kappa^2 m^2 r^2} \Big) \mathsf{Tr}(\mathbf{I}_{r_A} - \boldsymbol{\Phi}_t \boldsymbol{\Phi}_t^\top) \\
&\overset{(e)}{\le} \frac{1}{2} \Big( -\frac{\eta(r - r_A)^4}{4c_1^2 \kappa^2 m^2 r^2} + 6\eta^2 r_A + \frac{\eta^3(r - r_A)^4}{c_1^2 \kappa^2 m^2 r^2} \Big),
\end{aligned}
$$

where $(d)$ is by $\mathsf{Tr}(\mathbf{I}_{r_A} - \boldsymbol{\Phi}_t \boldsymbol{\Phi}_t^\top) \ge 0.5$; and $(e)$ holds if the expression in bracket is less than zero. Recall that $\eta = \frac{(r - r_A)^4}{975 c_1^2 \kappa^2 m^2 r^2 r_A}$. The summation of the terms in bracket is negative, which implies that at each step, $\mathsf{Tr}(\mathbf{I}_{r_A} - \boldsymbol{\Phi}_t \boldsymbol{\Phi}_t^\top)$ decreases at least by $\Delta := \frac{(r - r_A)^8}{7000 c_1^4 \kappa^4 m^4 r^4 r_A}$. Consequently, after at most $(r_A - 0.5)/\Delta \le \frac{7000 c_1^4 \kappa^4 m^4 r^4 r_A^2}{(r - r_A)^8}$ iterations, RGD leaves Phase I.

Let $c_2 := \frac{1}{7000 c_1^4} \in (0, 1)$. Denote $t_0 \ge 1$ as the last iteration in this phase. The analysis above implies that $\mathsf{Tr}(\mathbf{I}_{r_A} - \boldsymbol{\Phi}_t \boldsymbol{\Phi}_t^\top) \le r_A - \frac{c_2(r - r_A)^8 t}{\kappa^4 m^4 r^4 r_A}$ for all $1 \le t \le t_0$ and $t_0 \le \frac{7000 c_1^4 \kappa^4 m^4 r^4 r_A^2}{(r - r_A)^8}$.

From Lemma D.2 and inequality (30), we obtain the following bound for $1 \le t \le t_0$:

$$
\begin{aligned}
\|\mathbf{X}_t \boldsymbol{\Theta}_t \mathbf{X}_t^\top - \mathbf{A}\|_{\mathsf{F}} &\le 2\sqrt{\mathsf{Tr}(\mathbf{I}_{r_A} - \boldsymbol{\Phi}_t \boldsymbol{\Phi}_t^\top)} + \|\boldsymbol{\Delta}_{t-1}\|_{\mathsf{F}} \\
&\le 2\sqrt{r_A - \frac{c_2(r - r_A)^8 t}{\kappa^4 m^4 r^4 r_A}} + \|\boldsymbol{\Delta}_{t-1}\|_{\mathsf{F}} \\
&\le 2\sqrt{r_A - \frac{c_2(r - r_A)^8 t}{\kappa^4 m^4 r^4 r_A}} + 1.
\end{aligned}
$$

We now prove that $\sigma_{r_A}^2(\boldsymbol{\Phi}_t) \ge (r - r_A)^2/(2c_1 mr)$ holds in Phase I. By Lemma D.1, it holds w.h.p.,

$$
\sigma_{r_A}^2(\boldsymbol{\Phi}_0) = \sigma_{r_A}^2(\mathbf{U}^\top \mathbf{X}_0) \ge \frac{(r - r_A)^2}{c_1 mr}.
$$

Moreover, by Lemma F.2 and the assumption $r_A \leq \frac{m}{2}$, it follows that

$$\sigma_{r_A}^2(\boldsymbol{\Psi}_0) = 1 - \sigma_{r_A}^2(\boldsymbol{\Phi}_0) \leq 1 - \frac{(r - r_A)^2}{c_1 m r}.$$

Since $\eta = \frac{(r - r_A)^4}{975 c_1^2 \kappa^2 m^2 r^2 r_A}$ and $\delta = \frac{c_4(r - r_A)^6}{\kappa^2 m^3 r^4 r_A}$, we can deduce that

$$\eta(\sqrt{r_A} + 2\sqrt{r})\delta \leq \frac{c_4(r - r_A)^{10}}{325 c_1^2 \kappa^4 m^5 r^5 r_A^2}.$$

From Lemma D.6 and the upper bound on $\eta(\sqrt{r_A} + 2\sqrt{r})\delta$, we obtain the following inequality

$$\boldsymbol{\Psi}_1 \boldsymbol{\Psi}_1^\top \preceq (1 + \frac{4(r - r_A)^4}{975 c_1^2 \kappa^2 m^2 r^2 r_A})^2 \boldsymbol{\Psi}_0 \boldsymbol{\Psi}_0^\top + \frac{4 c_4(r - r_A)^{10}}{65 c_1^2 \kappa^4 m^5 r^5 r_A^2} \mathbf{I}_{m - r_A}.$$

Using Weyl's inequality and $c_4 = \mathcal{O}(\frac{1}{c_1^3})$, we have the following upper bound on $\sigma_{r_A}^2(\boldsymbol{\Psi}_1)$

$$\sigma_{r_A}^2(\boldsymbol{\Psi}_1) \leq (1 + \frac{4(r - r_A)^4}{975 c_1^2 \kappa^2 m^2 r^2 r_A})^2 \sigma_{r_A}^2(\boldsymbol{\Psi}_0) + \frac{4 c_4(r - r_A)^{10}}{65 c_1^2 \kappa^4 m^5 r^5 r_A^2}$$

$$\leq (1 + \frac{4(r - r_A)^4}{975 c_1^2 \kappa^2 m^2 r^2 r_A})^2 (1 - \frac{(r - r_A)^2}{c_1 m r}) + \frac{4 c_4(r - r_A)^{10}}{65 c_1^2 \kappa^4 m^5 r^5 r_A^2}$$

$$\overset{(f)}{\leq} 1 + \frac{16(r - r_A)^4}{195 c_1^2 \kappa^2 m^2 r^2 r_A} - \frac{(r - r_A)^2}{c_1 m r}$$

$$\overset{(f)}{\leq} 1 - \frac{2(r - r_A)^2}{3 c_1 m r},$$

where $(f)$ is by $r - r_A \leq r \leq m$ and $c_1, \kappa, r_A \geq 1$. Applying Lemma D.6 with the upper bound on $\eta(\sqrt{r_A} + 2\sqrt{r})\delta$, we obtain

$$\boldsymbol{\Psi}_{t+1} \boldsymbol{\Psi}_{t+1}^\top \preceq \left(1 + \frac{c_4(r - r_A)^{10}}{40 c_1^2 \kappa^4 m^5 r^5 r_A^2}\right)^2 \boldsymbol{\Psi}_t \boldsymbol{\Psi}_t^\top + \frac{c_4(r - r_A)^{10}}{40 c_1^2 \kappa^4 m^5 r^5 r_A^2} \mathbf{I}_{m - r_A}, \quad t \geq 1.$$

Using Weyl's inequality, we have the following relationship between $\sigma_{r_A}^2(\boldsymbol{\Psi}_{t+1})$ and $\sigma_{r_A}^2(\boldsymbol{\Psi}_t)$

$$\sigma_{r_A}^2(\boldsymbol{\Psi}_{t+1}) = \sigma_{r_A}(\boldsymbol{\Psi}_{t+1} \boldsymbol{\Psi}_{t+1}^\top) \leq \left(1 + \frac{c_4(r - r_A)^{10}}{40 c_1^2 \kappa^4 m^5 r^5 r_A^2}\right)^2 \sigma_{r_A}(\boldsymbol{\Psi}_t \boldsymbol{\Psi}_t^\top) + \frac{c_4(r - r_A)^{10}}{40 c_1^2 \kappa^4 m^5 r^5 r_A^2}$$

$$= \left(1 + \frac{c_4(r - r_A)^{10}}{40 c_1^2 \kappa^4 m^5 r^5 r_A^2}\right)^2 \sigma_{r_A}^2(\boldsymbol{\Psi}_t) + \frac{c_4(r - r_A)^{10}}{40 c_1^2 \kappa^4 m^5 r^5 r_A^2}.$$

Denote $\zeta := \frac{c_4(r - r_A)^{10}}{40 c_1^2 \kappa^4 m^5 r^5 r_A^2}$. By iterating the recursive inequality, the following upper bound holds

$$\sigma_{r_A}^2(\boldsymbol{\Psi}_t) \leq \left(1 + \zeta\right)^{2(t-1)} \sigma_{r_A}^2(\boldsymbol{\Psi}_1) + \zeta \sum_{i=0}^{t-2} \left(1 + \zeta\right)^{2i}$$

$$\leq \left(1 + \zeta\right)^{2t} \sigma_{r_A}^2(\boldsymbol{\Psi}_1) + \zeta \sum_{i=0}^{t-1} \left(1 + \zeta\right)^{2i}$$

$$= \left(1 + \zeta\right)^{2t} \sigma_{r_A}^2(\boldsymbol{\Psi}_1) + \zeta \left[\left(1 + \zeta\right)^{2t} - 1\right] \Big/ \left[\left(1 + \zeta\right)^2 - 1\right]$$

$$\leq \left(1 + \zeta\right)^{2t} \sigma_{r_A}^2(\boldsymbol{\Psi}_1) + \left(1 + \zeta\right)^{2t} - 1,$$

for all $1 \leq t \leq \frac{7000 c_1^4 \kappa^4 m^4 r^4 r_A^2}{(r - r_A)^8}$. Invoking Lemma F.13 and noting that $\zeta \leq \frac{1}{2t}$, which is ensured by $c_4 = \mathcal{O}(\frac{1}{c_1^3})$, we obtain

$$\sigma_{r_A}^2(\boldsymbol{\Psi}_t) \leq \left(1 + 6t\zeta\right) \sigma_{r_A}^2(\boldsymbol{\Psi}_1) + 6t\zeta$$

$$\overset{(g)}{\leq} \sigma_{r_A}^2(\boldsymbol{\Psi}_1) + \frac{2100 c_1^2 c_4(r - r_A)^2}{m r}$$

$$\leq 1 - \frac{2(r - r_A)^2}{3 c_1 m r} + \frac{2100 c_1^2 c_4(r - r_A)^2}{m r}$$

$$\overset{(h)}{\leq} 1 - \frac{(r - r_A)^2}{2 c_1 m r},$$

where $(g)$ is by $t\zeta \leq \frac{175c_1^2 c_4 (r-r_A)^2}{mr}$ and $\sigma_{r_A}^2(\mathbf{\Psi}_1) \leq 1$; and $(h)$ holds by $c_4 = \mathcal{O}(\frac{1}{c_1^3})$. By Lemma F.2 and the assumption $r_A \leq \frac{m}{2}$, it can be seen that $\sigma_{r_A}^2(\mathbf{\Phi}_t) = 1 - \sigma_{r_A}^2(\mathbf{\Psi}_t) \geq \frac{(r-r_A)^2}{2c_1 mr}$ holds for all $t \leq t_0 \leq \frac{7000c_1^2 \kappa^4 m^4 r^4 r_A^2}{(r-r_A)^8}$, i.e., throughout Phase I.

**Phase II (Linearly convergent phase).** $\mathsf{Tr}(\mathbf{I}_{r_A} - \mathbf{\Phi}_t \mathbf{\Phi}_t^\top) < 0.5$.

This corresponds to a near-optimal regime. An immediate implication of this phase is that $\mathsf{Tr}(\mathbf{\Phi}_t \mathbf{\Phi}_t^\top) \geq r_A - 0.5 \geq r_A - 0.6$. Recall that $t_0 \geq 1$ is the last iteration in the first phase. We assume that $\mathsf{Tr}(\mathbf{\Phi}_t \mathbf{\Phi}_t^\top) \geq r_A - 0.6$ for all $t \geq t_0 + 1$, and we will prove this later.

Given that the singular values of $\mathbf{\Phi}_t \mathbf{\Phi}_t^\top$ lie in $[0, 1]$, we have

$$0.4 \leq \sigma_{r_A}^2(\mathbf{\Phi}_t) = \sigma_{r_A}(\mathbf{\Phi}_t \mathbf{\Phi}_t^\top) \leq \sigma_1^2(\mathbf{\Phi}_t) \leq 1.$$

Moreover, since $\beta_t = \sigma_1(\mathbf{I}_{r_A} - \mathbf{\Phi}_t \mathbf{\Phi}_t^\top) \leq \mathsf{Tr}(\mathbf{I}_{r_A} - \mathbf{\Phi}_t \mathbf{\Phi}_t^\top)$ and $\beta_t \leq 1$, it follows that

$$\beta_t \mathsf{Tr}(\mathbf{\Phi}_t \mathbf{\Phi}_t^\top) \leq r_A \mathsf{Tr}(\mathbf{I}_{r_A} - \mathbf{\Phi}_t \mathbf{\Phi}_t^\top), \quad \chi_t \mathsf{Tr}(\mathbf{\Phi}_t \mathbf{\Phi}_t^\top) \leq r_A \chi_t.$$

In addition, it can be derived that

$$\frac{4}{25} \mathsf{Tr}(\mathbf{I}_{r_A} - \mathbf{\Phi}_t \mathbf{\Phi}_t^\top) \leq \sigma_{r_A}^2(\mathbf{\Phi}_t) \mathsf{Tr}\big((\mathbf{I}_{r_A} - \mathbf{\Phi}_t \mathbf{\Phi}_t^\top)\mathbf{\Phi}_t \mathbf{\Phi}_t^\top\big)$$
$$\leq \sigma_1^2(\mathbf{\Phi}_t) \mathsf{Tr}(\mathbf{I}_{r_A} - \mathbf{\Phi}_t \mathbf{\Phi}_t^\top)$$
$$\leq \mathsf{Tr}(\mathbf{I}_{r_A} - \mathbf{\Phi}_t \mathbf{\Phi}_t^\top).$$

With the inequalities above, we can simplify (11) as follows:

$$\mathsf{Tr}(\mathbf{I}_{r_A} - \mathbf{\Phi}_{t+1} \mathbf{\Phi}_{t+1}^\top) \leq (1 - \frac{8\eta}{25\kappa^2} + \eta^2 r_A + \frac{2\eta^3}{\kappa^2} + \frac{32\eta^3}{\kappa^2}\chi_t)\mathsf{Tr}(\mathbf{I}_{r_A} - \mathbf{\Phi}_t \mathbf{\Phi}_t^\top) \tag{32}$$
$$+ 16\eta^2 r_A \chi_t + 2\eta \sqrt{\mathsf{Tr}(\mathbf{I}_{r_A} - \mathbf{\Phi}_t \mathbf{\Phi}_t^\top)}(\|\mathbf{\Delta}_{t-1}\|_{\mathsf{F}} + 2\|\mathbf{\Xi}_t\|_{\mathsf{F}} + \eta\|\mathbf{H}_t\|_{\mathsf{F}}).$$

Recall that $\chi_t = (\|\mathbf{\Delta}_{t-1}\| + \|\mathbf{\Xi}_t\|)^2 + \sqrt{\mathsf{Tr}(\mathbf{I}_{r_A} - \mathbf{\Phi}_t \mathbf{\Phi}_t^\top)}(\|\mathbf{\Delta}_{t-1}\| + \|\mathbf{\Xi}_t\|)$ and $\|\mathbf{H}_t\|_{\mathsf{F}} \leq 2\|\mathbf{\Delta}_{t-1}\|_{\mathsf{F}} + 4\|\mathbf{\Xi}_t\|_{\mathsf{F}}$. Since $\|\mathbf{\Delta}_{t-1}\| \leq 1$ and $\|\mathbf{\Xi}_t\| \leq 1$, (32) can be written as

$$\mathsf{Tr}(\mathbf{I}_{r_A} - \mathbf{\Phi}_{t+1} \mathbf{\Phi}_{t+1}^\top) \leq (1 - \frac{8\eta}{25\kappa^2} + \eta^2 r_A + \frac{194\eta^3}{\kappa^2})\mathsf{Tr}(\mathbf{I}_{r_A} - \mathbf{\Phi}_t \mathbf{\Phi}_t^\top)$$
$$+ 16\eta^2 r_A (\|\mathbf{\Delta}_{t-1}\| + \|\mathbf{\Xi}_t\|)^2$$
$$+ \sqrt{\mathsf{Tr}(\mathbf{I}_{r_A} - \mathbf{\Phi}_t \mathbf{\Phi}_t^\top)} \cdot \big(16\eta^2 r_A (\|\mathbf{\Delta}_{t-1}\| + \|\mathbf{\Xi}_t\|)\big)$$
$$+ \sqrt{\mathsf{Tr}(\mathbf{I}_{r_A} - \mathbf{\Phi}_t \mathbf{\Phi}_t^\top)} \cdot (4\eta^2 + 2\eta)(\|\mathbf{\Delta}_{t-1}\|_{\mathsf{F}} + 2\|\mathbf{\Xi}_t\|_{\mathsf{F}}).$$

Substituting $\eta = \frac{(r-r_A)^4}{975c_1^2 \kappa^2 m^2 r^2 r_A}$ into the inequality above, it follows that

$$\mathsf{Tr}(\mathbf{I}_{r_A} - \mathbf{\Phi}_{t+1} \mathbf{\Phi}_{t+1}^\top) \leq q\mathsf{Tr}(\mathbf{I}_{r_A} - \mathbf{\Phi}_t \mathbf{\Phi}_t^\top) + \frac{1}{\kappa^4 r_A}(\|\mathbf{\Delta}_{t-1}\| + \|\mathbf{\Xi}_t\|)^2$$
$$+ \sqrt{\mathsf{Tr}(\mathbf{I}_{r_A} - \mathbf{\Phi}_t \mathbf{\Phi}_t^\top)} \cdot \frac{1}{\kappa^4 r_A}(\|\mathbf{\Delta}_{t-1}\| + \|\mathbf{\Xi}_t\|) \tag{33}$$
$$+ \sqrt{\mathsf{Tr}(\mathbf{I}_{r_A} - \mathbf{\Phi}_t \mathbf{\Phi}_t^\top)} \cdot (\frac{1}{\kappa^4 r_A^2} + \frac{1}{\kappa^2 r_A})(\|\mathbf{\Delta}_{t-1}\|_{\mathsf{F}} + 2\|\mathbf{\Xi}_t\|_{\mathsf{F}}),$$

where $q := 1 - \frac{(r-r_A)^4}{8125c_1^2 \kappa^4 m^2 r^2 r_A}$ is a constant in $(\frac{1}{2}, 1)$.

From Lemma D.3 and the RIP property of $\mathcal{M}(\cdot)$ with $\delta = \frac{c_4(r-r_A)^6}{\kappa^2 m^3 r^4 r_A}$, it guarantees that

$$
\begin{aligned}
\|\boldsymbol{\Delta}_{t-1}\| \leq \|\boldsymbol{\Delta}_{t-1}\|_{\mathsf{F}} &\leq \frac{c_4(r-r_A)^6}{\kappa^2 m^{5/2} r^{7/2} r_A} \|\mathbf{X}_{t-1}\boldsymbol{\Theta}_{t-1}\mathbf{X}_{t-1}^\top - \mathbf{A}\|_{\mathsf{F}} \\
&\leq \frac{c_4(r-r_A)^4}{\kappa^2 m^2 r^2} \|\mathbf{X}_{t-1}\boldsymbol{\Theta}_{t-1}\mathbf{X}_{t-1}^\top - \mathbf{A}\|_{\mathsf{F}}, \\
\|\boldsymbol{\Xi}_t\| \leq \|\boldsymbol{\Xi}_t\|_{\mathsf{F}} &\leq \frac{c_4(r-r_A)^6}{\kappa^2 m^{5/2} r^{7/2} r_A} \|\mathbf{X}_t \boldsymbol{\Theta}_t \mathbf{X}_t^\top - \mathbf{A}\|_{\mathsf{F}} \\
&\leq \frac{c_4(r-r_A)^4}{\kappa^2 m^2 r^2} \|\mathbf{X}_t \boldsymbol{\Theta}_t \mathbf{X}_t^\top - \mathbf{A}\|_{\mathsf{F}}.
\end{aligned}
$$

Together with $c_4 = \mathcal{O}(\frac{1}{c_1^3})$, we can rewrite inequality (33) as

$$
\begin{aligned}
\mathsf{Tr}&(\mathbf{I}_{r_A} - \boldsymbol{\Phi}_{t+1}\boldsymbol{\Phi}_{t+1}^\top) \\
&\leq q\mathsf{Tr}(\mathbf{I}_{r_A} - \boldsymbol{\Phi}_t \boldsymbol{\Phi}_t^\top) + \frac{1-q}{180}\Big(\|\mathbf{X}_{t-1}\boldsymbol{\Theta}_{t-1}\mathbf{X}_{t-1}^\top - \mathbf{A}\|_{\mathsf{F}}^2 + \|\mathbf{X}_t \boldsymbol{\Theta}_t \mathbf{X}_t^\top - \mathbf{A}\|_{\mathsf{F}}^2 \\
&\quad + 2\|\mathbf{X}_{t-1}\boldsymbol{\Theta}_{t-1}\mathbf{X}_{t-1}^\top - \mathbf{A}\|_{\mathsf{F}} \|\mathbf{X}_t \boldsymbol{\Theta}_t \mathbf{X}_t^\top - \mathbf{A}\|_{\mathsf{F}} \\
&\quad + \sqrt{\mathsf{Tr}(\mathbf{I}_{r_A} - \boldsymbol{\Phi}_t \boldsymbol{\Phi}_t^\top)}\big(\|\mathbf{X}_{t-1}\boldsymbol{\Theta}_{t-1}\mathbf{X}_{t-1}^\top - \mathbf{A}\|_{\mathsf{F}} + \|\mathbf{X}_t \boldsymbol{\Theta}_t \mathbf{X}_t^\top - \mathbf{A}\|_{\mathsf{F}}\big)\Big).
\end{aligned}
\tag{34}
$$

Denote $b_t := \mathsf{Tr}(\mathbf{I}_{r_A} - \boldsymbol{\Phi}_t \boldsymbol{\Phi}_t^\top)$, $a_t := \|\mathbf{X}_t \boldsymbol{\Theta}_t \mathbf{X}_t^\top - \mathbf{A}\|_{\mathsf{F}}$. Inequality (34) can be expressed as

$$
b_{t+1} \leq qb_t + \frac{1-q}{180}\big(a_{t-1}^2 + a_t^2 + 2a_{t-1}a_t + \sqrt{b_t}(a_{t-1} + a_t)\big).
\tag{35}
$$

Combining Lemma D.2, Lemma D.3 and the RIP property of $\mathcal{M}(\cdot)$ with $\delta = \frac{c_4(r-r_A)^6}{\kappa^2 m^3 r^4 r_A}$, we obtain

$$
a_t \leq 2\sqrt{b_t} + \frac{1}{6}a_{t-1}.
\tag{36}
$$

Since $t_0 + 1$ is the first iteration in Phase II, we have $\mathsf{Tr}(\mathbf{I}_{r_A} - \boldsymbol{\Phi}_{t_0+1}\boldsymbol{\Phi}_{t_0+1}^\top) \leq 0.5$. From Lemma D.7, it follows that $\mathsf{Tr}(\mathbf{I}_{r_A} - \boldsymbol{\Phi}_{t_0+2}\boldsymbol{\Phi}_{t_0+2}^\top) \leq 0.6$. Hence, $b_{t_0+1}, b_{t_0+2} \in [0, 0.6]$.

From Lemma F.14, to establish the linear convergence rate of $a_t$, it suffices to analyze the following equality system of $\{\tilde{b}_t\}_{t=t_0+1}^\infty$ and $\{\tilde{a}_t\}_{t=t_0+1}^\infty$:

$$
\begin{aligned}
\tilde{b}_{t+1} &= q\tilde{b}_t + \frac{1-q}{180}\big(\tilde{a}_{t-1}^2 + \tilde{a}_t^2 + 2\tilde{a}_{t-1}\tilde{a}_t + \sqrt{\tilde{b}_t}(\tilde{a}_{t-1} + \tilde{a}_t)\big), \\
\tilde{a}_t &= 2\sqrt{\tilde{b}_t} + \frac{1}{6}\tilde{a}_{t-1}, \quad t = t_0 + 2, t_0 + 3, \dots, \\
\tilde{a}_{t_0+1} &= a_{t_0+1}, \tilde{b}_{t_0+1} = 0.6, \tilde{b}_{t_0+2} = 0.6.
\end{aligned}
$$

By Lemma D.2 and the RIP property of $\mathcal{M}(\cdot)$ with $\delta = \frac{c_4(r-r_A)^6}{\kappa^2 m^3 r^4 r_A}$, we derive

$$
\tilde{a}_{t_0+1} = a_{t_0+1} \leq 2\sqrt{b_{t_0+1}} + \|\boldsymbol{\Delta}_{t_0}\|_{\mathsf{F}} \overset{(i)}{\leq} 2\sqrt{\tilde{b}_{t_0+1}} + \frac{1}{48\sqrt{r_A}} \leq 3\sqrt{\tilde{b}_{t_0+1}} \leq \frac{3\sqrt{2}}{\sqrt{1+q}}\sqrt{\tilde{b}_{t_0+2}},
$$

where $(i)$ is from inequality (30). From the update of $\tilde{a}_t$ at $t = t_0 + 2$, it follows that

$$
\tilde{a}_{t_0+2} = 2\sqrt{\tilde{b}_{t_0+2}} + \frac{1}{6}\tilde{a}_{t_0+1} \leq 2\sqrt{\tilde{b}_{t_0+2}} + \frac{1}{3}\sqrt{\tilde{b}_{t_0+1}} + \frac{1}{288\sqrt{r_A}} \leq 3\sqrt{\tilde{b}_{t_0+2}}.
$$

Therefore, applying Lemma F.14 and Lemma F.15, we arrive at

$$
a_{t_0+1+t} \leq \tilde{a}_{t_0+1+t} \leq 3\sqrt{\tilde{b}_{t_0+1}}\left(\frac{1+q}{2}\right)^{t/2} \leq 3\left(1 - \frac{1-q}{4}\right)^{t+1} = 3\left(1 - \frac{c_3(r-r_A)^4}{\kappa^4 m^2 r^2 r_A}\right)^{t+1},
$$

for all $t \geq 0$, with $c_3 := \frac{1}{32500 c_1^2} \in (0,1)$. This establishes the linear convergence rate of $a_t$.

We now prove that $\mathsf{Tr}(\mathbf{\Phi}_t\mathbf{\Phi}_t^\top) \geq r_A - 0.6$ for all $t \geq t_0 + 1$. This amounts to proving that $\mathsf{Tr}(\mathbf{I}_{r_A} - \mathbf{\Phi}_t\mathbf{\Phi}_t^\top) = b_t \leq 0.6$ for all $t \geq t_0 + 1$.

Since $b_{t_0+2} \leq 0.6$, inequality (35) holds for $t = t_0 + 2$. Hence,

$$
\begin{aligned}
b_{t_0+3} &\leq q b_{t_0+2} + \frac{1-q}{180}\left(a_{t_0+1}^2 + a_{t_0+2}^2 + 2a_{t_0+1}a_{t_0+2} + \sqrt{b_{t_0+2}}(a_{t_0+1} + a_{t_0+2})\right) \\
&\overset{(j)}{\leq} 0.6q + \frac{1-q}{180}(9 \times 0.6 + 9 \times 0.6 + 18 \times 0.6 + 6 \times 0.6) \\
&\leq \frac{1+q}{2} \times 0.6 \\
&\leq 0.6,
\end{aligned}
$$

where $(j)$ is from the fact that $a_{t_0+1} \leq 3\sqrt{\tilde{b}_{t_0+1}} = 3\sqrt{0.6}$, $a_{t_0+2} \leq \tilde{a}_{t_0+2} \leq 3\sqrt{\tilde{b}_{t_0+2}} = 3\sqrt{0.6}$. Therefore, inequality (35) holds for $t = t_0 + 3$. From inequality (36), we have $a_{t_0+3} \leq 2\sqrt{b_{t_0+3}} + \frac{1}{6}a_{t_0+2} \leq 3\sqrt{0.6}$. By recursion, it follows that $\mathsf{Tr}(\mathbf{I}_{r_A} - \mathbf{\Phi}_t\mathbf{\Phi}_t^\top) = b_t \leq 0.6$ for all $t \geq t_0 + 1$.

To conclude, by choosing stepsizes $\eta = \frac{(r-r_A)^4}{975c_1^2\kappa^2 m^2 r^2 r_A}$ and $\mu = 2$, we have that $\|\mathbf{X}_t\mathbf{\Theta}_t\mathbf{X}_t^\top - \mathbf{A}\|_\mathsf{F} \leq 3\left(1 - \frac{c_3(r-r_A)^4}{\kappa^4 m^2 r^2 r_A}\right)^{t-t_0}$ for all $t \geq t_0 + 1$, with high probability over the initialization. $\qquad\square$

### E.9 PROOF OF LEMMA 4.1

*Proof.* Let $\mathbf{U}\mathbf{\Sigma}\mathbf{U}^\top$ be the compact SVD of $\mathbf{A}$, where $\mathbf{U} = [\mathbf{u}_1, \mathbf{u}_2, \ldots, \mathbf{u}_{r_A}]$ and $\mathbf{\Sigma} = \mathrm{diag}(\lambda_1, \lambda_2, \ldots, \lambda_{r_A})$, with $\lambda_1 \geq \lambda_2 \geq \cdots \geq \lambda_{r_A} > 0$. Here, $\mathrm{diag}(\lambda_1, \lambda_2, \ldots, \lambda_{r_A})$ denotes the diagonal matrix whose diagonal entries are $\lambda_1, \lambda_2, \ldots, \lambda_{r_A}$.

We first consider $\rho \geq 1$. From the Eckart–Young–Mirsky theorem, we have that the best rank $- \rho$ approximation of $\mathbf{A}$ under the Frobenius norm is $\mathbf{A}_\rho = \mathbf{U}_1\mathbf{\Sigma}_1\mathbf{U}_1^\top$, where $\mathbf{U}_1 = [\mathbf{u}_1, \mathbf{u}_2, \ldots, \mathbf{u}_\rho]$ and $\mathbf{\Sigma} = \mathrm{diag}(\lambda_1, \lambda_2, \ldots, \lambda_\rho)$, without considering the ordering of the eigenvalues.

We begin by analyzing the form of $\mathbf{X}$ and $\mathbf{\Theta}$. Since $\mathrm{rank}(\mathbf{A}_\rho) = \mathrm{rank}(\mathbf{U}_1) = \rho$ and $\mathrm{range}(\mathbf{A}_\rho) \subseteq \mathrm{range}(\mathbf{U}_1)$, it follows that $\mathrm{range}(\mathbf{A}_\rho) = \mathrm{range}(\mathbf{U}_1)$. Together with $\mathrm{range}(\mathbf{A}_\rho) = \mathrm{range}(\mathbf{X}\mathbf{\Theta}\mathbf{X}^\top) \subseteq \mathrm{range}(\mathbf{X})$, we can obtain $\mathrm{range}(\mathbf{U}_1) \subseteq \mathrm{range}(\mathbf{X})$. Therefore, there exsits a matrix $\mathbf{Q} \in \mathbb{R}^{r \times \rho}$, such that $\mathbf{U}_1 = \mathbf{X}\mathbf{Q}$. By the definition of $\mathbf{U}_1$, we derive that

$$
\mathbf{U}_1^\top\mathbf{U}_1 = \mathbf{Q}^\top\mathbf{X}^\top\mathbf{X}\mathbf{Q} = \mathbf{Q}^\top\mathbf{Q} = \mathbf{I}_\rho,
$$

which implies that $\mathbf{Q}$ is a column-orthonormal matrix.

We extend $\mathbf{Q}$ to an $r \times r$ orthogonal matrix $\tilde{\mathbf{Q}} = [\mathbf{Q}, \mathbf{P}]$. Let $\mathbf{V}_1 = \mathbf{X}\mathbf{P}$, then $[\mathbf{U}_1, \mathbf{V}_1] = [\mathbf{X}\mathbf{Q}, \mathbf{X}\mathbf{P}] = \mathbf{X}\tilde{\mathbf{Q}}$. Since $\tilde{\mathbf{Q}}^\top\mathbf{X}^\top\mathbf{X}\tilde{\mathbf{Q}} = \tilde{\mathbf{Q}}^\top\tilde{\mathbf{Q}} = \mathbf{I}_r$, then $[\mathbf{U}_1, \mathbf{V}_1]$ is also a column-orthonormal matrix, which means that

$$
\mathbf{V}_1 = [\mathbf{v}_1, \mathbf{v}_2, \ldots, \mathbf{v}_{r-\rho}], \text{ with } \mathbf{v}_1, \mathbf{v}_2, \ldots, \mathbf{v}_{r-\rho} \in \mathbf{U}_1^\perp; \mathbf{V}_1^\top\mathbf{V}_1 = \mathbf{I}_{r-\rho}.
$$

Let $\mathbf{U}_2 = [\mathbf{u}_{\rho+1}, \mathbf{u}_{\rho+2}, \ldots, \mathbf{u}_{r_A}]$, and then $\mathbf{U} = [\mathbf{U}_1, \mathbf{U}_2]$. By substituting $\mathbf{U}$ and $\mathbf{X}$, we obtain

$$
\begin{aligned}
\mathbf{X}^\top\mathbf{U} &= \tilde{\mathbf{Q}}\begin{bmatrix}\mathbf{U}_1^\top \\ \mathbf{V}_1^\top\end{bmatrix}[\mathbf{U}_1, \mathbf{U}_2] \\
&\overset{(a)}{=} \tilde{\mathbf{Q}}\begin{bmatrix}\mathbf{I}_\rho & \mathbf{0} \\ \mathbf{0} & \mathbf{V}_1^\top\mathbf{U}_2\end{bmatrix},
\end{aligned}
$$

where $(a)$ is from $\mathbf{v}_1, \mathbf{v}_2, \ldots, \mathbf{v}_{r-\rho} \in \mathbf{U}_1^\perp$.

Since $\mathsf{Tr}(\mathbf{X}^\top \mathbf{U} \mathbf{U}^\top \mathbf{X}) = \rho$, it follows that

$$
\begin{aligned}
\rho &= \mathsf{Tr}(\mathbf{X}^\top \mathbf{U} \mathbf{U}^\top \mathbf{X}) \\
&= \mathsf{Tr}\left( \tilde{\mathbf{Q}} \begin{bmatrix} \mathbf{I}_\rho & \mathbf{0} \\ \mathbf{0} & \mathbf{V}_1^\top \mathbf{U}_2 \end{bmatrix} \begin{bmatrix} \mathbf{I}_\rho & \mathbf{0} \\ \mathbf{0} & \mathbf{U}_2^\top \mathbf{V}_1 \end{bmatrix} \tilde{\mathbf{Q}}^\top \right) \\
&= \mathsf{Tr}\left( \begin{bmatrix} \mathbf{I}_\rho & \mathbf{0} \\ \mathbf{0} & \mathbf{V}_1^\top \mathbf{U}_2 \mathbf{U}_2^\top \mathbf{V}_1 \end{bmatrix} \right) \\
&= \rho + \mathsf{Tr}(\mathbf{V}_1^\top \mathbf{U}_2 \mathbf{U}_2^\top \mathbf{V}_1).
\end{aligned}
$$

After cancelling the term $\rho$ on both sides, we obtain $\mathsf{Tr}(\mathbf{V}_1^\top \mathbf{U}_2 \mathbf{U}_2^\top \mathbf{V}_1) = \|\mathbf{U}_2^\top \mathbf{V}_1\|_{\mathsf{F}}^2 = 0$. Hence, we have that $\mathbf{U}_2^\top \mathbf{V}_1 = \mathbf{0}$, which implies that $\mathbf{v}_1, \mathbf{v}_2, \ldots, \mathbf{v}_{r-\rho} \in \mathbf{U}_2^\perp$. Moreover, since $\mathbf{v}_1, \mathbf{v}_2, \ldots, \mathbf{v}_{r-\rho} \in \mathbf{U}_1^\perp$ as well, we conclude that $\mathbf{v}_1, \mathbf{v}_2, \ldots, \mathbf{v}_{r-\rho} \in \mathbf{U}^\perp$.

Substituting $\mathbf{X} = [\mathbf{U}_1, \mathbf{V}_1]\tilde{\mathbf{Q}}^\top$ into $\mathbf{X}\mathbf{\Theta}\mathbf{X}^\top = \mathbf{A}_\rho$, we can obtain

$$
\begin{aligned}
[\mathbf{U}_1, \mathbf{V}_1]\tilde{\mathbf{Q}}^\top \mathbf{\Theta}\tilde{\mathbf{Q}}[\mathbf{U}_1, \mathbf{V}_1]^\top &= \mathbf{A}_\rho \\
&= \mathbf{U}_1 \mathsf{diag}(\lambda_1, \lambda_2, \ldots, \lambda_\rho)\mathbf{U}_1^\top \\
&= [\mathbf{U}_1, \mathbf{V}_1]\mathsf{diag}(\lambda_1, \lambda_2, \ldots, \lambda_\rho, 0, \ldots, 0)[\mathbf{U}_1, \mathbf{V}_1]^\top.
\end{aligned}
$$

Expanding both sides of the equation, together with $\mathbf{v}_1, \mathbf{v}_2, \ldots, \mathbf{v}_{r-\rho} \in \mathbf{U}^\perp$, we can obtain

$$
\tilde{\mathbf{Q}}^\top \mathbf{\Theta}\tilde{\mathbf{Q}} = \mathsf{diag}(\lambda_1, \lambda_2, \ldots, \lambda_\rho, 0, \ldots, 0).
$$

This implies that

$$
\mathbf{\Theta} = \tilde{\mathbf{Q}}\mathsf{diag}(\lambda_1, \lambda_2, \ldots, \lambda_\rho, 0, \ldots, 0)\tilde{\mathbf{Q}}^\top.
$$

To proceed, we first verify that $(\tilde{\mathbf{X}}, \tilde{\mathbf{\Theta}}) := ([\mathbf{U}_1, \mathbf{V}_1], \mathsf{diag}(\lambda_1, \lambda_2, \ldots, \lambda_\rho, 0, \ldots, 0))$ is indeed a saddle point and then prove $(\mathbf{X}, \mathbf{\Theta})$ is also a saddle point.

We compute the Euclidean gradients of $f_\infty$ with respect to $\mathbf{X}$ and $\mathbf{\Theta}$ as follows:

$$
\begin{aligned}
\nabla_{\mathbf{X}} f_\infty(\tilde{\mathbf{X}}, \tilde{\mathbf{\Theta}}) &= (\tilde{\mathbf{X}}\tilde{\mathbf{\Theta}}\tilde{\mathbf{X}}^\top - \mathbf{A})\tilde{\mathbf{X}}\tilde{\mathbf{\Theta}} = \tilde{\mathbf{X}}\tilde{\mathbf{\Theta}}^2 - \mathbf{A}\tilde{\mathbf{X}}\tilde{\mathbf{\Theta}}, \\
\nabla_{\mathbf{\Theta}} f_\infty(\tilde{\mathbf{X}}, \tilde{\mathbf{\Theta}}) &= \frac{1}{2}\tilde{\mathbf{X}}^\top(\tilde{\mathbf{X}}\tilde{\mathbf{\Theta}}\tilde{\mathbf{X}}^\top - \mathbf{A})\tilde{\mathbf{X}} = \frac{1}{2}(\tilde{\mathbf{\Theta}} - \tilde{\mathbf{X}}^\top \mathbf{A}\tilde{\mathbf{X}}).
\end{aligned}
$$

By plugging the expression of $(\tilde{\mathbf{X}}, \tilde{\mathbf{\Theta}})$ in, we obtain

$$
\begin{aligned}
\nabla_{\mathbf{X}} f_\infty(\tilde{\mathbf{X}}, \tilde{\mathbf{\Theta}}) &= \tilde{\mathbf{X}}\tilde{\mathbf{\Theta}}^2 - \mathbf{A}\tilde{\mathbf{X}}\tilde{\mathbf{\Theta}} \\
&= [\lambda_1^2 \mathbf{u}_1, \lambda_2^2 \mathbf{u}_2, \ldots, \lambda_\rho^2 \mathbf{u}_\rho, \mathbf{0}, \ldots, \mathbf{0}] - [\lambda_1^2 \mathbf{u}_1, \lambda_2^2 \mathbf{u}_2, \ldots, \lambda_\rho^2 \mathbf{u}_\rho, \mathbf{0}, \ldots, \mathbf{0}] \\
&= \mathbf{0}, \\
\nabla_{\mathbf{\Theta}} f_\infty(\tilde{\mathbf{X}}, \tilde{\mathbf{\Theta}}) &= \frac{1}{2}(\tilde{\mathbf{\Theta}} - \tilde{\mathbf{X}}^\top \mathbf{A}\tilde{\mathbf{X}}) \\
&= \frac{1}{2}(\mathsf{diag}(\lambda_1, \lambda_2, \ldots, \lambda_\rho, 0, \ldots, 0) - \mathsf{diag}(\lambda_1, \lambda_2, \ldots, \lambda_\rho, 0, \ldots, 0)) \\
&= \mathbf{0}.
\end{aligned}
$$

Then, the Riemannian gradient is

$$
(\mathbf{I}_m - \tilde{\mathbf{X}}\tilde{\mathbf{X}}^\top)\nabla_{\mathbf{X}} f_\infty(\tilde{\mathbf{X}}, \tilde{\mathbf{\Theta}}) + \frac{\tilde{\mathbf{X}}}{2}(\tilde{\mathbf{X}}^\top \nabla_{\mathbf{X}} f_\infty(\tilde{\mathbf{X}}, \tilde{\mathbf{\Theta}}) - \nabla_{\mathbf{X}} f_\infty(\tilde{\mathbf{X}}, \tilde{\mathbf{\Theta}})^\top \tilde{\mathbf{X}}) = \mathbf{0}.
$$

Therefore, $(\tilde{\mathbf{X}}, \tilde{\mathbf{\Theta}})$ is a stationary point in the Riemannian sense.

We now show that $(\tilde{\mathbf{X}}, \tilde{\mathbf{\Theta}})$ is neither a local minimum nor a local maximum of the objective function.

For any $0 < \nu < \lambda_{r_A}$, we will construct a pair $(\tilde{\mathbf{X}}_+, \tilde{\mathbf{\Theta}}_+)$, such that $f_\infty(\tilde{\mathbf{X}}_+, \tilde{\mathbf{\Theta}}_+) > f_\infty(\tilde{\mathbf{X}}, \tilde{\mathbf{\Theta}})$, $d\left((\tilde{\mathbf{X}}_+, \tilde{\mathbf{\Theta}}_+), (\tilde{\mathbf{X}}, \tilde{\mathbf{\Theta}})\right) := \sqrt{\|\tilde{\mathbf{X}}_+ - \tilde{\mathbf{X}}\|_{\mathsf{F}}^2 + \|\tilde{\mathbf{\Theta}}_+ - \tilde{\mathbf{\Theta}}\|_{\mathsf{F}}^2} \leq \nu$ and $\tilde{\mathbf{X}}_+^\top \tilde{\mathbf{X}}_+ = \mathbf{I}_r$.

Let $\tilde{\mathbf{X}}_+ = \tilde{\mathbf{X}} = [\mathbf{U}_1, \mathbf{V}_1]$ and $\tilde{\mathbf{\Theta}}_+ = \mathrm{diag}(\lambda_1 - \nu, \lambda_2, \ldots, \lambda_\rho, 0, \ldots, 0)$. By construction, $\tilde{\mathbf{X}}_+^\top \tilde{\mathbf{X}}_+ = \mathbf{I}_r$ and $d\left((\tilde{\mathbf{X}}_+, \tilde{\mathbf{\Theta}}_+), (\tilde{\mathbf{X}}, \tilde{\mathbf{\Theta}})\right) = \sqrt{\nu^2} \le \nu$ hold. The value of the objective function is

$$
\begin{aligned}
f_\infty(\tilde{\mathbf{X}}_+, \tilde{\mathbf{\Theta}}_+) &= \frac{1}{4} \|\tilde{\mathbf{X}}_+ \tilde{\mathbf{\Theta}}_+ \tilde{\mathbf{X}}_+^\top - \mathbf{A}\|_\mathsf{F}^2 \\
&= \frac{1}{4} \|(\lambda_1 - \nu)\mathbf{u}_1 \mathbf{u}_1^\top + \sum_{i=2}^\rho \lambda_i \mathbf{u}_i \mathbf{u}_i^\top - \sum_{i=1}^{r_A} \lambda_i \mathbf{u}_i \mathbf{u}_i^\top\|_\mathsf{F}^2 \\
&= \frac{1}{4} \|\nu \mathbf{u}_1 \mathbf{u}_1^\top + \sum_{i=\rho+1}^{r_A} \lambda_i \mathbf{u}_i \mathbf{u}_i^\top\|_\mathsf{F}^2 \\
&\overset{(b)}{=} \frac{1}{4}(\nu^2 \|\mathbf{u}_1 \mathbf{u}_1^\top\|_\mathsf{F}^2 + \|\tilde{\mathbf{X}} \tilde{\mathbf{\Theta}} \tilde{\mathbf{X}}^\top - \mathbf{A}\|_\mathsf{F}^2) \\
&> f_\infty(\tilde{\mathbf{X}}, \tilde{\mathbf{\Theta}}),
\end{aligned}
$$

where $(b)$ is by the orthogonality of $\{\mathbf{u}_1, \mathbf{u}_2, \ldots, \mathbf{u}_{r_A}\}$.

We now try to construct a pair $(\tilde{\mathbf{X}}_-, \tilde{\mathbf{\Theta}}_-)$, such that $f_\infty(\tilde{\mathbf{X}}_-, \tilde{\mathbf{\Theta}}_-) < f_\infty(\tilde{\mathbf{X}}, \tilde{\mathbf{\Theta}})$, $d\left((\tilde{\mathbf{X}}_-, \tilde{\mathbf{\Theta}}_-), (\tilde{\mathbf{X}}, \tilde{\mathbf{\Theta}})\right) := \sqrt{\|\tilde{\mathbf{X}}_- - \tilde{\mathbf{X}}\|_\mathsf{F}^2 + \|\tilde{\mathbf{\Theta}}_- - \tilde{\mathbf{\Theta}}\|_\mathsf{F}^2} \le \nu$, and $\tilde{\mathbf{X}}_-^\top \tilde{\mathbf{X}}_- = \mathbf{I}_r$.

Since $\mathbf{v}_i \in \mathbf{U}^\perp$ for any $i \in \{1, 2, \ldots, r - \rho\}$, it follows that $\mathbf{v}_i \in \mathrm{span}\{\mathbf{u}_{r_A+1}, \mathbf{u}_{r_A+2}, \ldots, \mathbf{u}_m\}$. Accordingly, we consider

$$
\begin{aligned}
\tilde{\mathbf{X}}_- &= [\mathbf{U}_1, k\mathbf{v}_1 + s\mathbf{u}_{\rho+1}, \mathbf{v}_2, \ldots, \mathbf{v}_{r-\rho}], \\
\tilde{\mathbf{\Theta}}_- &= \mathrm{diag}(\lambda_1, \lambda_2, \ldots, \lambda_\rho, \nu_0, 0, \ldots, 0),
\end{aligned}
$$

where $k, s, \nu_0 > 0, k^2 + s^2 = 1$ and $k, s, \nu_0$ will be given later. We can easily verify that $\tilde{\mathbf{X}}_-^\top \tilde{\mathbf{X}}_- = \mathbf{I}_r$ holds. The distance is

$$
\begin{aligned}
d\left((\tilde{\mathbf{X}}_-, \tilde{\mathbf{\Theta}}_-), (\tilde{\mathbf{X}}, \tilde{\mathbf{\Theta}})\right) &= \sqrt{\|\tilde{\mathbf{X}}_- - \tilde{\mathbf{X}}\|_\mathsf{F}^2 + \|\tilde{\mathbf{\Theta}}_- - \tilde{\mathbf{\Theta}}\|_\mathsf{F}^2} \\
&= \sqrt{\|(k-1)\mathbf{v}_1 + s\mathbf{u}_{\rho+1}\|^2 + \nu_0^2} \\
&= \sqrt{(k-1)^2 + s^2 + \nu_0^2} \\
&= \sqrt{2 - 2k + \nu_0^2}.
\end{aligned}
$$

Let $k = 1 - \frac{\nu^2}{4}$, $s = \sqrt{1 - k^2}$ and $\nu_0 \le \frac{\nu}{2}$, then $d\left((\tilde{\mathbf{X}}_-, \tilde{\mathbf{\Theta}}_-), (\tilde{\mathbf{X}}, \tilde{\mathbf{\Theta}})\right) \le \sqrt{\frac{\nu^2}{2} + \frac{\nu^2}{4}} \le \nu$. The value of the objective function is

$$
\begin{aligned}
&f_\infty(\tilde{\mathbf{X}}_-, \tilde{\mathbf{\Theta}}_-) \\
&= \frac{1}{4} \|\tilde{\mathbf{X}}_- \tilde{\mathbf{\Theta}}_- \tilde{\mathbf{X}}_-^\top - \mathbf{A}\|_\mathsf{F}^2 \\
&= \frac{1}{4} \|\nu_0(k^2 \mathbf{v}_1 \mathbf{v}_1^\top + ks\mathbf{u}_{\rho+1}\mathbf{v}_1^\top + ks\mathbf{v}_1\mathbf{u}_{\rho+1}^\top) + (\nu_0 s^2 - \lambda_\rho)\mathbf{u}_{\rho+1}\mathbf{u}_{\rho+1}^\top - \sum_{i=\rho+2}^{r_A} \lambda_i \mathbf{u}_i \mathbf{u}_i^\top\|_\mathsf{F}^2 \\
&\overset{(c)}{=} \frac{1}{4}\left(\nu_0^2(k^4 + k^2 s^2 \|\mathbf{u}_{\rho+1}\mathbf{v}_1^\top\|_\mathsf{F}^2 + k^2 s^2 \|\mathbf{v}_1\mathbf{u}_{\rho+1}^\top\|_\mathsf{F}^2) + (\nu_0 s^2 - \lambda_\rho)^2\right) + f_\infty(\tilde{\mathbf{X}}, \tilde{\mathbf{\Theta}}) - \frac{1}{4}\lambda_\rho^2 \\
&= \frac{1}{4}\nu_0^2\left(k^4 + 2k^2 s^2 + s^4\right) - \frac{1}{2}\nu_0 \lambda_\rho s^2 + f_\infty(\tilde{\mathbf{X}}, \tilde{\mathbf{\Theta}}),
\end{aligned}
$$

where $(c)$ is from the orthogonality of $\{\mathbf{u}_1, \mathbf{u}_2, \ldots, \mathbf{u}_{r_A}, \mathbf{v}_1\}$. Let $\nu_0 > 0$ be sufficiently small. Then $\frac{1}{4}\nu_0^2\left(k^4 + 2k^2 s^2 + s^4\right) - \frac{1}{2}\nu_0 \lambda_\rho s^2 < 0$. This ensures that the perturbed pair leads to a strictly smaller objective value, i.e., $f_\infty(\tilde{\mathbf{X}}_-, \tilde{\mathbf{\Theta}}_-) < f_\infty(\tilde{\mathbf{X}}, \tilde{\mathbf{\Theta}})$.

Therefore, we have verified that $(\tilde{\mathbf{X}}, \tilde{\mathbf{\Theta}})$ is a saddle point. Building upon this result, we now proceed to show that $(\mathbf{X}, \mathbf{\Theta}) = (\tilde{\mathbf{X}}\tilde{\mathbf{Q}}^\top, \tilde{\mathbf{Q}}\tilde{\mathbf{\Theta}}\tilde{\mathbf{Q}}^\top)$ is also a saddle point.

Plugging in the expression of $(\mathbf{X}, \boldsymbol{\Theta})$, we obtain the Euclidean gradients as follows:

$$
\begin{aligned}
\nabla_{\mathbf{X}} f_\infty(\mathbf{X}, \boldsymbol{\Theta}) &= (\mathbf{X}\boldsymbol{\Theta}\mathbf{X}^\top - \mathbf{A})\mathbf{X}\boldsymbol{\Theta} \\
&= (\tilde{\mathbf{X}}\tilde{\mathbf{Q}}^\top\tilde{\mathbf{Q}}\tilde{\boldsymbol{\Theta}}\tilde{\mathbf{Q}}^\top\tilde{\mathbf{Q}}\tilde{\mathbf{X}}^\top - \mathbf{A})\tilde{\mathbf{X}}\tilde{\mathbf{Q}}^\top\tilde{\mathbf{Q}}\tilde{\boldsymbol{\Theta}}\tilde{\mathbf{Q}}^\top \\
&= (\tilde{\mathbf{X}}\tilde{\boldsymbol{\Theta}}^2 - \mathbf{A}\tilde{\mathbf{X}}\tilde{\boldsymbol{\Theta}})\tilde{\mathbf{Q}}^\top \\
&= \mathbf{0}, \\
\nabla_{\boldsymbol{\Theta}} f_\infty(\mathbf{X}, \boldsymbol{\Theta}) &= \frac{1}{2}\mathbf{X}^\top(\mathbf{X}\boldsymbol{\Theta}\mathbf{X}^\top - \mathbf{A})\mathbf{X} \\
&= \frac{1}{2}\tilde{\mathbf{Q}}\tilde{\mathbf{X}}^\top(\tilde{\mathbf{X}}\tilde{\mathbf{Q}}^\top\tilde{\mathbf{Q}}\tilde{\boldsymbol{\Theta}}\tilde{\mathbf{Q}}^\top\tilde{\mathbf{Q}}\tilde{\mathbf{X}}^\top - \mathbf{A})\tilde{\mathbf{X}}\tilde{\mathbf{Q}}^\top \\
&= \frac{1}{2}\tilde{\mathbf{Q}}(\tilde{\boldsymbol{\Theta}} - \tilde{\mathbf{X}}^\top\mathbf{A}\tilde{\mathbf{X}})\tilde{\mathbf{Q}}^\top \\
&= \mathbf{0}.
\end{aligned}
$$

Then, the Riemannian gradient is

$$
(\mathbf{I}_m - \mathbf{X}\mathbf{X}^\top)\nabla_{\mathbf{X}} f_\infty(\mathbf{X}, \boldsymbol{\Theta}) + \frac{\mathbf{X}}{2}(\mathbf{X}^\top\nabla_{\mathbf{X}} f_\infty(\mathbf{X}, \boldsymbol{\Theta}) - \nabla_{\mathbf{X}} f_\infty(\mathbf{X}, \boldsymbol{\Theta})^\top\mathbf{X}) = \mathbf{0}.
$$

Therefore, $(\mathbf{X}, \boldsymbol{\Theta})$ is a stationary point in the Riemannian sense.

Let $(\mathbf{X}_+, \boldsymbol{\Theta}_+) = (\tilde{\mathbf{X}}_+\tilde{\mathbf{Q}}^\top, \tilde{\mathbf{Q}}\boldsymbol{\Theta}_+\tilde{\mathbf{Q}}^\top)$, $(\mathbf{X}_-, \boldsymbol{\Theta}_-) = (\tilde{\mathbf{X}}_-\tilde{\mathbf{Q}}^\top, \tilde{\mathbf{Q}}\boldsymbol{\Theta}_-\tilde{\mathbf{Q}}^\top)$. The distance is

$$
\begin{aligned}
d\left((\mathbf{X}_+, \boldsymbol{\Theta}_+), (\mathbf{X}, \boldsymbol{\Theta})\right) &= \sqrt{\|\mathbf{X}_+ - \mathbf{X}\|_{\mathsf{F}}^2 + \|\boldsymbol{\Theta}_+ - \boldsymbol{\Theta}\|_{\mathsf{F}}^2} \\
&= \sqrt{\|(\tilde{\mathbf{X}}_+ - \tilde{\mathbf{X}})\tilde{\mathbf{Q}}^\top\|_{\mathsf{F}}^2 + \|\tilde{\mathbf{Q}}(\tilde{\boldsymbol{\Theta}}_+ - \tilde{\boldsymbol{\Theta}})\tilde{\mathbf{Q}}^\top\|_{\mathsf{F}}^2} \\
&= \sqrt{\|\tilde{\mathbf{X}}_+ - \tilde{\mathbf{X}}\|_{\mathsf{F}}^2 + \|\tilde{\boldsymbol{\Theta}}_+ - \tilde{\boldsymbol{\Theta}}\|_{\mathsf{F}}^2} \\
&= d\left((\tilde{\mathbf{X}}_+, \tilde{\boldsymbol{\Theta}}_+), (\tilde{\mathbf{X}}, \tilde{\boldsymbol{\Theta}})\right).
\end{aligned}
$$

In the same manner, we can obtain that $d\left((\mathbf{X}_-, \boldsymbol{\Theta}_-), (\mathbf{X}, \boldsymbol{\Theta})\right) = d\left((\tilde{\mathbf{X}}_-, \tilde{\boldsymbol{\Theta}}_-), (\tilde{\mathbf{X}}, \tilde{\boldsymbol{\Theta}})\right)$. By the orthogonality of $\tilde{\mathbf{Q}}$, the following three identities hold:

$$
\begin{aligned}
\mathbf{X}\boldsymbol{\Theta}\mathbf{X}^\top &= \tilde{\mathbf{X}}\tilde{\mathbf{Q}}^\top\tilde{\mathbf{Q}}\tilde{\boldsymbol{\Theta}}\tilde{\mathbf{Q}}^\top\tilde{\mathbf{Q}}\tilde{\mathbf{X}}^\top = \tilde{\mathbf{X}}\tilde{\boldsymbol{\Theta}}\tilde{\mathbf{X}}^\top, \\
\mathbf{X}_+\boldsymbol{\Theta}_+\mathbf{X}_+^\top &= \tilde{\mathbf{X}}_+\tilde{\mathbf{Q}}^\top\tilde{\mathbf{Q}}\tilde{\boldsymbol{\Theta}}_+\tilde{\mathbf{Q}}^\top\tilde{\mathbf{Q}}\tilde{\mathbf{X}}_+^\top = \tilde{\mathbf{X}}_+\tilde{\boldsymbol{\Theta}}_+\tilde{\mathbf{X}}_+^\top, \\
\mathbf{X}_-\boldsymbol{\Theta}_-\mathbf{X}_-^\top &= \tilde{\mathbf{X}}_-\tilde{\mathbf{Q}}^\top\tilde{\mathbf{Q}}\tilde{\boldsymbol{\Theta}}_-\tilde{\mathbf{Q}}^\top\tilde{\mathbf{Q}}\tilde{\mathbf{X}}_-^\top = \tilde{\mathbf{X}}_-\tilde{\boldsymbol{\Theta}}_-\tilde{\mathbf{X}}_-^\top.
\end{aligned}
$$

Then, we have $f(\mathbf{X}, \boldsymbol{\Theta}) = f(\tilde{\mathbf{X}}, \tilde{\boldsymbol{\Theta}})$, $f(\mathbf{X}_+, \boldsymbol{\Theta}_+) = f(\tilde{\mathbf{X}}_+, \tilde{\boldsymbol{\Theta}}_+)$ and $f(\mathbf{X}_-, \boldsymbol{\Theta}_-) = f(\tilde{\mathbf{X}}_-, \tilde{\boldsymbol{\Theta}}_-)$. Thus, we obtain the strict inequality $f(\mathbf{X}_-, \boldsymbol{\Theta}_-) < f(\mathbf{X}, \boldsymbol{\Theta}) < f(\mathbf{X}_+, \boldsymbol{\Theta}_+)$. Therefore, $(\mathbf{X}, \boldsymbol{\Theta})$ is also a saddle point.

We now turn to the case $\rho = 0$, i.e., $\mathbf{X}\boldsymbol{\Theta}\mathbf{X}^\top = \mathbf{A}_0 = \mathbf{0}$. Consequently, $\boldsymbol{\Theta} = \mathbf{X}^\top\mathbf{A}_0\mathbf{X} = \mathbf{0}$. Let $\mathbf{X}$ be expressed as $\mathbf{X} = [\mathbf{x}_1, \mathbf{x}_2, \ldots, \mathbf{x}_r]$, where each $\mathbf{x}_i$ is a column vector. Since $\mathsf{Tr}(\mathbf{X}^\top\mathbf{U}\mathbf{U}^\top\mathbf{X}) = \|\mathbf{U}^\top\mathbf{X}\|_{\mathsf{F}}^2 = 0$, it follows that $\mathbf{U}^\top\mathbf{X} = \mathbf{0}$. Hence, each $\mathbf{x}_i$ lies in $\mathbf{U}^\perp$ for $i \in \{1, 2, \ldots, r\}$.

We compute the Euclidean gradient of the objective function $f_\infty$ with respect to $\mathbf{X}$ and $\boldsymbol{\Theta}$

$$
\begin{aligned}
\nabla_{\mathbf{X}} f_\infty(\mathbf{X}, \boldsymbol{\Theta}) &= (\mathbf{X}\boldsymbol{\Theta}\mathbf{X}^\top - \mathbf{A})\mathbf{X}\boldsymbol{\Theta} = \mathbf{X}\boldsymbol{\Theta}^2 - \mathbf{A}\mathbf{X}\boldsymbol{\Theta} = \mathbf{0}, \\
\nabla_{\boldsymbol{\Theta}} f_\infty(\mathbf{X}, \boldsymbol{\Theta}) &= \frac{1}{2}\mathbf{X}^\top(\mathbf{X}\boldsymbol{\Theta}\mathbf{X}^\top - \mathbf{A})\mathbf{X} = \frac{1}{2}(\boldsymbol{\Theta} - \mathbf{X}^\top\mathbf{A}\mathbf{X}) = \mathbf{0}.
\end{aligned}
$$

Then, the Riemannian gradient is

$$
(\mathbf{I}_m - \mathbf{X}\mathbf{X}^\top)\nabla_{\mathbf{X}} f_\infty(\mathbf{X}, \boldsymbol{\Theta}) + \frac{\mathbf{X}}{2}(\mathbf{X}^\top\nabla_{\mathbf{X}} f_\infty(\mathbf{X}, \boldsymbol{\Theta}) - \nabla_{\mathbf{X}} f_\infty(\mathbf{X}, \boldsymbol{\Theta})^\top\mathbf{X}) = \mathbf{0}.
$$

Therefore, $(\mathbf{X}, \boldsymbol{\Theta})$ is a stationary point in the Riemannian sense.

For any $0 < \nu < \lambda_{r_A}$, we construct the pair $(\mathbf{X}_+, \boldsymbol{\Theta}_+)$ as follows:

$$\mathbf{X}_+ = [k\mathbf{x}_1 + s\mathbf{u}_1, \mathbf{x}_2, \ldots, \mathbf{x}_r],$$
$$\boldsymbol{\Theta}_+ = \mathrm{diag}(-\nu_1, 0, \ldots, 0),$$

where $k = 1 - \frac{\nu^2}{4}, s = \sqrt{1 - k^2}$, and $0 < \nu_1 \leq \frac{\nu}{2}$. We can easily verify that $\mathbf{X}_+^\top \mathbf{X}_+ = \mathbf{I}_r$ and the distance is

$$
\begin{aligned}
d\left((\mathbf{X}_+, \boldsymbol{\Theta}_+), (\mathbf{X}, \boldsymbol{\Theta})\right) &= \sqrt{\|\mathbf{X}_+ - \mathbf{X}\|_\mathsf{F}^2 + \|\boldsymbol{\Theta}_+ - \boldsymbol{\Theta}\|_\mathsf{F}^2} \\
&= \sqrt{\|(k-1)\mathbf{x}_1 + s\mathbf{u}_1\|^2 + \nu_1^2} \\
&= \sqrt{(k-1)^2 + s^2 + \nu_1^2} \\
&\leq \sqrt{\frac{\nu^2}{2} + \frac{\nu^2}{4}} \\
&\leq \nu.
\end{aligned}
$$

The value of the objective function is

$$
\begin{aligned}
f_\infty(\mathbf{X}_+, \boldsymbol{\Theta}_+) &= \frac{1}{4}\|\mathbf{X}_+ \boldsymbol{\Theta}_+ \mathbf{X}_+^\top - \mathbf{A}\|_\mathsf{F}^2 \\
&= \frac{1}{4}\| -\nu_1 \left(k^2 \mathbf{x}_1 \mathbf{x}_1^\top + s^2 \mathbf{u}_1 \mathbf{u}_1^\top\right) - \sum_{i=1}^{r_A} \lambda_i \mathbf{u}_i \mathbf{u}_i^\top \|_\mathsf{F}^2 \\
&= \frac{1}{4}\|\nu_1 k^2 \mathbf{x}_1 \mathbf{x}_1^\top + \nu_1 s^2 \mathbf{u}_1 \mathbf{u}_1^\top + \sum_{i=1}^{r_A} \lambda_i \mathbf{u}_i \mathbf{u}_i^\top \|_\mathsf{F}^2 \\
&\stackrel{(d)}{=} \frac{1}{4}\left(\nu_1^2 k^4 + \nu_1^2 s^4 + 2\nu_1 \lambda_1 s^2 + \|\mathbf{X}\boldsymbol{\Theta}\mathbf{X}^\top - \mathbf{A}\|_\mathsf{F}^2\right) \\
&> f_\infty(\mathbf{X}, \boldsymbol{\Theta}),
\end{aligned}
$$

where $(d)$ is due to the orthogonality of $\{\mathbf{u}_1, \mathbf{u}_2, \ldots, \mathbf{u}_{r_A}, \mathbf{x}_1\}$. Now consider the pair $(\mathbf{X}_-, \boldsymbol{\Theta}_-)$ defined as:

$$\mathbf{X}_- = [k\mathbf{x}_1 + s\mathbf{u}_1, \mathbf{x}_2, \ldots, \mathbf{x}_r],$$
$$\boldsymbol{\Theta}_- = \mathrm{diag}(\nu_2, 0, \ldots, 0),$$

where $k = 1 - \frac{\nu^2}{4}, s = \sqrt{1 - k^2}$, and $0 < \nu_2 \leq \frac{\nu}{2}$. It can be verified that $\mathbf{X}_-^\top \mathbf{X}_- = \mathbf{I}_r$, and the distance is

$$
\begin{aligned}
d\left((\mathbf{X}_-, \boldsymbol{\Theta}_-), (\mathbf{X}, \boldsymbol{\Theta})\right) &= \sqrt{\|\mathbf{X}_- - \mathbf{X}\|_\mathsf{F}^2 + \|\boldsymbol{\Theta}_- - \boldsymbol{\Theta}\|_\mathsf{F}^2} \\
&= \sqrt{\|(k-1)\mathbf{x}_1 + s\mathbf{u}_1\|^2 + \nu_2^2} \\
&= \sqrt{(k-1)^2 + s^2 + \nu_2^2} \\
&\leq \sqrt{\frac{\nu^2}{2} + \frac{\nu^2}{4}} \\
&\leq \nu.
\end{aligned}
$$

The value of the objective function is

$$
\begin{aligned}
f_\infty(\mathbf{X}_-, \boldsymbol{\Theta}_-) &= \frac{1}{4} \|\mathbf{X}_- \boldsymbol{\Theta}_- \mathbf{X}_-^\top - \mathbf{A}\|_\mathsf{F}^2 \\
&= \frac{1}{4} \|\nu_2 \left(k^2 \mathbf{x}_1 \mathbf{x}_1^\top + s^2 \mathbf{u}_1 \mathbf{u}_1^\top\right) - \sum_{i=1}^{r_A} \lambda_i \mathbf{u}_i \mathbf{u}_i^\top\|_\mathsf{F}^2 \\
&= \frac{1}{4} \|\nu_2 k^2 \mathbf{x}_1 \mathbf{x}_1^\top + \nu_2 s^2 \mathbf{u}_1 \mathbf{u}_1^\top - \sum_{i=1}^{r_A} \lambda_i \mathbf{u}_i \mathbf{u}_i^\top\|_\mathsf{F}^2 \\
&\overset{(e)}{=} \frac{1}{4} \left(\nu_2^2 k^4 + \nu_2^2 s^4 - 2\nu_2 \lambda_1 s^2 + \|\mathbf{X}\boldsymbol{\Theta}\mathbf{X}^\top - \mathbf{A}\|_\mathsf{F}^2\right) \\
&= \frac{1}{4} \left(\nu_2^2 (k^4 + s^4) - 2\nu_2 \lambda_1 s^2\right) + f_\infty(\mathbf{X}, \boldsymbol{\Theta}),
\end{aligned}
$$

where $(e)$ is by the orthogonality of $\{\mathbf{u}_1, \mathbf{u}_2, \ldots, \mathbf{u}_{r_A}, \mathbf{x}_1\}$. Let $\nu_2 > 0$ be sufficiently small. Then $\frac{1}{4} \left(\nu_2^2(k^4 + s^4) - 2\nu_2 \lambda_1 s^2\right) < 0$. This guarantees that $f_\infty(\mathbf{X}_-, \boldsymbol{\Theta}_-) < f_\infty(\mathbf{X}, \boldsymbol{\Theta})$. Therefore, $(\mathbf{X}, \boldsymbol{\Theta})$ is also a saddle point when $\rho = 0$. $\qquad\square$

### E.10 PROOF OF LEMMA 4.2

*Proof.* We begin by computing the Euclidean gradients of $f_\infty$ and $f$ with respect to $\mathbf{X}$ and $\boldsymbol{\Theta}$:

$$
\begin{aligned}
\nabla_{\mathbf{X}} f_\infty(\mathbf{X}, \boldsymbol{\Theta}) &= (\mathbf{X}\boldsymbol{\Theta}\mathbf{X}^\top - \mathbf{A})\mathbf{X}\boldsymbol{\Theta}, \\
\nabla_{\boldsymbol{\Theta}} f_\infty(\mathbf{X}, \boldsymbol{\Theta}) &= \frac{1}{2} \mathbf{X}^\top (\mathbf{X}\boldsymbol{\Theta}\mathbf{X}^\top - \mathbf{A})\mathbf{X}, \\
\nabla_{\mathbf{X}} f(\mathbf{X}, \boldsymbol{\Theta}) &= \mathcal{M}^* \mathcal{M}(\mathbf{X}\boldsymbol{\Theta}\mathbf{X}^\top - \mathbf{A})\mathbf{X}\boldsymbol{\Theta}, \\
\nabla_{\boldsymbol{\Theta}} f(\mathbf{X}, \boldsymbol{\Theta}) &= \frac{1}{2} \mathbf{X}^\top \mathcal{M}^* \mathcal{M}(\mathbf{X}\boldsymbol{\Theta}\mathbf{X}^\top - \mathbf{A})\mathbf{X}.
\end{aligned}
$$

Then, we can obtain that the gap between population gradient and sensing gradient is

$$
\begin{aligned}
\|\nabla_{\mathbf{X}} f_\infty(\mathbf{X}, \boldsymbol{\Theta}) - \nabla_{\mathbf{X}} f(\mathbf{X}, \boldsymbol{\Theta})\|_\mathsf{F} &= \|(\mathcal{M}^* \mathcal{M} - \mathcal{I})(\mathbf{X}\boldsymbol{\Theta}\mathbf{X}^\top - \mathbf{A})\mathbf{X}\boldsymbol{\Theta}\|_\mathsf{F} \\
&\leq \|(\mathcal{M}^* \mathcal{M} - \mathcal{I})(\mathbf{X}\boldsymbol{\Theta}\mathbf{X}^\top - \mathbf{A})\|_\mathsf{F} \|\mathbf{X}\| \|\boldsymbol{\Theta}\| \\
&\overset{(f)}{\leq} 2\|(\mathcal{M}^* \mathcal{M} - \mathcal{I})(\mathbf{X}\boldsymbol{\Theta}\mathbf{X}^\top - \mathbf{A})\|_\mathsf{F} \\
&\leq 2\sqrt{m} \|(\mathcal{M}^* \mathcal{M} - \mathcal{I})(\mathbf{X}\boldsymbol{\Theta}\mathbf{X}^\top - \mathbf{A})\| \\
&\overset{(g)}{\leq} 2\sqrt{m}\delta \|\mathbf{X}\boldsymbol{\Theta}\mathbf{X}^\top - \mathbf{A}\|_\mathsf{F} \\
&\leq 2m\delta \|\mathbf{X}\boldsymbol{\Theta}\mathbf{X}^\top - \mathbf{A}\| \\
&\leq 2m\delta (\|\mathbf{X}\| \|\boldsymbol{\Theta}\| \|\mathbf{X}\| + \|\mathbf{A}\|) \\
&\overset{(f)}{\leq} 6m\delta,
\end{aligned}
$$

$$\|\nabla_{\boldsymbol{\Theta}} f_\infty(\mathbf{X}, \boldsymbol{\Theta}) - \nabla_{\boldsymbol{\Theta}} f(\mathbf{X}, \boldsymbol{\Theta})\|_{\mathsf{F}} = \frac{1}{2}\|\mathbf{X}^\top(\mathcal{M}^*\mathcal{M} - \mathcal{I})(\mathbf{X}\boldsymbol{\Theta}\mathbf{X}^\top - \mathbf{A})\mathbf{X}\|_{\mathsf{F}}$$

$$\leq \frac{1}{2}\|\mathbf{X}\|\|(\mathcal{M}^*\mathcal{M} - \mathcal{I})(\mathbf{X}\boldsymbol{\Theta}\mathbf{X}^\top - \mathbf{A})\|_{\mathsf{F}}\|\mathbf{X}\|$$

$$\leq \frac{1}{2}\|(\mathcal{M}^*\mathcal{M} - \mathcal{I})(\mathbf{X}\boldsymbol{\Theta}\mathbf{X}^\top - \mathbf{A})\|_{\mathsf{F}}$$

$$\leq \frac{1}{2}\sqrt{m}\|(\mathcal{M}^*\mathcal{M} - \mathcal{I})(\mathbf{X}\boldsymbol{\Theta}\mathbf{X}^\top - \mathbf{A})\|$$

$$\overset{(g)}{\leq} \frac{1}{2}\sqrt{m}\delta\|\mathbf{X}\boldsymbol{\Theta}\mathbf{X}^\top - \mathbf{A}\|_{\mathsf{F}}$$

$$\leq \frac{1}{2}m\delta\|\mathbf{X}\boldsymbol{\Theta}\mathbf{X}^\top - \mathbf{A}\|$$

$$\leq \frac{1}{2}m\delta(\|\mathbf{X}\|\|\boldsymbol{\Theta}\|\|\mathbf{X}\| + \|\mathbf{A}\|)$$

$$\overset{(f)}{\leq} \frac{3}{2}m\delta,$$

where $(f)$ is by $\|\mathbf{X}\| \leq 1, \|\boldsymbol{\Theta}\| \leq 2, \|\mathbf{A}\| \leq 1$; and $(g)$ is from Lemma F.11. Then, the difference between the two Riemannian gradients can be bounded as

$$\|\nabla_{\mathbf{X}}^R f_\infty(\mathbf{X}, \boldsymbol{\Theta}) - \nabla_{\mathbf{X}}^R f(\mathbf{X}, \boldsymbol{\Theta})\|_{\mathsf{F}} = \|(\mathbf{I}_m - \mathbf{X}\mathbf{X}^\top)(\nabla_{\mathbf{X}} f_\infty(\mathbf{X}, \boldsymbol{\Theta}) - \nabla_{\mathbf{X}} f(\mathbf{X}, \boldsymbol{\Theta}))$$

$$+ \frac{1}{2}\mathbf{X}\mathbf{X}^\top(\nabla_{\mathbf{X}} f_\infty(\mathbf{X}, \boldsymbol{\Theta}) - \nabla_{\mathbf{X}} f(\mathbf{X}, \boldsymbol{\Theta}))$$

$$- \frac{1}{2}\mathbf{X}(\nabla_{\mathbf{X}} f_\infty(\mathbf{X}, \boldsymbol{\Theta})^\top - \nabla_{\mathbf{X}} f(\mathbf{X}, \boldsymbol{\Theta})^\top)\mathbf{X}\|_{\mathsf{F}}$$

$$\leq \|(\mathbf{I}_m - \mathbf{X}\mathbf{X}^\top)(\nabla_{\mathbf{X}} f_\infty(\mathbf{X}, \boldsymbol{\Theta}) - \nabla_{\mathbf{X}} f(\mathbf{X}, \boldsymbol{\Theta}))\|_{\mathsf{F}}$$

$$+ \frac{1}{2}\|\mathbf{X}\mathbf{X}^\top(\nabla_{\mathbf{X}} f_\infty(\mathbf{X}, \boldsymbol{\Theta}) - \nabla_{\mathbf{X}} f(\mathbf{X}, \boldsymbol{\Theta}))\|_{\mathsf{F}}$$

$$+ \frac{1}{2}\|\mathbf{X}(\nabla_{\mathbf{X}} f_\infty(\mathbf{X}, \boldsymbol{\Theta})^\top - \nabla_{\mathbf{X}} f(\mathbf{X}, \boldsymbol{\Theta})^\top)\mathbf{X}\|_{\mathsf{F}}$$

$$\leq 6m\delta(\|\mathbf{I}_m - \mathbf{X}\mathbf{X}^\top\| + \frac{1}{2}\|\mathbf{X}\|\|\mathbf{X}\| + \frac{1}{2}\|\mathbf{X}\|\|\mathbf{X}\|)$$

$$\overset{(h)}{\leq} 12m\delta,$$

where $(h)$ is due to $\|\mathbf{I}_m - \mathbf{X}\mathbf{X}^\top\|, \|\mathbf{X}\| \leq 1$. $\qquad\square$

## F  OTHER USEFUL LEMMAS

**Lemma F.1** *Given a PSD matrix* $\mathbf{A}$, *we have that* $(\mathbf{I} + \mathbf{A})^{-1} \succeq \mathbf{I} - \mathbf{A}$.

*Proof.* Diagonalizing both sides and using $1/(1 + \lambda) \geq 1 - \lambda, \forall \lambda \geq 0$ yields the result.

$\qquad\square$

**Lemma F.2** *Let* $\mathbf{X} \in \mathsf{St}(m, r)$ *and* $\mathbf{U} \in \mathsf{St}(m, r_A)$. *Let* $\mathbf{U}_\perp \in \mathbb{R}^{m \times (m-r_A)}$ *be an orthonormal basis for the orthogonal complement of* $\mathsf{span}(\mathbf{U})$. *Denote* $\boldsymbol{\Phi} = \mathbf{U}^\top\mathbf{X} \in \mathbb{R}^{r_A \times r}$ *and* $\boldsymbol{\Psi} = \mathbf{U}_\perp^\top\mathbf{X} \in \mathbb{R}^{(m-r_A) \times r}$. *It is guaranteed that* $\sigma_i^2(\boldsymbol{\Phi}) + \sigma_i^2(\boldsymbol{\Psi}) = 1$ *holds for* $i \in \{1, 2, \ldots, r\}$.

*Proof.* Since $\mathbf{X}$ lies in the Stiefel manifold, we have that

$$\mathbf{I}_r = \mathbf{X}^\top\mathbf{X} = \mathbf{X}^\top\mathbf{I}_m\mathbf{X} = \mathbf{X}^\top[\mathbf{U}, \mathbf{U}_\perp]\begin{bmatrix} \mathbf{U}^\top \\ \mathbf{U}_\perp^\top \end{bmatrix}\mathbf{X} \tag{37}$$

$$= \boldsymbol{\Phi}^\top\boldsymbol{\Phi} + \boldsymbol{\Psi}^\top\boldsymbol{\Psi}.$$

Equation (37) shows that $\mathbf{\Psi}^\top \mathbf{\Psi}$ and $\mathbf{\Phi}^\top \mathbf{\Phi}$ commute, i.e.,

$$(\mathbf{\Phi}^\top \mathbf{\Phi})(\mathbf{\Psi}^\top \mathbf{\Psi}) = (\mathbf{\Phi}^\top \mathbf{\Phi})(\mathbf{I}_r - \mathbf{\Phi}^\top \mathbf{\Phi}) = \mathbf{\Phi}^\top \mathbf{\Phi} - \mathbf{\Phi}^\top \mathbf{\Phi}\mathbf{\Phi}^\top \mathbf{\Phi}$$
$$= (\mathbf{I}_r - \mathbf{\Phi}^\top \mathbf{\Phi})(\mathbf{\Phi}^\top \mathbf{\Phi}) = (\mathbf{\Psi}^\top \mathbf{\Psi})(\mathbf{\Phi}^\top \mathbf{\Phi}).$$

The commutativity shows that the eigenspaces of $\mathbf{\Phi}^\top \mathbf{\Phi}$ and $\mathbf{\Psi}^\top \mathbf{\Psi}$ coincide. As a result, we have again from (37) that $\sigma_i^2(\mathbf{\Phi}) + \sigma_i^2(\mathbf{\Psi}) = 1$ for $i \in \{1, 2, \ldots, r\}$. $\qquad\square$

**Lemma F.3** *Suppose that $\mathbf{P}$ and $\mathbf{Q}$ are $m \times m$ diagonal matrices, with non-negative diagonal entries. Let $\mathbf{S} \in \mathbb{S}^m$ be a positive definite matrix with smallest eigenvalue $\lambda_{\min}$, then we have that*

$$\mathsf{Tr}(\mathbf{PSQ}) \geq \lambda_{\min}\mathsf{Tr}(\mathbf{PQ}).$$

*Proof.* Let $p_i$ and $q_i$ be the $(i, i)$-th entry of $\mathbf{P}$ and $\mathbf{Q}$, respectively. Then we have that

$$\mathsf{Tr}(\mathbf{PSQ}) = \sum_i p_i \mathbf{S}_{i,i} q_i \geq \lambda_{\min} \sum_i p_i q_i = \lambda_{\min}\mathsf{Tr}(\mathbf{PQ}),$$

where the last inequality comes from $\mathbf{S}$ being positive definite, i.e., $\mathbf{S}_{i,i} = \mathbf{e}_i^\top \mathbf{S} \mathbf{e}_i \geq \lambda_{\min}$. $\qquad\square$

**Lemma F.4** *Let $\mathbf{A} \in \mathbb{R}^{m \times n}$ be a matrix with full column rank and $\mathbf{B} \in \mathbb{R}^{n \times p}$ be a non-zero matrix. Let $\sigma_{\min}(\cdot)$ denote the smallest non-zero singular value. Then it holds that $\sigma_{\min}(\mathbf{AB}) \geq \sigma_{\min}(\mathbf{A})\sigma_{\min}(\mathbf{B})$.*

*Proof.* Using the min-max principle for singular values,

$$\sigma_{\min}(\mathbf{AB}) = \min_{\|\mathbf{x}\|=1, \mathbf{x} \in \mathrm{ColSpan}(\mathbf{B})} \|\mathbf{AB}\mathbf{x}\|$$
$$= \min_{\|\mathbf{x}\|=1, \mathbf{x} \in \mathrm{ColSpan}(\mathbf{B})} \left\|\mathbf{A}\frac{\mathbf{B}\mathbf{x}}{\|\mathbf{B}\mathbf{x}\|}\right\| \cdot \|\mathbf{B}\mathbf{x}\|$$
$$\overset{(a)}{=} \min_{\|\mathbf{x}\|=1, \|\mathbf{y}\|=1, \mathbf{x} \in \mathrm{ColSpan}(\mathbf{B}), \mathbf{y} \in \mathrm{ColSpan}(\mathbf{B})} \|\mathbf{A}\mathbf{y}\| \cdot \|\mathbf{B}\mathbf{x}\|$$
$$\geq \min_{\|\mathbf{y}\|=1, \mathbf{y} \in \mathrm{ColSpan}(\mathbf{B})} \|\mathbf{A}\mathbf{y}\| \cdot \min_{\|\mathbf{x}\|=1, \mathbf{x} \in \mathrm{ColSpan}(\mathbf{B})} \|\mathbf{B}\mathbf{x}\|$$
$$\geq \min_{\|\mathbf{y}\|=1} \|\mathbf{A}\mathbf{y}\| \cdot \min_{\|\mathbf{x}\|=1, \mathbf{x} \in \mathrm{ColSpan}(\mathbf{B})} \|\mathbf{B}\mathbf{x}\|$$
$$= \sigma_{\min}(\mathbf{A})\sigma_{\min}(\mathbf{B}),$$

where $(a)$ is by changing of variables, i.e., $\mathbf{y} = \mathbf{B}\mathbf{x}/\|\mathbf{B}\mathbf{x}\|$. $\qquad\square$

**Lemma F.5 (Theorem 2.2.1 of (Chikuse, 2012))** *If $\mathbf{Z} \in \mathbb{R}^{m \times r}$ has entries drawn i.i.d. from Gaussian distribution $\mathcal{N}(0, 1)$, then $\mathbf{X} = \mathbf{Z}(\mathbf{Z}^\top \mathbf{Z})^{-1/2}$ is a random matrix uniformly distributed on $\mathsf{St}(m, r)$.*

**Lemma F.6 (Vershynin, 2010)** *If $\mathbf{Z} \in \mathbb{R}^{m \times r}$ is a matrix whose entries are independently drawn from $\mathcal{N}(0, 1)$. Then for every $\tau \geq 0$, with probability at least $1 - \exp(-\tau^2/2)$, we have*

$$\sigma_1(\mathbf{Z}) \leq \sqrt{m} + \sqrt{r} + \tau.$$

**Lemma F.7 (Rudelson & Vershynin, 2009)** *If $\mathbf{Z} \in \mathbb{R}^{m \times r}$ is a matrix whose entries are independently drawn from $\mathcal{N}(0, 1)$. Suppose that $m \geq r$. Then for every $\tau \geq 0$, we have for two universal constants $C_1 > 0$ and $C_2 > 0$ that*

$$\mathbb{P}\left(\sigma_r(\mathbf{Z}) \leq \tau(\sqrt{m} - \sqrt{r - 1})\right) \leq (C_1\tau)^{m-r+1} + \exp(-C_2 m).$$

**Lemma F.8** *If $\mathbf{U} \in \mathsf{St}(m, r_A)$ is a fixed matrix, $\mathbf{X} \in \mathsf{St}(m, r)$ is uniformly sampled from $\mathsf{St}(m, r)$ using methods described in Lemma F.5, and $r > r_A$, then we have that with probability at least $1 - \exp(-m/2) - (C_1\tau)^{r-r_A+1} - \exp(-C_2 r)$,*

$$\sigma_{r_A}(\mathbf{U}^\top \mathbf{X}) \geq \frac{\tau(r - r_A + 1)}{6\sqrt{mr}}.$$

*Proof.* Since $\mathbf{X} \in \mathsf{St}(m, r)$ is uniformly sampled from $\mathsf{St}(m, r)$ using methods described in Lemma F.5, we can write $\mathbf{X} = \mathbf{Z}(\mathbf{Z}^\top \mathbf{Z})^{-1/2}$, where $\mathbf{Z} \in \mathbb{R}^{m \times r}$ has entries i.i.d. sampled from $\mathcal{N}(0, 1)$. We thus have

$$\sigma_{r_A}(\mathbf{U}^\top \mathbf{X}) = \sigma_{r_A}\big(\mathbf{U}^\top \mathbf{Z}(\mathbf{Z}^\top \mathbf{Z})^{-1/2}\big).$$

We now consider $\mathbf{U}^\top \mathbf{Z} \in \mathbb{R}^{r_A \times r}$. It is clear that the entries of $\mathbf{U}^\top \mathbf{Z}$ are also i.i.d $\mathcal{N}(0, 1)$ random variables. As a consequence of Lemma F.7, we have that with probability at least $1 - (C_1\tau)^{r-r_A+1} - \exp(-C_2 r)$,

$$\sigma_{r_A}\big(\mathbf{U}^\top \mathbf{Z}\big) \geq \tau(\sqrt{r} - \sqrt{r_A - 1}).$$

We also have from Lemma F.6 that with probability at least $1 - \exp(-m/2)$,

$$\sigma_1(\mathbf{Z}^\top \mathbf{Z}) = \sigma_1^2(\mathbf{Z}) \leq (2\sqrt{m} + \sqrt{r})^2.$$

Taking union bound, we have with probability at least $1 - \exp(-m/2) - (C_1\tau)^{r-r_A+1} - \exp(-C_2 r)$,

$$\sigma_{r_A}(\mathbf{U}^\top \mathbf{X}) \overset{(a)}{\geq} \frac{\sigma_{r_A}\big(\mathbf{U}^\top \mathbf{Z}\big)}{\sigma_1(\mathbf{Z})} = \frac{\tau(\sqrt{r} - \sqrt{r_A - 1})}{2\sqrt{m} + \sqrt{r}} \geq \frac{\tau(r - r_A + 1)}{3\sqrt{m} \cdot 2\sqrt{r}} = \frac{\tau(r - r_A + 1)}{6\sqrt{mr}},$$

where $(a)$ comes from Lemma F.4. $\qquad\square$

**Lemma F.9** *Suppose $\mathbf{\Theta}_t \in \mathbb{S}^r$. Then the update rule (10) guarantees that $\mathbf{\Theta}_{t+1}$ also belongs to $\mathbb{S}^r$.*

*Proof.* From the update rule, we have that

$$\mathbf{\Theta}_{t+1} = \mathbf{X}_{t+1}^\top \mathbf{A} \mathbf{X}_{t+1} - \mathbf{X}_{t+1}^\top \big[(\mathcal{M}^*\mathcal{M} - \frac{\mu}{2}\mathcal{I})(\mathbf{X}_{t+1} \mathbf{\Theta}_t \mathbf{X}_{t+1}^\top - \mathbf{A})\big] \mathbf{X}_{t+1}.$$

Since $\mathbf{\Theta}_t \in \mathbb{S}^r$ and $\mathbf{A} \in \mathbb{S}^m$, it follows that $\mathbf{X}_{t+1} \mathbf{\Theta}_t \mathbf{X}_{t+1}^\top - \mathbf{A} \in \mathbb{S}^m$ and $\mathbf{X}_{t+1}^\top \mathbf{A} \mathbf{X}_{t+1} \in \mathbb{S}^r$.

By definition of $\mathcal{M}$ and $\mathcal{M}^*$, the composition $\mathcal{M}^*\mathcal{M}$ defines a self-adjoint operator in $\mathbb{S}^m$. Hence,

$$\mathbf{X}_{t+1}^\top \big[(\mathcal{M}^*\mathcal{M} - \frac{\mu}{2}\mathcal{I})(\mathbf{X}_{t+1} \mathbf{\Theta}_t \mathbf{X}_{t+1}^\top - \mathbf{A})\big] \mathbf{X}_{t+1} \in \mathbb{S}^r.$$

Thus, $\mathbf{\Theta}_{t+1} \in \mathbb{S}^r$, which completes the proof. $\qquad\square$

**Lemma F.10** *Let $\mathcal{M}(\cdot) : \mathbb{S}^m \to \mathbb{R}^n$ be a linear mapping that is $(r + r', \delta)$-RIP with $\delta \in [0, 1)$. Then for any symmetric matrix $\mathbf{Z}$ of rank at most $r$ and any symmetric matrix $\mathbf{Y}$ of rank at most $r'$, we have that*

$$|\langle (\mathcal{M}^*\mathcal{M} - \mathcal{I})(\mathbf{Z}), \mathbf{Y} \rangle| \leq \delta \|\mathbf{Z}\|_\mathsf{F} \|\mathbf{Y}\|_\mathsf{F}.$$

*Proof.* Denote $\Delta(\mathbf{Z}, \mathbf{Y}) := \langle (\mathcal{M}^*\mathcal{M} - \mathcal{I})(\mathbf{Z}), \mathbf{Y} \rangle = \langle \mathcal{M}(\mathbf{Z}), \mathcal{M}(\mathbf{Y}) \rangle - \langle \mathbf{Z}, \mathbf{Y} \rangle$. The above inequality trivially holds when $\|\mathbf{Z}\|_\mathsf{F} = 0$ or $\|\mathbf{Y}\|_\mathsf{F} = 0$. Without loss of generality, we assume that $\|\mathbf{Z}\|_\mathsf{F} \neq 0$ and $\|\mathbf{Y}\|_\mathsf{F} \neq 0$. Define $\tilde{\mathbf{Z}} := \frac{\mathbf{Z}}{\|\mathbf{Z}\|_\mathsf{F}}$ and $\tilde{\mathbf{Y}} := \frac{\mathbf{Y}}{\|\mathbf{Y}\|_\mathsf{F}}$. It then follows that

$$\Delta(\mathbf{Z}, \mathbf{Y}) = \Delta(\tilde{\mathbf{Z}}, \tilde{\mathbf{Y}}) \cdot \|\mathbf{Z}\|_\mathsf{F} \|\mathbf{Y}\|_\mathsf{F}.$$

Using the polarization identity, we obtain

$$\langle \mathcal{M}(\tilde{\mathbf{Z}}), \mathcal{M}(\tilde{\mathbf{Y}}) \rangle = \frac{1}{4}(\|\mathcal{M}(\tilde{\mathbf{Z}} + \tilde{\mathbf{Y}})\|^2 - \|\mathcal{M}(\tilde{\mathbf{Z}} - \tilde{\mathbf{Y}})\|^2),$$

$$\langle \tilde{\mathbf{Z}}, \tilde{\mathbf{Y}} \rangle = \frac{1}{4}(\|\tilde{\mathbf{Z}} + \tilde{\mathbf{Y}}\|_\mathsf{F}^2 - \|\tilde{\mathbf{Z}} - \tilde{\mathbf{Y}}\|_\mathsf{F}^2).$$

Substituting the two equalities into the expression of $\Delta(\tilde{\mathbf{Z}}, \tilde{\mathbf{Y}})$, we have that

$$
\begin{aligned}
|\Delta(\tilde{\mathbf{Z}}, \tilde{\mathbf{Y}})| &= |\langle \mathcal{M}(\tilde{\mathbf{Z}}), \mathcal{M}(\tilde{\mathbf{Y}}) \rangle - \langle \tilde{\mathbf{Z}}, \tilde{\mathbf{Y}} \rangle| \\
&= \frac{1}{4}|(\|\mathcal{M}(\tilde{\mathbf{Z}} + \tilde{\mathbf{Y}})\|^2 - \|\mathcal{M}(\tilde{\mathbf{Z}} - \tilde{\mathbf{Y}})\|^2) - (\|\tilde{\mathbf{Z}} + \tilde{\mathbf{Y}}\|_{\mathsf{F}}^2 - \|\tilde{\mathbf{Z}} - \tilde{\mathbf{Y}}\|_{\mathsf{F}}^2)| \\
&\leq \frac{1}{4}(|\|\mathcal{M}(\tilde{\mathbf{Z}} + \tilde{\mathbf{Y}})\|^2 - \|\tilde{\mathbf{Z}} + \tilde{\mathbf{Y}}\|_{\mathsf{F}}^2| + |\|\mathcal{M}(\tilde{\mathbf{Z}} - \tilde{\mathbf{Y}})\|^2 - \|\tilde{\mathbf{Z}} - \tilde{\mathbf{Y}}\|_{\mathsf{F}}^2|) \\
&\overset{(a)}{\leq} \frac{\delta}{4}(\|\tilde{\mathbf{Z}} + \tilde{\mathbf{Y}}\|_{\mathsf{F}}^2 + \|\tilde{\mathbf{Z}} - \tilde{\mathbf{Y}}\|_{\mathsf{F}}^2) \\
&= \frac{\delta}{2}(\|\tilde{\mathbf{Z}}\|_{\mathsf{F}}^2 + \|\tilde{\mathbf{Y}}\|_{\mathsf{F}}^2) \\
&= \delta,
\end{aligned}
$$

where $(a)$ is from the facts that $\mathcal{M}(\cdot)$ is $(r + r', \delta)$-RIP with constant $\delta$, $\mathsf{rank}(\tilde{\mathbf{Z}} + \tilde{\mathbf{Y}}) \leq \mathsf{rank}(\tilde{\mathbf{Z}}) + \mathsf{rank}(\tilde{\mathbf{Y}}) \leq r + r'$, and $\mathsf{rank}(\tilde{\mathbf{Z}} - \tilde{\mathbf{Y}}) \leq \mathsf{rank}(\tilde{\mathbf{Z}}) + \mathsf{rank}(\tilde{\mathbf{Y}}) \leq r + r'$. Therefore, we have that

$$
|\langle (\mathcal{M}^*\mathcal{M} - \mathcal{I})(\mathbf{Z}), \mathbf{Y} \rangle| = |\Delta(\mathbf{Z}, \mathbf{Y})| = |\Delta(\tilde{\mathbf{Z}}, \tilde{\mathbf{Y}}) \cdot \|\mathbf{Z}\|_{\mathsf{F}}\|\mathbf{Y}\|_{\mathsf{F}} \leq \delta\|\mathbf{Z}\|_{\mathsf{F}}\|\mathbf{Y}\|_{\mathsf{F}},
$$

which completes the proof.

$\square$

**Lemma F.11 (Lemma 7.3 of (Stöger & Soltanolkotabi, 2021))** *Let $\mathcal{M}(\cdot) : \mathbb{S}^m \to \mathbb{R}^n$ be a linear mapping that is $(r + r_A + 1, \delta)$-RIP with $\delta \in [0, 1)$, then $\|(\mathcal{M}^*\mathcal{M} - \mathcal{I})(\mathbf{A})\| \leq \delta\|\mathbf{A}\|_{\mathsf{F}}$ for all matrices $\mathbf{A} \in \mathbb{S}^m$ of rank at most $r + r_A$.*

*Proof.* By Lemma F.10, if $\mathbf{A} \in \mathbb{S}^m$ has rank at most $r + r_A$ and $\mathbf{Y} \in \mathbb{S}^m$ has rank at most 1, then it holds that

$$
|\langle (\mathcal{M}^*\mathcal{M} - \mathcal{I})(\mathbf{A}), \mathbf{Y} \rangle| \leq \delta\|\mathbf{A}\|_{\mathsf{F}}\|\mathbf{Y}\|_{\mathsf{F}}.
$$

Hence, it suffices to prove that there exists a matrix $\mathbf{Y}$ of rank 1, such that $|\langle (\mathcal{M}^*\mathcal{M} - \mathcal{I})(\mathbf{A}), \mathbf{Y} \rangle| = \|(\mathcal{M}^*\mathcal{M} - \mathcal{I})(\mathbf{A})\|$ and $\|\mathbf{Y}\|_{\mathsf{F}} \leq 1$. Since $(\mathcal{M}^*\mathcal{M} - \mathcal{I})(\mathbf{A})$ is a symmetric matrix, it follows that

$$
\begin{aligned}
\|(\mathcal{M}^*\mathcal{M} - \mathcal{I})(\mathbf{A})\| &= \max_{\|\mathbf{u}\|=1} \mathbf{u}^\top (\mathcal{M}^*\mathcal{M} - \mathcal{I})(\mathbf{A})\mathbf{u} \\
&= \max_{\|\mathbf{u}\|=1} \mathsf{Tr}((\mathcal{M}^*\mathcal{M} - \mathcal{I})(\mathbf{A})\mathbf{u}\mathbf{u}^\top) \\
&= \max_{\|\mathbf{u}\|=1} \langle (\mathcal{M}^*\mathcal{M} - \mathcal{I})(\mathbf{A}), \mathbf{u}\mathbf{u}^\top \rangle.
\end{aligned}
$$

Let $\mathbf{Y} = \tilde{\mathbf{u}}\tilde{\mathbf{u}}^\top$, where $\tilde{\mathbf{u}} \in \arg\max_{\|\mathbf{u}\|=1} \langle (\mathcal{M}^*\mathcal{M} - \mathcal{I})(\mathbf{A}), \mathbf{u}\mathbf{u}^\top \rangle$. We then have that $\mathsf{rank}(\mathbf{Y}) = 1$, $\mathbf{Y} \in \mathbb{S}^m$, $|\langle (\mathcal{M}^*\mathcal{M} - \mathcal{I})(\mathbf{A}), \mathbf{Y} \rangle| = \|(\mathcal{M}^*\mathcal{M} - \mathcal{I})(\mathbf{A})\|$, and $\|\mathbf{Y}\|_{\mathsf{F}} \leq 1$. $\square$

**Lemma F.12** *Let $\mathbf{A} \in \mathbb{R}^{n \times m}$, $\mathbf{B} \in \mathbb{R}^{m \times n}$ be two real matrices, then the following inequality holds*

$$
\mathbf{A}\mathbf{B} + \mathbf{B}^\top\mathbf{A}^\top \preceq 2\|\mathbf{A}\|\|\mathbf{B}\|\mathbf{I}_n.
$$

*Proof.* For any unit vector $\boldsymbol{x} \in \mathbb{R}^n$ with $\|\boldsymbol{x}\| = 1$, we can obtain that

$$
\boldsymbol{x}^\top(\mathbf{A}\mathbf{B} + \mathbf{B}^\top\mathbf{A}^\top)\boldsymbol{x} = \boldsymbol{x}^\top\mathbf{A}\mathbf{B}\boldsymbol{x} + \boldsymbol{x}^\top\mathbf{B}^\top\mathbf{A}^\top\boldsymbol{x} \overset{(a)}{=} 2\boldsymbol{x}^\top\mathbf{A}\mathbf{B}\boldsymbol{x},
$$

where $(a)$ is from the fact that $\boldsymbol{x}^\top\mathbf{B}^\top\mathbf{A}^\top\boldsymbol{x}$ is a scalar. By the Cauchy–Schwarz inequality and the definition of the spectral norm, we have that

$$
|\boldsymbol{x}^\top\mathbf{A}\mathbf{B}\boldsymbol{x}| \leq \|\mathbf{A}\mathbf{B}\boldsymbol{x}\| \cdot \|\boldsymbol{x}\| \leq \|\mathbf{A}\| \cdot \|\mathbf{B}\| \cdot \|\boldsymbol{x}\|^2 = \|\mathbf{A}\|\|\mathbf{B}\|.
$$

Hence, we obtain the following inequality:

$$
\boldsymbol{x}^\top(\mathbf{A}\mathbf{B} + \mathbf{B}^\top\mathbf{A}^\top)\boldsymbol{x} \leq 2\|\mathbf{A}\|\|\mathbf{B}\|.
$$

Since this holds for any unit vector $\boldsymbol{x}$, it follows that

$$
\mathbf{A}\mathbf{B} + \mathbf{B}^\top\mathbf{A}^\top \preceq 2\|\mathbf{A}\|\|\mathbf{B}\|\mathbf{I}_n.
$$

$\square$

**Lemma F.13** *Let $t \geq 1$ be a positive integer. For all real numbers $x$, satisfying $0 \leq x \leq \frac{1}{t}$, the following inequality holds:*

$$(1+x)^t \leq 1 + 3tx.$$

*Proof.* Let $f(x) := 1 + 3tx - (1+x)^t$, $x \in [0, \frac{1}{t}]$. Then, for all $x \in [0, \frac{1}{t}]$, we obtain

$$f'(x) = 3t - t(1+x)^{t-1} \geq 3t - t(1+\frac{1}{t})^{t-1} \geq (3-e)t > 0.$$

Therefore, for all $x \in [0, \frac{1}{t}]$, $f(x) \geq f(0) = 0$, which means $(1+x)^t \leq 1 + 3tx$ for all $x \in [0, \frac{1}{t}]$.
$\square$

**Lemma F.14** *Let $k \in \mathbb{R}_{\geq 1}$, $q \in (\frac{1}{2}, 1)$. Suppose that sequences $\{a_t\}_{t=0}^{\infty}, \{b_t\}_{t=0}^{\infty} \subset \mathbb{R}_{\geq 0}$ satisfy*

$$b_{t+1} \leq qb_t + \frac{1-q}{180k^2}\left(a_{t-1}^2 + a_t^2 + 2a_{t-1}a_t + \sqrt{b_t}(a_{t-1} + a_t)\right), \tag{38}$$

$$a_t \leq 2k\sqrt{b_t} + \frac{1}{6}a_{t-1}, \quad t = 1, 2 \ldots, \tag{39}$$

*and another pair of sequences $\{\tilde{a}_t\}_{t=0}^{\infty}, \{\tilde{b}_t\}_{t=0}^{\infty} \subset \mathbb{R}_{\geq 0}$ satisfy*

$$\tilde{b}_{t+1} = q\tilde{b}_t + \frac{1-q}{180k^2}\left(\tilde{a}_{t-1}^2 + \tilde{a}_t^2 + 2\tilde{a}_{t-1}\tilde{a}_t + \sqrt{\tilde{b}_t}(\tilde{a}_{t-1} + \tilde{a}_t)\right), \tag{40}$$

$$\tilde{a}_t = 2k\sqrt{\tilde{b}_t} + \frac{1}{6}\tilde{a}_{t-1}, \quad t = 1, 2 \ldots. \tag{41}$$

*If the initial conditions satisfy*

$$a_0 \leq \tilde{a}_0, b_0 \leq \tilde{b}_0, b_1 \leq \tilde{b}_1,$$

*then $a_t \leq \tilde{a}_t$ and $b_t \leq \tilde{b}_t$ hold for all $t \geq 0$.*

*Proof.* We proceed by mathematical induction. From inequality (38), we obtain

$$a_1 \leq 2k\sqrt{b_1} + \frac{1}{6}a_0$$
$$\overset{(a)}{\leq} 2k\sqrt{\tilde{b}_1} + \frac{1}{6}\tilde{a}_0$$
$$\overset{(b)}{=} \tilde{a}_1,$$

where $(a)$ is by initial conditions; and $(b)$ is from equality (40). Analogously, inequality (39) implies

$$b_2 \leq qb_1 + \frac{1-q}{180k^2}\left(a_0^2 + a_1^2 + 2a_0a_1 + \sqrt{b_1}(a_0 + a_1)\right)$$
$$\overset{(c)}{\leq} q\tilde{b}_1 + \frac{1-q}{180k^2}\left(\tilde{a}_0^2 + \tilde{a}_1^2 + 2\tilde{a}_0\tilde{a}_1 + \sqrt{\tilde{b}_1}(\tilde{a}_0 + \tilde{a}_1)\right)$$
$$\overset{(d)}{=} \tilde{b}_2,$$

where $(c)$ is due to initial conditions and $a_1 \leq \tilde{a}_1$; and $(d)$ is by equality (41). By induction, we conclude that $a_t \leq \tilde{a}_t$ and $b_t \leq \tilde{b}_t$ for all $t \geq 0$, which completes the proof. $\square$

**Lemma F.15** *Let $k \in \mathbb{R}_{\geq 1}$, $q \in (\frac{1}{2}, 1)$. Suppose that sequences $\{\tilde{a}_t\}_{t=0}^{\infty}, \{\tilde{b}_t\}_{t=0}^{\infty} \subset \mathbb{R}_{\geq 0}$ satisfy:*

$$\tilde{b}_{t+1} = q\tilde{b}_t + \frac{1-q}{180k^2}\left(\tilde{a}_{t-1}^2 + \tilde{a}_t^2 + 2\tilde{a}_{t-1}\tilde{a}_t + \sqrt{\tilde{b}_t}(\tilde{a}_{t-1} + \tilde{a}_t)\right), \tag{42}$$

$$\tilde{a}_t = 2k\sqrt{\tilde{b}_t} + \frac{1}{6}\tilde{a}_{t-1}, \quad t = 1, 2 \ldots. \tag{43}$$

*If the initial conditions satisfy*

$$\tilde{a}_0, \tilde{a}_1, \tilde{b}_0, \tilde{b}_1 \in \mathbb{R}_{\geq 0}, \tilde{a}_0 \leq 3k\sqrt{\tilde{b}_0} \leq \frac{3\sqrt{2}k}{\sqrt{1+q}}\sqrt{\tilde{b}_1}, \tilde{a}_1 \leq 3k\sqrt{\tilde{b}_1},$$

*then we have that $\tilde{a}_t \leq 3k\sqrt{\tilde{b}_0}\left(\frac{1+q}{2}\right)^{t/2}$ for all $t \geq 0$.*

*Proof.* We proceed by mathematical induction. We first consider the following auxiliary system:

$$\hat{b}_{t+1} = \max\{q\hat{b}_t + \frac{1-q}{180k^2}\big(\hat{a}_{t-1}^2 + \hat{a}_t^2 + 2\hat{a}_{t-1}\hat{a}_t + \sqrt{\hat{b}_t}(\hat{a}_{t-1} + \hat{a}_t)\big), \frac{1+q}{2}\hat{b}_t\}, \quad (44)$$

$$\hat{a}_t = \max\{2k\sqrt{\hat{b}_t} + \frac{1}{6}\hat{a}_{t-1}, 3k\sqrt{\hat{b}_t}\}, \quad t = 1, 2, \ldots. \quad (45)$$

Let $\hat{a}_0 = \tilde{a}_0, \hat{b}_0 = \tilde{b}_0$, and $\hat{b}_1 = \tilde{b}_1$. It holds that $\hat{a}_0 \leq 3k\sqrt{\hat{b}_0} \leq \frac{3\sqrt{2}k}{\sqrt{1+q}}\sqrt{\hat{b}_1}$, and thus we have

$$2k\sqrt{\hat{b}_1} + \frac{1}{6}\hat{a}_0 \leq 2k\sqrt{\hat{b}_1} + \frac{\sqrt{2}k}{2\sqrt{1+q}}\sqrt{\hat{b}_1} \leq 3k\sqrt{\hat{b}_1}.$$

From equality (45) at $t = 1$, we obtain $\hat{a}_1 = 3k\sqrt{\hat{b}_1}$. Since $\hat{a}_0 \leq \frac{3\sqrt{2}k}{\sqrt{1+q}}\sqrt{\hat{b}_1}$ and $\hat{a}_1 = 3k\sqrt{\hat{b}_1}$, it follows that

$$q\hat{b}_1 + \frac{1-q}{180k^2}\big(\hat{a}_0^2 + \hat{a}_1^2 + 2\hat{a}_0\hat{a}_1 + \sqrt{\hat{b}_1}(\hat{a}_0 + \hat{a}_1)\big)$$

$$\leq q\hat{b}_1 + \frac{1-q}{180k^2}\big(\frac{18k^2}{1+q}\hat{b}_1 + 9k^2\hat{b}_1 + \frac{18\sqrt{2}k^2}{\sqrt{1+q}}\hat{b}_1 + \frac{3\sqrt{2}k}{\sqrt{1+q}}\hat{b}_1 + 3k\hat{b}_1\big)$$

$$\leq q\hat{b}_1 + \frac{1-q}{180k^2}\big(18k^2 + 9k^2 + 18\sqrt{2}k^2 + 3\sqrt{2}k^2 + 3k^2\big)\hat{b}_1$$

$$\leq q\hat{b}_1 + \frac{1-q}{2}\hat{b}_1$$

$$\leq \frac{1+q}{2}\hat{b}_1.$$

From equality (44) at $t = 1$, we have $\hat{b}_2 = \frac{1+q}{2}\hat{b}_1$. Using the same reasoning for $t = 2$ yields

$$2k\sqrt{\hat{b}_2} + \frac{1}{6}\hat{a}_1 = 2k\sqrt{\hat{b}_2} + \frac{k}{2}\sqrt{\hat{b}_1} = 2k\sqrt{\hat{b}_2} + \frac{k}{2}\sqrt{\frac{2}{1+q}}\sqrt{\hat{b}_2} \leq 3k\sqrt{\hat{b}_2}.$$

Equality (45) at $t = 1$ implies that $\hat{a}_2 = 3k\sqrt{\hat{b}_2}$. Since $\hat{a}_1 = 3k\sqrt{\hat{b}_1}$ and $\hat{a}_2 = 3k\sqrt{\hat{b}_2}$, we obtain

$$q\hat{b}_2 + \frac{1-q}{180k^2}\big(\hat{a}_1^2 + \hat{a}_2^2 + 2\hat{a}_1\hat{a}_2 + \sqrt{\hat{b}_2}(\hat{a}_1 + \hat{a}_2)\big)$$

$$= q\hat{b}_2 + \frac{1-q}{180k^2}\big(9k^2\hat{b}_1 + 9k^2\hat{b}_2 + 18k^2\sqrt{\hat{b}_1\hat{b}_2} + 3k(\sqrt{\hat{b}_1\hat{b}_2} + \hat{b}_2)\big)$$

$$= q\hat{b}_2 + \frac{1-q}{180k^2}\big(\frac{18k^2}{1+q} + 9k^2 + \frac{18\sqrt{2}k^2}{\sqrt{1+q}} + \frac{3\sqrt{2}k}{\sqrt{1+q}} + 3k\big)\hat{b}_2$$

$$\leq q\hat{b}_2 + \frac{1-q}{180k^2}\big(18k^2 + 9k^2 + 18\sqrt{2}k^2 + 3\sqrt{2}k^2 + 3k^2\big)\hat{b}_2$$

$$\leq q\hat{b}_2 + \frac{1-q}{2}\hat{b}_2$$

$$\leq \frac{1+q}{2}\hat{b}_2.$$

Applying equality (44) at $t = 2$, $\hat{b}_3 = \frac{1+q}{2}\hat{b}_2$ is derived. Therefore, we have that $\hat{a}_1 = 3k\sqrt{\hat{b}_1}$, $\hat{a}_2 = 3k\sqrt{\hat{b}_2}$, and $\hat{b}_3 = \frac{1+q}{2}\hat{b}_2$. Assume that $\hat{a}_{t-1} = 3k\sqrt{\hat{b}_{t-1}}$, $\hat{a}_t = 3k\sqrt{\hat{b}_t}$, and $\hat{b}_{t+1} = \frac{1+q}{2}\hat{b}_t$, we claim that $\hat{a}_{t+1} = 3k\sqrt{\hat{b}_{t+1}}$ and $\hat{b}_{t+2} = \frac{1+q}{2}\hat{b}_{t+1}$. From equality (45), we obtain

$$\hat{a}_{t+1} = \max\{2k\sqrt{\hat{b}_{t+1}} + \frac{1}{6}\hat{a}_t, 3k\sqrt{\hat{b}_{t+1}}\}$$

$$= \max\{2k\sqrt{\hat{b}_{t+1}} + \frac{1}{2}k\sqrt{\hat{b}_t}, 3k\sqrt{\hat{b}_{t+1}}\}$$

$$= \max\{2k\sqrt{\hat{b}_{t+1}} + \frac{k\sqrt{\frac{2}{1+q}}}{2}\sqrt{\hat{b}_{t+1}}, 3k\sqrt{\hat{b}_{t+1}}\}$$

$$= 3k\sqrt{\hat{b}_{t+1}}.$$

Analogously, equality (44) implies that

$$\hat{b}_{t+2} = \max\{q\hat{b}_{t+1} + \frac{1-q}{180k^2}(\hat{a}_t^2 + \hat{a}_{t+1}^2 + 2\hat{a}_t\hat{a}_{t+1} + \sqrt{\hat{b}_{t+1}}(\hat{a}_t + \hat{a}_{t+1})), \frac{1+q}{2}\hat{b}_{t+1}\}$$

$$= \max\{q\hat{b}_{t+1} + \frac{1-q}{180k^2}(9k^2\hat{b}_t + 9k^2\hat{b}_{t+1} + 18k^2\sqrt{\hat{b}_t\hat{b}_{t+1}} + \sqrt{\hat{b}_{t+1}}(3k\sqrt{\hat{b}_t} + 3k\sqrt{\hat{b}_{t+1}})),$$

$$\frac{1+q}{2}\hat{b}_{t+1}\}$$

$$= \max\{q\hat{b}_{t+1} + \frac{1-q}{20}(\frac{2}{1+q} + 1 + 2(\frac{2}{1+q})^{1/2} + \frac{1}{3k}(\frac{2}{1+q})^{1/2} + \frac{1}{3k})\hat{b}_{t+1}, \frac{1+q}{2}\hat{b}_{t+1}\}$$

$$= \frac{1+q}{2}\hat{b}_{t+1}.$$

Therefore, we have that $\{\hat{b}_t\}_{t=0}^{t=\infty}$ decreases in a linear rate and that $\hat{a}_t = 3k\sqrt{\hat{b}_t}$ in the system (44) and (45), which means that $\hat{a}_t \leq 3k\sqrt{\hat{b}_0}\left(\frac{1+q}{2}\right)^{t/2} = 3k\sqrt{\hat{b}_0}\left(\frac{1+q}{2}\right)^{t/2}$ for all $t \geq 0$.

We now prove that $\tilde{a}_t \leq \hat{a}_t, \tilde{b}_t \leq \hat{b}_t$ for all $t \geq 0$. Obviously, $\tilde{a}_0 \leq \hat{a}_0, \tilde{a}_1 \leq \hat{a}_1, \tilde{b}_0 \leq \hat{b}_0$, and $\tilde{b}_1 \leq \hat{b}_1$ hold. Applying equality (42) at $t = 1$ and equality (43) at $t = 2$, we obtain

$$\tilde{b}_2 = q\tilde{b}_1 + \frac{1-q}{180k^2}(\tilde{a}_0^2 + \tilde{a}_1^2 + 2\tilde{a}_0\tilde{a}_1 + \sqrt{\tilde{b}_1}(\tilde{a}_0 + \tilde{a}_1))$$

$$\leq q\hat{b}_1 + \frac{1-q}{180k^2}(\hat{a}_0^2 + \hat{a}_1^2 + 2\hat{a}_0\hat{a}_1 + \sqrt{\hat{b}_1}(\hat{a}_0 + \hat{a}_1))$$

$$\leq \max\{q\hat{b}_1 + \frac{1-q}{180k^2}(\hat{a}_0^2 + \hat{a}_1^2 + 2\hat{a}_0\hat{a}_1 + \sqrt{\hat{b}_1}(\hat{a}_0 + \hat{a}_1)), \frac{1+q}{2}\hat{b}_1\}$$

$$= \hat{b}_2,$$

$$\tilde{a}_2 = 2k\sqrt{\tilde{b}_2} + \frac{1}{6}\tilde{a}_1$$

$$\leq 2k\sqrt{\hat{b}_2} + \frac{1}{6}\hat{a}_1$$

$$\leq \max\{2k\sqrt{\hat{b}_2} + \frac{1}{6}\hat{a}_1, 3k\sqrt{\hat{b}_2}\}$$

$$= \hat{a}_2.$$

Hence, $\tilde{a}_2 \leq \hat{a}_2, \tilde{b}_2 \leq \hat{b}_2$ and recursively, we can obtain $\tilde{a}_t \leq \hat{a}_t, \tilde{b}_t \leq \hat{b}_t$ for all $t \geq 0$. Consequently, $\{\tilde{a}_t\}_{t=0}^{t=\infty}$ achieves at least a linear convergence rate in the system (42) and (43), which means that $\tilde{a}_t \leq 3k\sqrt{\tilde{b}_0}\left(\frac{1+q}{2}\right)^{t/2}$ for all $t \geq 0$.

$\square$

## G    EXPERIMENTAL SETUPS

In this section, we provide experimental setups in detail.

### G.1    SETUP FOR FIGURE 1

We apply Algorithm 1 to problem (2) and study the trajectory generated by the algorithm.

In this experiment, we set $m = 300, r = 10, r_A = 5$, and $\kappa = 3$. The ground-truth matrix $\mathbf{A} \in \mathbb{R}^{m \times m}$ is constructed as $\mathbf{A} = \mathbf{U}\boldsymbol{\Sigma}\mathbf{U}^\top$, where $\mathbf{U} \in \mathbb{R}^{m \times r_A}$ is a random orthonormal matrix and $\boldsymbol{\Sigma} \in \mathbb{S}_+^{r_A}$ is diagonal with entries generated by a power spacing scheme. Specifically, the $j$-th entry of $\boldsymbol{\Sigma}$ is given by $\sigma_j = \kappa^{-(\frac{j-1}{r_A-1})^p}$ for $j = 1, \ldots, r_A$, where we set $p = 0.6$.

We generate $n = 50000$ independent feature matrices $\{\mathbf{M}_i\}_{i=1}^n \subset \mathbb{S}^m$ in the following manner. For each $i \in \{1, \ldots, n\}$, we sample $\mathbf{R}_i \in \mathbb{R}^{m \times m}$ with i.i.d. standard Gaussian entries and define $\mathbf{M}_i = \frac{1}{2\sqrt{n}}(\mathbf{R}_i + \mathbf{R}_i^\top)$, which ensures the symmetry of $\mathbf{M}_i$.

We initialize $\mathbf{X}_0 = \mathbf{Z}_0(\mathbf{Z}_0^\top \mathbf{Z}_0)^{-1/2}$ and $\mathbf{\Theta}_0 = 0.5\mathbf{I}_r$, where $\mathbf{Z}_0 \in \mathbb{R}^{m \times r}$ has i.i.d. standard Gaussian entries. This initialization ensures that $\mathbf{X}_0$ lies on the manifold $\mathsf{St}(m, r)$ and $\mathbf{\Theta}_0 \in \mathbb{S}^r$. For this experiment, we set the stepsizes to $\eta = 0.2$ and $\mu = 2$.

## G.2    SETUP FOR SECTION 5

### G.2.1    SETUP FOR THE EXPERIMENTS WITH SYNTHETIC DATA

We apply Algorithm 1 to problem (2) and compare its convergence with standard GD applied to (1).

In the noisy measurement experiment, we set $m = 10, r = 5, r_A = 3$, and $n = 1000$, while for other experiments, the corresponding values are given in the main text. In these settings, the ground truth matrix $\mathbf{A} \in \mathbb{R}^{m \times m}$ is formed as $\mathbf{A} = \mathbf{U}\mathbf{\Sigma}\mathbf{U}^\top$, where $\mathbf{U} \in \mathbb{R}^{m \times r_A}$ is a random orthonormal matrix and $\mathbf{\Sigma} \in \mathbb{S}_+^{r_A}$ is a diagonal matrix with entries evenly distributed on a logarithmic scale in the interval $[1/\kappa, 1]$. Feature matrices $\{\mathbf{M}_i\}_{i=1}^n$ are generated as described in G.1.

For other experiments, we initialize $\mathbf{X}_0 = \mathbf{Z}_0(\mathbf{Z}_0^\top \mathbf{Z}_0)^{-1/2}$ and $\mathbf{\Theta}_0 = \mathbf{I}_r$ for Algorithm 1, where $\mathbf{Z}_0 \in \mathbb{R}^{m \times r}$ consists of i.i.d. standard Gaussian entries, and $\mathbf{X}_0^{\mathrm{GD}} = 0.1\mathbf{Z}_0$ for GD.

Throughout the experiments, we use stepsizes $\eta = 0.1$ and $\mu = 2$ for RGD and $\eta = 0.1$ for GD, except for the first experiment in Subsection 5.2. In that case, we set $\eta = 0.1, 0.12, 0.14$ and $\mu = 2$ for RGD, and $\eta = 0.1, 0.12, 0.14$ for GD, corresponding to $r = 50, 75, 100$, respectively.

### G.2.2    SETUP FOR IMAGE RECONSTRUCTION EXPERIMENTS

For the image reconstruction experiments, we conduct two setups: one based on recovering a CIFAR-10 image from linear measurements and the other on direct matrix sensing of a structured image.

For the CIFAR-10 experiment, we take the first horse image from CIFAR-10 dataset, convert it to grayscale, and vectorize it as $\boldsymbol{a} \in \mathbb{R}^{1024}$. The ground-truth matrix is set as $\mathbf{A} = \boldsymbol{a}\boldsymbol{a}^\top \in \mathbb{S}_+^{1024}$. The overparameterization level is set to $r = 100$, with $n = 50000$ feature matrices generated as in G.1. RGD is initialized with $\mathbf{X}_0 = \mathbf{Z}_0(\mathbf{Z}_0^\top \mathbf{Z}_0)^{-1/2}$ and $\mathbf{\Theta}_0 = \mathbf{I}_r$, where $\mathbf{Z}_0 \in \mathbb{R}^{m \times r}$ has i.i.d. standard Gaussian entries. GD uses small random initialization: $\mathbf{X}_0^{\mathrm{GD}} = 0.1\mathbf{Z}_0$. We run RGD for $t_{\mathrm{RGD}} = 100$ and GD for $t_{\mathrm{GD}} = 200$ iterations. For RGD, we adopt stepsizes of $\eta = 0.01$ for updating $\mathbf{X}$ and $\mu = 2$ for updating $\mathbf{\Theta}$. For GD, we apply stepsize $\eta = 0.01$ to update $\mathbf{X}$.

After optimization, following the approach of (Duchi et al., 2020, Section 4.1), we perform a rank-one truncated SVD on the recovered matrix $\hat{\mathbf{A}}$, and the estimate of the original signal is constructed as the leading singular vector multiplied by the square root of its corresponding singular value. The resulting vector is then reshaped into a $32 \times 32$ reconstruction image.

For the structured image experiment, we generate a grayscale matrix $\mathbf{A} \in \mathbb{S}_+^{128}$ of rank $r_A = 2$ using block-wave basis functions. Specifically, we construct $r_A$ one-dimensional signals of length 128, where each signal is a normalized block wave taking values in $\pm 1$ with a random period. Stacking these signals forms a matrix $\mathbf{U} \in \mathbb{R}^{128 \times r_A}$. The ground-truth image is defined as $\mathbf{A} = \mathbf{U}\mathbf{\Lambda}\mathbf{U}^\top$, where $\mathbf{\Lambda}$ is a $2 \times 2$ diagonal matrix with diagonal entries 1 and 0.9. This diagonal matrix assigns geometrically decaying weights to different block-wave modes.

We again fix $r = 100$ and use $n = 50000$ feature matrices generated as in G.1. Both RGD and GD are randomly initialized as above. We run RGD for $t_{\mathrm{RGD}} = 100$ and GD for $t_{\mathrm{GD}} = 200$ iterations. We adopt stepsizes of $\eta = 0.03$ and $\mu = 2$ in RGD and a stepsize of $\eta = 0.03$ in GD.

The per-iteration computational complexity of both RGD and GD is $\mathcal{O}(nm^2r)$, which is dominated by the operation of sensing. Since each RGD iteration requires performing two sensing operations while GD requires only one, we set the number of iterations as $t_{\mathrm{GD}} = 2t_{\mathrm{RGD}}$ to make the overall runtime roughly comparable between the two methods.

## G.3    SETUP FOR THE EXPERIMENTS OF PRECONDITIONED ALGORITHMS

We apply Algorithm 1 and PrecRGD to problem (2) and compare their convergence behavior with GD, PrecGD and ScaledGD($\lambda$) applied to problem (1).

In this experiment, we set $m = 50, r = 40, r_A = 3, \kappa = 10$, and $n = 8000$ for the first instance and increase $r$ to $45$ in the second instance while keeping all other parameters fixed. The ground-truth matrix $\mathbf{A} \in \mathbb{R}^{m \times m}$ is constructed as $\mathbf{A} = \mathbf{U \Sigma U}^\top$, where $\mathbf{U} \in \mathbb{R}^{m \times r_A}$ is a random orthonormal matrix and $\mathbf{\Sigma} \in \mathbb{S}_+^{r_A}$ is a diagonal matrix with entries evenly distributed on a logarithmic scale in the interval $[1/\kappa, 1]$. Feature matrices $\{\mathbf{M}_i\}_{i=1}^n$ are generated as described in G.1.

The regularization parameter $\lambda$ used in PrecRGD is chosen adaptively as $\lambda = \|\mathcal{M}^* \mathcal{M}(\mathbf{X}_t \mathbf{\Theta}_t \mathbf{X}_t^\top - \mathbf{A})\|_\infty$, where $\| \cdot \|_\infty$ denotes the matrix infinity norm. For PrecGD and ScaledGD($\lambda$), we set the regularization parameters as $\lambda = \|\mathcal{M}^* \mathcal{M}(\mathbf{Y}_t^\top \mathbf{Y}_t - \mathbf{A})\|_\infty$ and $\lambda = 0.1$, respectively.

We initialize $\mathbf{X}_0 = \mathbf{Z}_0(\mathbf{Z}_0^\top \mathbf{Z}_0)^{-1/2}$ and $\mathbf{\Theta}_0 = \mathbf{I}_r$ for RGD and PrecRGD, where $\mathbf{Z}_0 \in \mathbb{R}^{m \times r}$ consists of i.i.d. standard Gaussian entries. For GD and PrecGD, we use initialization $\mathbf{X}_0^{\text{GD}} = \mathbf{Z}_0$. For ScaledGD($\lambda$), we set $\mathbf{X}_0^{\text{Scaled}} = 10^{-6} \cdot \mathbf{Z}_0$. For RGD and PrecRGD, we adopt stepsizes of $\eta = 0.02$ for updating $\mathbf{X}$ and $\mu = 2$ for updating $\mathbf{\Theta}$. For GD, PrecGD and ScaledGD($\lambda$), we apply stepsize $\eta = 0.02$.

### G.4 SETUP FOR THE MATRIX COMPLETION PROBLEMS

We apply RGD to problem (7) and compare its convergence behavior with that of GD applied to (6).

We directly use the code provided in the supplementary material of (Yalçın et al., 2022), replacing only the objective function with our WN-based formulation (7). The update of $\mathbf{X}_t$ follows the same step-size strategy as that of $\mathbf{Y}_t$ in their implementation, and $\mathbf{\Theta}_t$ is updated using a fixed step size of $\mu = 2$. Except for these modifications, other settings are kept identical to the original code.

### G.5 SETUP FOR THE EXPERIMENT OF DIAGONAL AND NONNEGATIVE $\mathbf{\Theta}$

We apply algorithm (1) to WN and compare its convergence behavior with that of constraining $\mathbf{\Theta}$ to be diagonal and nonnegative.

In this experiment, we set $m = 10, r = 5, r_A = 3, \kappa = 3$, and $n = 1000$. The ground-truth matrix $\mathbf{A} \in \mathbb{R}^{m \times m}$ is constructed as $\mathbf{A} = \mathbf{U \Sigma U}^\top$, where $\mathbf{U} \in \mathbb{R}^{m \times r_A}$ is a random orthonormal matrix and $\mathbf{\Sigma} \in \mathbb{S}_+^{r_A}$ is a diagonal matrix with entries evenly distributed on a logarithmic scale in the interval $[1/\kappa, 1]$. Feature matrices $\{\mathbf{M}_i\}_{i=1}^n$ are generated as described in G.1.

We initialize $\mathbf{X}_0 = \mathbf{Z}_0(\mathbf{Z}_0^\top \mathbf{Z}_0)^{-1/2}$ and $\mathbf{\Theta}_0 = \mathbf{I}_r$ for both settings, where $\mathbf{Z}_0 \in \mathbb{R}^{m \times r}$ consists of i.i.d. standard Gaussian entries. We adopt stepsizes of $\eta = 0.1$ for updating $\mathbf{X}$ and $\mu = 2$ for updating $\mathbf{\Theta}$. To enforce $\mathbf{\Theta}$ to be diagonal and nonnegative in the "Diagonalized" setting, we perform an SVD on $\mathbf{\Theta}_t$ after each update, extract the diagonal matrix, and apply hard thresholding to ensure that all the diagonal entries are nonnegative.

