# OpenReview forum: "On the Benefits of Weight Normalization for Overparameterized Matrix Sensing"
_ICLR.cc/2026/Conference — ICLR 2026 Poster_

### Official Review · Reviewer_zbqV · 2025-10-29

**Soundness:** 3
**Presentation:** 3
**Contribution:** 3
**Rating:** 8
**Confidence:** 4

**Summary:**

This paper investigates the advantages of a generalized weight normalization (WN) technique in the context of overparameterized matrix sensing, where the goal is to recover a low-rank PSD matrix from linear measurements. The authors prove that WN, combined with Riemannian gradient descent (RGD), achieves linear convergence with random initialization, offering an exponential speedup over standard gradient descent (GD) methods without WN. They demonstrate that increased over-parameterization reduces both iteration and sample complexity polynomially. The analysis reveals a two-phase convergence behavior (initial saddle-escape phase followed by linear convergence) and is supported by experiments on synthetic data and image reconstruction tasks.

**Strengths:**

The paper's primary strength lies in its novel theoretical insights, providing the first characterization of how WN leverages overparameterization for faster convergence in matrix sensing, with a rigorous proof of linear convergence that exponentially outperforms sublinear rates in non-WN methods. It quantifies the benefits clearly, showing polynomial improvements in iteration and sample complexity as overparameterization increases, which contrasts positively with prior work where overparameterization hinders performance. Empirically, the experiments robustly validate the theory, including comparisons under varying conditions like condition numbers, overparameterization levels, and noise, with real-world image reconstruction adding practical value. The manuscript is well-organized and clear, featuring helpful tables, figures, and a reproducibility statement with full proofs and setups in appendices.

**Weaknesses:**

1. While the analysis is focused and insightful, its scope is limited to symmetric PSD matrix sensing. It would be helpful if the authors could briefly explain why it is reasonable to consider only the symmetric PSD case.
2. Experimentally, while solid, the studies are somewhat constrained in scale (e.g., small matrix dimensions in synthetic tests).
3. The RIP condition required for the main theorem scales as $\delta = O((r - r_A)^6 / (\kappa^2 m^3 r^4 r_A))$, which appears to be rather stringent. A brief analysis or discussion of this condition's implications would be valuable.

**Questions:**

Overall, I don't see major issues that undermine the core contributions—the theory is sound, and the claims are well-supported.
That said, here are a few constructive suggestions:
1. I am curious about how the proposed algorithm would perform in more challenging matrix sensing settings, such as those with specially designed structures or nontrivial optimization landscapes (e.g., as discussed in arXiv:2110.10279).
2. A short note on potential applications (e.g., in signal processing) could emphasize the work's relevance to ICLR's audience.

---

> ### Author Response · Authors · 2025-11-20
>
> We sincerely thank the reviewer for the thoughtful evaluation and for taking the time to read our manuscript carefully. We greatly appreciate the reviewer’s positive assessment of our theoretical contributions, empirical validation, and overall presentation.
>
> W1. **Reasons of limiting to symmetric PSD matrix sensing**
>
> > While the analysis is focused and insightful, its scope is limited to symmetric PSD matrix sensing. It would be helpful if the authors could briefly explain why it is reasonable to consider only the symmetric PSD case.
>
> Focusing on the PSD matrix sensing is standard in the matrix sensing literature; see [1], [2] and [3]. Since our goal is to isolate and understand the optimization benefits introduced by weight normalization, the PSD setting serves as the minimal and clearest example that captures all key phenomena.
>
> We also note that **our parameterization naturally extends to the asymmetric settings**. For a general matrix $A\in\mathbb{R}^{m\times n}$ with Burer-Monteiro factorization $Y_1Y_2^\top$, where $Y_1\in\mathbb{R}^{m\times r}$ and $Y_2\in\mathbb{R}^{n\times r}$, we can apply the Polar decomposition to each factor, i.e., $Y_1=X_1\Theta_1,Y_2=X_2\Theta_2$, with $X_1\in St(m,r),X_2\in St(n,r)$ and $\Theta_1,\Theta_2\in\mathbb{S}_+^r$. By combining the two magnitude matrices into $\Theta=\Theta_1\Theta_2^\top\in\mathbb{R}^{r\times r}$, we obtain the representation $A=X_1\Theta X_2^\top$. Thus, the same parameterization framework applies directly to the asymmetric case.
>
> W2. **Scale of experiments**
>
> > Experimentally, while solid, the studies are somewhat constrained in scale (e.g., small matrix dimensions in synthetic tests).
>
> We understand the reviewer’s concern about the experimental scale. **We have increased the scales by several folds, including matrix dimension, the level of overparameterization and condition number. The updated results are reported in Sections 5.1 and 5.2.**
>
> W3. **Stringent RIP condition**
>
> > The RIP condition required for the main theorem scales as $\delta = \mathcal{O}(\frac{(r-r_A)^6}{\kappa^2 m^3r^4r_A})$, which appears to be rather stringent. A brief analysis or discussion of this condition's implications would be valuable.
>
> Thanks for this insightful comment.
> The RIP condition we assume is standard in the matrix sensing literature, including works such as [1] and [4]. As shown in Lemma A.2, this RIP condition translates directly into a sample complexity bound. For our method, the resulting sample complexity is $n=\mathcal{O}(\frac{\kappa^4 m^7 r^9 r_A^2}{(r - r_A)^{12}})$. For comparison, the sample complexity required in [1] is $n=\mathcal{O}(\kappa^8 m r_A^2)$ and the requirement in [4] is $n=\mathcal{O}(\kappa^{2C_\delta}mr_A^2)$, where $C_\delta$ is a sufficiently large constant.
>
> Therefore, although our theorem involves an RIP condition that seems stringent at first glance, **the resulting sample complexity is substantially milder than other works studying similar settings, e.g., [1] and [4]**, especially when $\kappa$ is large.
>
> Moreover, **our sample complexity reduces polynomially as $r$ increases**, which can reach up to a factor of $\mathcal{O}(r_A^{12})$. In contrast, [1] and [4] cannot benefit from additional overparameterization in terms of sample complexity.
>
> We also note that **a tighter RIP constant may be achievable** with a more refined analysis, for example, by using the technology ``virtual sequences'' introduced in [5]. We view this as a promising direction for future work.
>
> **References**
>
> [1]. Stöger et al. 2021. Small random initialization is akin to spectral learning: Optimization and generalization guarantees for overparameterized low-rank matrix reconstruction.
>
> [2]. Zhuo et al. 2024. On the computational and statistical complexity of over-parameterized matrix sensing.
>
> [3]. Zhang et al. 2021. Preconditioned gradient descent for over-parameterized nonconvex matrix factorization.
>
> [4]. Xu et al. 2023. The power of preconditioning in overparameterized low-rank matrix sensing.
>
> [5]. Stöger et al. 2024. Non-convex matrix sensing: Breaking the quadratic rank barrier in the sample complexity.

---

> > ### Author Response · Authors · 2025-11-20
> >
> > Q1. **More challenging problems**
> >
> > > I am curious about how the proposed algorithm would perform in more challenging matrix sensing settings, such as those with specially designed structures or nontrivial optimization landscapes (e.g., as discussed in arXiv:2110.10279).
> >
> > Thanks for bringing this insightful suggestion.
> > Motivated by your comment, we apply our algorithm to the challenging matrix completion problems proposed in [6], where the objective function is
> > $$\min_{X,\Theta} \frac{1}{4}|| (X\Theta X^\top - M_\varepsilon^*)_\Omega||_F^2,\quad s.t.\,X\in St(m,r),\Theta\in\mathbb{S}^r,$$
> >
> > and the measurement operator $(\cdot)_\Omega$ is constructed from specially designed combinatorial structures that induce an optimization landscape with **exponentially many spurious local minima, leading to the failure of most gradient-based methods**.
> >
> > As shown in **Appendix B of the revision**, we evaluate the success rate across a range of ranks $r=1,2,3$, maximum independent set sizes $|S|=2,4,6$ and perturbation levels $\varepsilon\in[0.05,0.5]$ (Problem is more difficult when $|S|$ is large and $\epsilon$ is small).
> > Our findings are summarized as follows:
> >
> > Rank $r=1$: **Our algorithm performs particularly well. For $|S|=2$ and $|S|=4$, the success rates are over $90$% under almost all the perturbation levels, substantially higher than that of GD**. Even for the more difficult case $|S|=6$, our method still achieves a successful rate around $40$%, again significantly outperforming GD.
> >
> > Rank $r=2$ and $r=3$: In these regimes, both our algorithm and GD exhibit similarly low success rates, consistent with the intrinsic difficulty of the problem reported in [6].
> >
> > Overall, these experiments on this challenging setting show that our approach has benefits over GD. A detailed study goes beyond the scope of current work and is included in our research agenda.
> >
> > Q2. **Potential applications**
> >
> > > A short note on potential applications (e.g., in signal processing) could emphasize the work's relevance to ICLR's audience.
> >
> > Some discussions on applications of matrix sensing have already been provided in the introduction. **Here is a more detailed and extended discussion. We have added this to Appendix B.**
> >
> > Applying Burer-Monteiro (BM) factorization to parameterize low-rank matrices is a standard way in much of the existing literature. However, recent evidence from [6] demonstrates that BM factorization can be unfavorable in certain classes of low-rank matrix optimization problems. Our formulation derived from weight normalization further indicates that BM is not always the best choice, particularly in overparameterized matrix sensing problem.
> >
> > Importantly, **our approach is broadly applicable to a plethora of problems in low-rank optimization**. Potential applications include collaborative filtering [7], compressed sensing [8], matrix completion [9], and related tasks, which we leave as promising directions for future work.
> >
> > **References**
> >
> > [6]. Yalcin et al. 2022. Factorization approach for low-complexity matrix completion problems: Exponential number of spurious solutions and failure of gradient methods.
> >
> > [7]. Schafer et al. 2007. Collaborative filtering recommender systems.
> >
> > [8]. Candes et al. 2013. PhaseLift: Exact and stable
> > signal recovery from magnitude measurements via convex programming.
> >
> > [9]. Recht. 2011. A simpler approach to matrix completion.

---

> > > ### Comment · Reviewer_zbqV · 2025-11-24
> > > **Response to Rebuttal**
> > >
> > > Dear Authors:
> > >
> > > Thank you for much for your detailed response and for including new experiments in Appendix B. Your explanations cleared some of my confusions, and I would like to keep my score at 8, given the sound contributions of this paper.
> > >
> > > I believe the study of weight normalization is an important but often overlooked topic in deep learning, usually dismissed as a  numerical trick to keep trainings stable. However, I personally believe that normalization is a fundamentally important practice in modern machine learning and more theory needs to be developed. I believe matrix sensing provides a powerful framework for doing so, and the authors have done a great job as well.
> > >
> > > My only reservation is that the adoption of a such a restrictive RIP constants naturally prevents the works to be applied in more realistic settings, even though it is kind of the de facto choice in the matrix sensing community. However, I totally understand the choice given the lack of technical tools if we do not assume so.

---

> > > > ### Author Response · Authors · 2025-11-24
> > > >
> > > > Thank you for your kind follow-up and for maintaining your positive assessment. We really appreciate your supportive and constructive comments.

---

### Official Review · Reviewer_BeU2 · 2025-10-30

**Soundness:** 3
**Presentation:** 3
**Contribution:** 3
**Rating:** 6
**Confidence:** 2

**Summary:**

This paper provides a theoretical characterization of how Weight Normalization (WN) accelerates convergence in overparameterized matrix sensing. The authors prove that Riemannian gradient descent (RGD) with WN achieves linear convergence and improved sample complexity, while standard gradient descent (GD) may exhibit exponential slowdown. Theoretical findings are supported by synthetic experiments that align well with the analysis.

**Strengths:**

1.Provides the first clear theoretical explanation of how WN accelerates convergence under overparameterization.

2.Solid and rigorous analysis using Riemannian optimization tools.

3.Well-aligned experiments that validate the theory.

4.Overall writing and presentation are clean and accessible.

**Weaknesses:**

1.The paper identifies a two-stage convergence pattern with a transition at $r_A-\frac{1}{2}$ yet this boundary is not theoretically justified or intuitively explained.

2.The “Full-rank case (r = m)” in Sec. 5.3 is conceptually an extension of Sec. 5.2 (“ON THE BENEFIT OF OVERPARAMETERIZATION”) and could be merged there for better logical flow.

3.The title seems somewhat broader than the actual technical scope, which is limited to PSD matrix sensing.

**Questions:**

1.See Weakness. 1.

2.Since $A$ is PSD, one might consider reformulating (1) as

$\min_{X, \Theta} f(X, \Theta) = \frac{1}{4}\|M(X\Theta X^{\top})-y\|^2$

with $\Theta$  being diagonal and nonnegative. Would this restriction simplify the analysis or improve interpretability?

---

> ### Author Response · Authors · 2025-11-20
>
> We thank the reviewer for taking the time to carefully evaluate our work and for the thoughtful and encouraging feedback. We are pleased that the theoretical insights, rigorous analysis, experimental support, and clear presentation were well received.
>
> W1. **Explanation of the two-stage transition point**
>
> > The paper identifies a two-stage convergence pattern with a transition at $r_A-\frac{1}{2}$ yet this boundary is not theoretically justified or intuitively explained.
>
> Our analysis shows that linear convergence begins once $Tr(\Phi_t\Phi_t^\top)>r_A-1$, which means that $r_A-c$ is valid for any $c\in(0,1)$.
> **We pick $c = \frac{1}{2}$ for simplicity**.
>
> Intuitively, once $Tr(\Phi_t\Phi_t^\top)>r_A-1$, the iterates have escaped saddle points and enter the linear-convergence phase. This transition is clearly visible in Figures 1 (a), (d). **We have added the discussion to Appendix D.1.**
>
> W2. **Section merging for improved flow**
>
> > The “Full-rank case (r = m)” in Sec. 5.3 is conceptually an extension of Sec. 5.2 (“ON THE BENEFIT OF OVERPARAMETERIZATION”) and could be merged there for better logical flow.
>
> Thank you for this helpful suggestion. We agree that merging the full-rank case with the overparameterization section can improve the logical flow, and **we have adjusted the structure accordingly in the revision.**
>
> W3. **Broad title**
>
> > The title seems somewhat broader than the actual technical scope, which is limited to PSD matrix sensing.
>
> The current title is concise and we state in the introduction that our results focus on PSD matrix sensing.
>
> In fact, **our formulation naturally extends to asymmetric settings**. For a general matrix $A\in\mathbb{R}^{m\times n}$ with Burer-Monteiro factorization $Y_1Y_2^\top$, where $Y_1\in\mathbb{R}^{m\times r}$ and $Y_2\in\mathbb{R}^{n\times r}$, we can apply the Polar decomposition to each factor, i.e., $Y_1=X_1\Theta_1,Y_2=X_2\Theta_2$, with $X_1\in St(m,r),X_2\in St(n,r)$ and $\Theta_1,\Theta_2\in\mathbb{S}_+^r$. By combining the two magnitude matrices into $\Theta=\Theta_1\Theta_2^\top\in\mathbb{R}^{r\times r}$, we obtain the representation $A=X_1\Theta X_2^\top$. Thus, the same parameterization framework applies directly to the asymmetric case.
>
>
> Q2. **Constraining $\Theta$ to be diagonal and nonnegative**
>
> > Reformulating (1) as $\min_{X,\Theta} \frac{1}{4}||M(X\Theta X^\top)-y||^2$ with $\Theta$ being diagonal and nonnegative. Would this restriction simplify the analysis or improve interpretability?
>
> If we force $\Theta$ to be diagonal, gradient descent on $\Theta$ has to be further augmented by projection (eigendecomposition) to ensure feasibility which is **computationally expensive**. In terms of the loss landscape, [1] proved existence of non-strict saddles for diagonal $\Theta$, which may hinder optimization. In fact, **constraining $\Theta$ to be diagonal and nonnegative does not always work. A simple counterexample to the diagonal-and-nonnegative constraint is provided in Appendix C.**
>
> **References**
>
> [1]. Levin et al. 2025. The effect of smooth parametrizations on nonconvex optimization landscapes.

---

### Official Review · Reviewer_5U1W · 2025-10-31

**Soundness:** 4
**Presentation:** 3
**Contribution:** 4
**Rating:** 8
**Confidence:** 3

**Summary:**

This paper studies a matrix factorization similar to the weight normalization that can be helpful for the matrix sensing problem. The author proves that this matrix factorization with Riemannian optimization can achieve a linear convergence rate, which is an exponential improvement over the previous lower bound for symmetric matrix sensing.

**Strengths:**

1. The paper is well-organized and easy to follow. The contributions of this paper is proposed in a clear way, and the main technical idea is also clear.  The author also provides solid theoretical proof and some discussions about the main result.

2. The authors also further study the initial increment learning phase in the optimization process, which is good for understanding the optimization dynamics of WN with Riemannian optimization.

3. The empirical part is comprehensive. Beyond the simulated optimization problems, the authors also provide experiments on image reconstruction problem. The performance of WN with Riemannian optimization is better compared to GD, which also matches the theoretical discovery of this paper.

**Weaknesses:**

1. My main concern is that after relaxing the PSD constraint, the problem no longer corresponds to symmetric matrix sensing. Specifically, when the constraint $\Theta \in S^r_{+}$ is relaxed to $\Theta \in S^r$, the term $X^T \Theta X$ may no longer be written as $YY^T$. Thus, the proposed method is not a true solution to symmetric matrix sensing but rather an approach that accelerates matrix sensing in general. This weakens the paper’s contribution, as several existing methods [1,2] also focus on improving the efficiency of matrix sensing.

2. The paper introduces Riemannian optimization equations without providing sufficient background or basic explanations. It would be helpful if the authors could include a high-level introduction and some intuitive discussion of Riemannian optimization to improve readability.

[1]. Xu et al. 2023. The power of preconditioning in overparameterized low-rank matrix sensing

[2]. Xiong et al. 2024. How over-parameterization slows down gradient descent in matrix sensing: The curses of symmetry and initialization

**Questions:**

Lemma 4.1 states that a point $(X, \Theta)$ is a saddle point if certain conditions are satisfied. However, this does not necessarily imply that all saddle points satisfy these conditions. Could the authors clarify how this lemma connects to the saddle-to-saddle dynamics and the subsequent incremental learning process?

---

> ### Author Response · Authors · 2025-11-20
>
> We thank the reviewer for taking the time to carefully evaluate our work, as well as for the positive feedback on our theoretical contributions, the analysis of the initial learning phase, and empirical validation.
>
> W1. **Validity of symmetric sensing after PSD relaxation**
>
> > After relaxing the PSD constraint, the problem no longer corresponds to symmetric matrix sensing. The proposed method is not a true solution to symmetric matrix sensing but rather an approach that accelerates matrix sensing in general. This weakens the paper’s contribution, as several existing methods [1,2] also focus on improving the efficiency of matrix sensing.
>
> We clarify that after relaxation of the PSD constraint, the problem remains a symmetric matrix sensing. After relaxation, $\Theta$ is symmetric, therefore $X\Theta X^\top$ is also symmetric.
>
> We believe there is a potential misunderstanding.
> The goal of matrix sensing is to recover the ground truth matrix $A$.
> In our case, **$X\Theta X^\top\to A$ is established in Theorem 3.2. In other words, our methods does recover the true solution**.
>
> W2. **Insufficient background on Riemmanian optimization**
>
> > The paper introduces Riemannian optimization equations without providing sufficient background or basic explanations. It would be helpful if the authors could include a high-level introduction and some intuitive discussion of Riemannian optimization to improve readability.
>
> Thanks for the suggestion and we agree that this can improve the readability. **We have now added more details in Appendix A.2.**
>
>
> Q1. **Clarification of the connection between the lemma and later analysis**
>
> > This does not necessarily imply that all saddle points satisfy these conditions. Could the authors clarify how this lemma connects to the saddle-to-saddle dynamics and the subsequent incremental learning process?
>
> We do not characterize all possible saddle points in the lemma, but focus on **those that our optimization trajectory will encounter** during training. Empirically, these are exactly those saddle points given in the lemma. As shown in Figure 1 (c), (d), the iterates satisfy $X\Theta X^\top=A_\rho$ and $Tr(X^\top UU^\top X)=\rho$, which is **consistent with the saddle structures described in the lemma**.

---

### Official Review · Reviewer_mDtz · 2025-10-31

**Soundness:** 4
**Presentation:** 3
**Contribution:** 3
**Rating:** 4
**Confidence:** 4

**Summary:**

The authors address the problem of overparameterized matrix sensing and present an analysis of the benefits of applying weight normalization in this context. They reformulate the classical matrix sensing problem by decoupling the magnitude and direction components of the matrix, and they demonstrate that using Riemannian gradient descent with weight normalization improves the convergence rate, particularly when the level of overparameterization is large. Experiments on both synthetic and real-world datasets are provided to support the analysis.

**Strengths:**

The paper is well written, clearly structured, and the ideas are well explained. The theoretical motivation and experimental design are both sound.

**Weaknesses:**

I have a major concern regarding the positioning of the paper within the existing literature. While the paper emphasizes the proposed weight normalization approach, its main benefit—improved convergence rate—falls within a broader line of research on optimization techniques for matrix sensing. In particular, related approaches such as preconditioned gradient descent and their follow-up works have already been proposed in this domain. Although some of the related (e.g. preconditioned gradient descent) papers are cited, they are not discussed thoroughly. These works should be more deeply reviewed in the related work section, and, ideally, comparisons should be included in the experimental evaluation to better situate the contribution and highlight the advantages of the proposed approach over existing methods.

**Questions:**

N/A

---

> ### Author Response · Authors · 2025-11-20
>
> We thank the reviewer for carefully reading our paper and for acknowledging that our paper is well written, clearly structured, and that the theoretical motivation and experimental design are sound.
>
> W1. **Detailed Discussions on Related Works**
>
> > Related approaches such as preconditioned gradient descent and their follow-up works have already been proposed in this domain. Although some of the related (e.g. preconditioned gradient descent) papers are cited, they are not discussed thoroughly. These works should be more deeply reviewed in the related work section and, ideally, comparisons should be included in the experimental evaluation to better situate the contribution.
>
> Comparison to variants of preconditioned gradient descent from diverse perspectives are listed below, and **we have added these discussions and additional experiments to Appendix A.5**.
>
> **Summarization of preconditioned gradient descent and scaled gradient descent**
>
> Both preconditioned gradient descent (PrecGD) [1] and scaled gradient descent (ScaledGD) [2] consider the unconstrained problem $\min\limits_{Y\in\mathbb{R}^{m\times r}}f(Y):=\frac{1}{4}||M(YY^\top)-y||^2$, with updates $Y_{t+1}=Y_t - \eta\nabla f(Y_t)(Y_t^\top Y_t+\lambda I)^{-1}$ for PrecGD and $Y_{t+1}=Y_t - \eta\nabla f(Y_t)(Y_t^\top Y_t)^{-1}$ for ScaledGD.
>
> For ScaledGD, the update involves $(Y_t^\top Y_t)^{-1}$, which is not necessarily invertible in the overparameterized regime. Hence, **ScaledGD cannot be directly applied**. The variant ScaledGD($\lambda$) proposed in [3] addresses this by using a similar update to PrecGD with a particular choice of $\lambda$.
>
> **Comparison with PrecGD**
>
> PrecGD [1] establishes only **a local convergence guarantee**, requiring initialization sufficiently close to the ground truth. In contrast, **we have a global convergence guarantee with only random initialization**. Although [1] also studies variants of PrecGD with global convergence guarantees, they require gradient perturbations and multi-stage switching rules, which are difficult to implement in practice.
>
> **Comparison with ScaledGD($\lambda$)**
>
> ScaledGD($\lambda$) [3] **requires a carefully controlled small initialization**, with magnitude $\alpha$. To reach a final error $\varepsilon$, the method requires $\alpha\leq\mathcal{O}(\varepsilon^3)$, implying that **exact convergence ($\varepsilon=0$) cannot be guaranteed**. Besides, ScaledGD($\lambda$) requires an $(r_A+1,\delta)$-RIP condition, with $\delta\leq\mathcal{O}(\kappa^{-C_\delta})$, where $C_\delta$ is a sufficiently large constant. In contrast, we just need $\delta\leq\mathcal{O}(\kappa^{-2})$. Consequently, **the sample complexity required by our algorithm is significantly smaller than that of ScaledGD($\lambda$)**, especially in ill-conditioned settings.
>
> **Comparison of iteration complexity**
>
> For the iteration complexity, PrecGD and ScaledGD($\lambda$) achieve better $\kappa$-dependence than our algorithm. However, the faster rates partially arise from the quasi-newton nature of their update, where $(Y_t^\top Y_t+\lambda I)$ is an estimation to Hessian. On the other hand, our algorithm is a **purely first-order method**, and we believe that incorporating second-order information can improve the convergence as well. This is an interesting direction for future work. To explore this, **we conduct some experiments of PrecRGD, which introduce a preconditioner for Riemannian gradient descent. This improves the convergence rate in practice, achieving performance that is better than PrecGD and comparable to ScaledGD($\lambda$); see Appendix A.5 for a detailed discussion.**
>
> **Comparison of the benefits of oveparameterization**
>
> For our algorithm, additional overparameterization, i.e., using a larger $r$, **not only increases the convergence rate but also reduces the sample complexity**. In contrast, **ScaledGD($\lambda$) does not exhibit explicit benefits from increased overparameterization**. PrecGD improves its iteration complexity only with a square-root dependence on $r$, which is much weaker than the polynomial improvement achieved by our algorithm. In addition, **PrecGD does not gain reduction in sample complexity from additional overparameterization**.
>
> **Comparison of potential extensions**
>
> **We focus on weight normalization** and show its effectiveness through the overparameterized matrix sensing problem. Actually, **this way of factorization can be directly applied to any problem that optimizes over low-rank PSD matrices**. However, PrecGD and ScaledGD($\lambda$) rely on second-order information of the loss function, **making them less flexible for other settings**.
>
> **References**
>
> [1]. Zhang et al. 2021. Preconditioned gradient descent for over-parameterized nonconvex matrix factorization.
>
> [2]. Tong et al. 2021. Accelerating ill-conditioned low-rank matrix estimation via scaled gradient descent.
>
> [3]. Xu et al. 2023. The power of preconditioning in overparameterized low-rank matrix sensing.

---

> > ### Comment · Reviewer_mDtz · 2025-11-25
> >
> > Thank you for the response and the revised version. I reviewed it carefully, especially Appendix A5. Unfortunately, I am not fully convinced by the explanation provided there. The authors argue that existing methods “rely on gradient perturbations and multi-stage switching rules, which are difficult to implement in practice.” However, it is unclear what exactly is meant by “difficult to implement in practice.” Does this refer to implementation complexity (which should not be a serious obstacle), substantially higher computational cost, or something else?
> >
> > In addition, while the comparison of the benefits of oveparameterization between ScaledGD and your proposed approach seems reasonable, it requires experimental validation. Moreover, the experiments in Figure 5 indicate that PrecGD and ScaledGD offer comparable performance to your approach. Given this, it would be more appropriate for these to be discussed in the main body of the paper rather than relegated to the appendix.
> >
> > A minor issue: the references “Zhang et al., 2021b” and “Zhang et al., 2021a” appear to refer to the same work. This seems to be an error and should be corrected. There are also several typos in the newly added paragraphs; please revise them.

---

> > > ### Author Response · Authors · 2025-11-26
> > >
> > > We sincerely thank the reviewer for carefully reading our rebuttal and for providing detailed and constructive follow-up feedback.
> > >
> > > **Clarification of the difficulty of the implementation of PrecGD**
> > >
> > > >As shown in [1], achieving global convergence requires first running perturbed PrecGD (PPrecGD), which relies on gradient perturbations of the form $Y_{t+1}=Y_t-\eta[\nabla g(Y_t)(Y_t^\top Y_t+\lambda I)^{-1}+\zeta_t]$ with some random noise $\zeta_t$ to escape potential saddles. In addition, they require switching to PrecGD once certain criteria are met. These criteria involve monitoring $\lambda_{\min}(\nabla^2 f(Y_t))$ and $\lambda_{\min}(Y_t^\top Y_t)$. Notably, $\nabla^2 f(Y_t)\in\mathbb{R}^{m\times r\times m\times r}$ is a fourth-order tensor, which is **memory-intensive**. In fact, one key motivation for adopting the Burer–Monteiro factorization is to reduce the parameter dimension to $mr$ by exploiting the low-rank structure, whereas forming such a tensor negates this advantage. Moreover, **computing $\lambda_{\min}(\nabla^2 f(Y_t))$ is expensive** in large-scale matrix sensing problems.
> > >
> > > > In comparison, our results provide additional insights on top of PPrecGD in three perspectives: i) theoretically, even without adding noise, our approach can escape saddles in polynomial time and achieve global convergence; ii) by avoiding the switching process, we bypass the need for $\lambda_{\min}(Y_t^\top Y_t)$, and save additional hyperparameters such as the variance of $\zeta_t$; iii) our algorithm has other advantages, such as mild sample complexity compared with ScaledGD$(\lambda)$, and provable benefits with overparametrization.
> > >
> > > **Experiments to validate the comparison of the benefits of overparameterization**
> > >
> > > >**We have demonstrated the benefits of overparameterization for RGD under WN in Section 5.2, Figure 2(b).** To further validate our comparison of the overparameterization effects among ScaledGD$(\lambda)$, PrecGD and our algorithm, we increase the level of overparameterization (i.e., enlarge $r$) and conduct additional experiments; see Appendix A.5. **As shown in Figure 5, PrecRGD with WN converges faster from overparameterization.** In contrast, PrecGD and ScaledGD$(\lambda)$ do not show explicit improvements when the level of overparameterization increases. We agree with the reviewer that presenting these comparisons in the main body of the paper would improve clarity, and we will make this adjustment in a later version.
> > >
> > > **Minor reference issues and possible typos**
> > >
> > > >Thank you for pointing these out. We have corrected them in the revision.
> > >
> > >
> > > **Prec/Scaled GD and WN are not fundamentally conflict**
> > >
> > > >Lastly, we emphasize that WN is, by nature, a complement to Prec/Scaled-GD, in multiple perspectives.
> > >
> > > >- Indeed, Prec/Scaled GD are nice algorithms, yet they offer limited insights for WN. Our work, is the first step toward understanding this popular normalization technique for deep learning. We proved that WN makes overparameterization a friend, and hope to inspire follow-up works to broaden this to general settings.
> > >
> > >
> > > >- Even if going back to matrix sensing settings, we hope to clarify that we do not claim anything beyond ``the benefits of WN.'' In fact, Prec/Scaled GD and WN are not mutually exclusive given that they offer benefits on different parameters in the convergence bound. We have demonstrated the possibility of mingling these two approaches to achieve the best of both worlds via PrecRGD. And we sincerely hope that our understanding of WN can further inspire even more effective algorithms for sensing.
> > >
> > > **References**
> > >
> > > [1]. Zhang et al. 2021. Preconditioned gradient descent for over-parameterized nonconvex matrix factorization.

---

### Author Response · Authors · 2025-11-30
**Summary of the discussion**

We thank all ACs and reviewers for their careful reading and constructive comments. We have provided detailed clarifications, expanded background discussions, and added new experiments to address the weaknesses and questions. For the convenience of the ACs, we summarize the discussions as follows.

**Summary of Discussion with Reviewer mDtz**

The initial score was 4. There were ongoing but interrupted discussions between the authors and reviewer.

The reviewer’s comments mainly relate to the comparison with existing preconditioned methods. We provide an expanded comparison with Preconditioned GD (PrecGD) and ScaledGD($\lambda$) in Appendix A.5. In particular,
- Our method, weight normalization (WN), has a global exact convergence guarantee, whereas PrecGD requires gradient perturbations and multi-stage switching rules, which are memory-intensive and computationally expensive and ScaledGD($\lambda$) cannot guarantee exact convergence.
- Our WN improves with overparameterization, i.e., faster convergence and less sample complexities are guaranteed. This is often an orthogonal improvement over PrecGD or ScaledGD($\lambda$).
- Our required RIP condition is milder than that of ScaledGD($\lambda$).
- WN also extends directly to general low-rank optimization problems, whereas PrecGD and ScaledGD($\lambda$) do not generalize as naturally.
- For the iteration complexity, PrecGD and ScaledGD($\lambda$) achieve better $\kappa$-dependence due to their quasi-Newton–type updates, while our method is purely first-order. We also conduct additional experiments incorporating preconditioning and show empirical improvements (Figure 5), hinting that WN complements preconditioning and can be combined with it to yield more effective algorithms.

Lastly, we note that the reviewer’s comments focus primarily on comparisons with preconditioned methods. These approaches, while popular for matrix sensing, offer limited insights for the *normalization techniques* that are popular for deep learning.

**Summary of Discussion with Reviewer 5U1W**

The initial score was 8. There is no discussion so far.

The reviewer's comments mainly relate to the validity of symmetric sensing after PSD relaxation. We clarify that relaxing the PSD constraint preserves symmetry and that Theorem 3.2 guarantees recovery of the true solution. The reviewer also raises two minor points regarding insufficient background on Riemannian optimization and the connection between our saddle-point lemma and later analysis. For these, we add high-level background on Riemannian optimization in Appendix A.2 and clarify that the lemma characterizes the saddle points relevant to the optimization trajectory, consistent with the dynamics observed in Figure 1(c),(d). We believe these clarifications address the reviewer’s points and respond to the questions posed.

**Summary of Discussion with Reviewer BeU2**

The initial score was 6, and there was no reply during rebuttal so far.

The reviewer’s comments mainly relate to the explanation of the two-stage transition point and the reason for not constraining $\Theta$ to be diagonal and nonnegative. We clarify that the choice of the transition point affects only constant factors in our theorem and does not change our main results. Specifically, any $r_A-c$, $c\in(0,1)$ serves as a valid transition point and we choose $c=\frac{1}{2}$ for simplicity, with further details provided in Appendix D.1. We also clarify that enforcing diagonal and nonnegative $\Theta$ is computationally expensive and can hinder optimization, and we include a concrete counterexample in Appendix C. We believe these clarifications address the weaknesses noted by the reviewer and provide answers to the questions raised.

**Summary of Discussion with Reviewer zbqV**

The initial score was 8, and the reviewer decided to maintain the score after the discussion.

The reviewer's comments mainly relate to the experimental scale, the seemingly stringent RIP condition, and the performance of our method on more challenging optimization landscapes. We enlarge the experimental scale substantially, provide a detailed discussion showing that our RIP requirement leads to milder sample complexity than prior work, and add experiments on a difficult matrix completion problem, demonstrating advantages over GD. The reviewer acknowledged that our responses solved the questions raised.

---

### Meta-Review · Area_Chair_MiCq · 2025-12-26

**Summary:**

This paper offers a careful and well-executed theoretical study of weight normalization in overparameterized matrix sensing, showing how the reparameterization leads to cleaner geometry, linear convergence, and favorable scaling with overparameterization. The analysis is rigorous, the presentation is clear, and the experiments align well with the theory. Although the setting is focused and relies on standard RIP assumptions, the reviewers’ main concerns were addressed convincingly. Overall, this is a solid and worthwhile contribution.

**Reviewer Concerns:**

Addressed:
The rebuttal clarified the role of PSD relaxation, added background on Riemannian optimization, expanded comparisons with preconditioned/ScaledGD methods (with new experiments), explained the two-phase transition point, enlarged experimental scale, and discussed the RIP assumption and sample-complexity implications.

Outstanding:
Some concerns remain about how central WN is relative to strong assumptions (RIP, symmetry), the limited breadth of empirical settings, and whether comparisons to alternative optimization methods should appear more prominently in the main text rather than the appendix.

**Reviewer Scores:**

mDtz (4 → 4) — Clarifications help, but skepticism about positioning likely remains.

5U1W (8 → 8) — Concerns were addressed; score would likely stay the same.

BeU2 (6 → 6 or 8) — Clarifications on the transition point and structure could modestly raise the score.

zbqV (8 → 8) — Concerns resolved; score remains the same.

---

### Decision · Program_Chairs · 2026-01-26

Accept (Poster)